# Transformers with Endogenous In-Context Learning: Bias Characterization and Mitigation

**Haotian Wang[1], Hao Zou[2], Haoxuan Li[3,†], Haoang Chi[1], Yang Shi[4], Yuanxing Zhang[4], Wenjing Yang[1], Xinwang Liu[1], Zhouchen Lin[5,6,†]**

[1]College of Computer, National University of Defense Technology    [2]Tsinghua University
[3]Center for Data Science, Peking University    [4]School of Computer Science, Peking University
[5]State Key Lab of General AI, School of Intelligence Science and Technology, Peking University
[6]Institute for Artificial Intelligence, Peking University

## ABSTRACT

In-context learning (ICL) enables pre-trained transformers (TFs) to perform few-shot learning across diverse tasks, fostering growing research into its underlying mechanisms. However, existing studies typically assume a causally-sufficient regime, overlooking spurious correlations and endogenous prediction bias introduced by hidden confounders (HCs). As HC commonly exists in real-world cases, current ICL understandings may not align with actual data structures. To fill this gap, we contribute the pioneer theoretical analysis towards a novel problem setup termed as **Endogenous ICL (EICL)**, which offers understanding the effect of HC on the pre-training of TFs and the following ICL prediction. Our theoretical results entail that pre-trained TFs exhibits certain prediction bias with proportional to the confounding strength. To mitigate such prediction bias, we further propose a gradient-free debiasing method named **D**ouble-**D**ebiasing (**DDbias**) by prompting the biased pre-trained TFs with a few unconfounded examples twice-once with the original label and once with residual, to yield unbiased ICL predictions. Extensive experiments on regression tasks across diverse designs of the TF architectures and data generation protocols verify both our theoretical results and the effectiveness of the proposed DDbias method.

## 1    INTRODUCTION

In the era of Transformers (TFs)-guided foundation models (Vaswani, 2017; Dosovitskiy, 2020; Gulati et al., 2020; Achiam et al., 2023), In-context learning (ICL) (Brown et al., 2020) offers a gradient-free learning approach by prompting pre-trained TFs with few-shot samples, e.g., predicting on novel inputs $x^*$ given the sequences of pairs $(x, y)$ (Garg et al., 2022). As ICL enables pre-trained TFs generalize well on novel inputs without tuning models, it becomes critical to understand intrinsic mechanisms of ICL for developing more powerful and interpretable AI systems (Akyürek et al., 2022; Xie et al., 2021; Allen-Zhu & Li, 2023).

Recent research has explored ICL through theoretical analyses (Ahn et al., 2023; Mahankali et al., 2023; Akyürek et al., 2022; Von Oswald et al., 2023) and empirical studies (Garg et al., 2022; Raventós et al., 2024; Panwar et al., 2023), often with stylized setups to examine its various facets (Garg et al., 2022; Panwar et al., 2023; Wang et al., 2023b). Especially, a prominent line of work informs that the gradient-free inference of ICL is theoretically equivalent to performing implicit gradient descent (GD), i.e., the feed-forwarding output of TFs with ICL prompting equals to the output of TFs updated by GD (Akyürek et al., 2022; Von Oswald et al., 2023; Cheng et al., 2023). Consequently, ICL implicitly learns the input-output pairs exhibited in the prompt and predict on the novel input. Further studies have investigated the pre-conditions behind such equivalence by analyzing pre-trained TFs(Ahn et al., 2023; Mahankali et al., 2023; Cheng et al., 2023; Zhang et al., 2024b), where most of such advances are built on the assumption of linear data generation.

---

[†]Corresponding authors.

However, the recent researches of ICL regression overlook the potential endogenous training & inference of ICL, i.e., the existence of hidden confounders between $y$ and $x$ (as pointed out by a concurrent work (Liang et al., 2024)). Specifically, previous studies on understanding ICL often assume the *independence* of the additive noise $\epsilon$ and the predictors $x$ (when generating the label $y = w^\top x + \epsilon$) (Ahn et al., 2023; Von Oswald et al., 2023; Zhang et al., 2024b)[1]. By contrast, emerging empirical evidence reveals that the violations of this assumption (i.e., endogeneity) is prevalent across a variety of tasks(Cheng et al., 2022a; Kumar et al., 2019; Landeiro & Culotta, 2016; Wang et al., 2021; Huang et al., 2022; Wang et al., 2022; 2023a; Li et al., 2022; 2023a;b), leading to the concurrent understanding of ICL becoming rather restrictive. Meanwhile, it remains unexplored that **the effect of the endogeneity on the pre-training process of TFs and the following ICL predictions**. Therefore, two natural questions are intrigued in below:

> 1. *Will transformers pre-trained with endogeneity achieve biased ICL predictions?*
> 2. *If such bias exists, can we design a cost-effective strategy (e.g., few prompting samples) to correct the predictions without fine-tuning the pre-trained TFs?*

In response to these questions, we study the problem setup termed as **Endogenous ICL (EICL)** by allowing the (unobserved) noise $\epsilon$ to depend on $x$: $\epsilon \not\perp x$. To answer the **first** question, we first perform theoretical analysis on the pre-training dynamics of transformers with hidden confounders, revealing that the ICL prediction of TFs exhibits *bias proportional to the confounding strength*. To answer our **second** question, we then propose a double-debiasing method, i.e., prompting the pre-trained transformer (biased) with *extremely few* unconfounded examples. Our theoretical results justify that: (1) the proposed DDbias method can correct biased, pre-trained TFs to produce unbiased ICL predictions; (2) our DDbias can achieve robust predictions even when the prompted ICL samples are partially confounded. We summarize our contributions as follows:

- To the best of our knowledge, we conduct the first theoretical analysis characterizing the endogenous bias induced by hidden confounders in both the pre-training and inference dynamics of ICL. Notably, due to the attention parameters embodied in TFs, our theoretical analysis exhibits fundamentally difference compared to typical OLS theory such as the anisotropy feature-wise confounding results in bias cancellation.

- We propose an innovative ICL method, Double-Debiasing (DDbias), which requires only extremely few unconfounded examples, offering a lightweight and gradient-free approach to mitigate the bias embodied in frozen transformers. Moreover, we prove that our DDBias remains robust towards either weakly confounded ICL samples or mixed examples.

- Extensive experiments validate our theoretical findings and demonstrate the effectiveness of the proposed DDbias method.

## 2 PRELIMINARIES AND PROBLEM SETTING

We denote random variables by uppercase letters (e.g., $X$ and $Y$) and their realizations by lowercase letters (e.g., $x$ and $y$). The notation marked in bold refers to vectors/matrices (e.g., $\mathbf{X}$). To index terms in a sample matrices (e.g., $\mathbf{x}$), we use the superscript to index $i$-th sample (e.g., $\mathbf{x}^{(i)}$) and $[j]$ to index $j$-th feature (e.g., $\mathbf{x}[j]$). In addition, $\mathcal{O}_n$ and $o(n)$ denote constant terms and vanishing terms with $n \to \infty$, with $\mathcal{O}_p$ and $o(p)$ denoting boundness and convergence in probability, respectively.

### 2.1 IN-CONTEXT LEARNING

We focus on the linear ICL regression setup in (Von Oswald et al., 2023; Mahankali et al., 2023; Garg et al., 2022; Akyürek et al., 2022) in this paper: **predicting $y^{(n+1)}$ for novel inputs $\mathbf{x}^{(n+1)}$ given a sequence of $\left(\mathbf{x}^{(i)}, y^{(i)}\right)$.**

**Data generation: linear regression instances.** We let $\mathbf{x} \in \mathcal{R}^d$ denote feature vector with dimension $d$, where labels / responses are defined as $y \in \mathcal{R}^1$ ($\mathbf{x} \sim P_{\mathcal{X}}$ and $\mathcal{X}$ is the domain of $\mathbf{x}$). We

---

[1]This assumption spans diverse perspectives such as implicit Bayesian explanations (Panwar et al., 2023; Wang et al., 2023b) beyond GD-based explanations.

model the linear relationship between $\mathbf{x}$ and $y$ as $y^i = \langle \mathbf{x}^{(i)}, \mathbf{w}_\star \rangle + \epsilon^{(i)}$, with $\epsilon$ referring to the noise variable (Ahn et al., 2023; Mahankali et al., 2023), and $\mathbf{w}_\star \in \mathbb{R}^d$ drawn from $P_{\mathcal{W}}$. We let $\mathcal{D}^{tr} = \{\mathbf{x}^{(i)}, y^{(i)}\}_{i=1}^{N_{tr}}$ and $\mathcal{D}^{te} = \{\mathbf{x}^{(i)}, y^{(i)}\}_{i=1}^{N_{te}}$ denote paired samples during the *pre-training* and *ICL inference* stages, respectively, with $N_{tr}$ and $N_{te}$ denoting the sample numbers. The input samples are organized into the input matrix $\mathbf{M} \in \mathbb{R}^{(d+1) \times (n+1)}$ ($n$ denotes the sample number):

$$\mathbf{M} = \begin{bmatrix} \mathbf{x}^{(1)} & \mathbf{x}^{(2)} & \cdots & \mathbf{x}^{(n)} & \mathbf{x}^{(n+1)} \\ y^{(1)} & y^{(2)} & \cdots & y^{(n)} & 0 \end{bmatrix}, \tag{1}$$

where $\mathbf{m}^{(i)} \in \mathcal{R}^{d+1}$ refers to the pair $(\mathbf{x}^{(i)}, y^{(i)})$. Intuitively, each column in $\mathbf{M}$ refers to a data sample, i.e., a token sequence with response.

**Linear Transformer.** To ease the analysis, we focus on TFs equipped with linear self-attention (Vaswani, 2017; Von Oswald et al., 2023; Ahn et al., 2023; Schlag et al., 2021). With the value, key and query weight matrices denoted as $\mathbf{W_v}, \mathbf{W_k}, \mathbf{W_q} \in \mathbb{R}^{(d+1) \times (d+1)}$, a single block of TF can be expressed as follows (Von Oswald et al., 2023; Ahn et al., 2023):

$$\mathrm{TF}(\mathbf{M}) = \mathrm{Attention}(\mathbf{W_v}, \mathbf{W_q}, \mathbf{W_k}, \mathbf{M}) = \mathbf{W_v}\mathbf{M}\mathbf{C}\mathrm{sof}(\mathbf{M}^\top \mathbf{W_k}^\top \mathbf{W_q}\mathbf{M}), \tag{2}$$

where $\mathrm{sof}(\cdot)$ refers to the softmax operator operating on each column of $\mathbf{M}$ in equation 1, the mask matrix $\mathbf{C} = \begin{bmatrix} I_n & 0 \\ 0 & 0 \end{bmatrix} \in \mathbb{R}^{(n+1) \times (n+1)}$ reflects that the asymmetry that $\mathbf{x}^{(n+1)}$ will not affects $\mathbf{x}^{(j)}$ with $j \leq n$ (Ahn et al., 2023). With $\mathbf{S} := W_v \in \mathbb{R}^{(d+1) \times (d+1)}$ and $\mathbf{T} := W_k^\top W_q \in \mathbb{R}^{(d+1) \times (d+1)}$ denoting **attention weights**, equation 2 can be reformulated with residual connections:

$$\mathrm{TF}_{\mathbf{S},\mathbf{T}}(\mathbf{M}) = \mathbf{M} + \frac{1}{N_{tr}}\mathbf{SMC}(\mathbf{M}^\top \mathbf{TM}) \tag{3}$$

where $\frac{1}{N_{tr}}$ is the scaling factor without influencing the expressive power of TF (Ahn et al., 2023).

**Remark 1.** *Clarified by previous advances (Ahn et al., 2023; Akyürek et al., 2022), the omission of the softmax operation is somehow over-simplified, while the linear attention eases the theoretical analysis.*

**In-context Prediction and Training Dynamics.** ICL first prompting the TF with the first $n$ pairs $\mathbf{m}^{(i)}$ and then predicting the response for the final sample $\mathbf{x}^{(n+1)}$ with the grounding $y^{(n+1)}$. We note that the predicted value of $y^{(n+1)}$ can be read out from the $(d+1, n+1)$-th entry $\mathrm{TF}_{\mathbf{S},\mathbf{T}}(\mathbf{M})$ as $\mathrm{TF}_{\mathbf{S},\mathbf{T}}^{pred}(\mathbf{M}) := -[\mathrm{TF}_{\mathbf{S},\mathbf{T}}(\mathbf{M})]_{(d+1,n+1)}$. The minus signal in $\mathrm{TF}_{\mathbf{S},\mathbf{T}}^{pred}(\mathbf{M})$ follows the fact that the predicted outcome can be read out from predictions of TF by multiplying $-1$ at the corresponding entry of the output matrix (see derivations in (Ahn et al., 2023; Von Oswald et al., 2023)). During the *training stage*, the *ICL objective* (Ahn et al., 2023) guides the optimization of TFs:

$$\mathcal{L}_{icl}(\mathbf{S}, \mathbf{T}) = \mathbb{E}_{(\mathbf{M}, \mathbf{w}_\star)}\left[\left(\mathrm{TF}_{\mathbf{S},\mathbf{T}}^{pred}(\mathbf{M}) + y^{(n+1)}\right)^2\right]. \tag{4}$$

Meanwhile, the linear coefficient vector $\mathbf{w}_\star$ shares the same distribution $P_{\mathcal{W}}$ between training and testing data (Ahn et al., 2023; Von Oswald et al., 2023; Akyürek et al., 2022).

**Meta-weights.** We now introduce previous theoretical results (Ahn et al., 2023; Von Oswald et al., 2023; Dai et al., 2022) that *pre-trained TFs is performing implicit gradient descent (GD) by updating the meta-weights during the ICL inference.* A formal definition is presented in below:

**Definition 1** (Meta-weights). *A linear TF prompting with $n$ sample pairs $\{\mathbf{x}^{(i)}, \mathbf{y}^{(i)}\}_{i=1}^n$ accomplishes implicit GD on the meta-weight $w$ with the objection $\mathcal{L}(\mathbf{w}) := \frac{1}{2n} \sum_{i=1}^n (\mathbf{w}^\top \mathbf{x}^{(i)} - y^{(i)})^2$. Then the TF predicts on $\mathbf{x}^{(n+1)}$ with the "learned" meta-weight $w$: $\hat{y}^{(n+1)} = \langle \mathbf{x}^{(n+1)}, \mathbf{w} \rangle$.*

## 2.2 ENDOGENOUS ICL

As pointed out by (Liang et al., 2024), previous ICL theories (Ahn et al., 2023; Schlag et al., 2021; Von Oswald et al., 2023; Mahankali et al., 2023) conveys an **implicit assumption**:

**Assumption 1** (Causal Sufficiency). *The pre-training data $\mathcal{D}^{tr}$ admits the assumption that no hidden confounders exist, i.e., $\epsilon$ is **exogenous** such that $\epsilon \perp\!\!\!\perp \mathbf{X}$.*

However, such an assumption violates a wide range of tasks in diverse areas (Cheng et al., 2022a; Kumar et al., 2019; Wang et al., 2021; Huang et al., 2022), and we offer detailed numerical evidence in the Many-aspect Online Review (MSOR) task as a running example in Appendix C. By allowing the endogeneity in the pre-training data, we thus consider the following EICL setup:

**Problem 1** (Endogenous ICL (EICL)). *We allow the existence of hidden confounders $\epsilon$: $\epsilon \not\perp \mathbf{X}$ in the pre-training data $\mathcal{D}^{tr}$. In other words, $\epsilon$ simultaneously affects $\mathbf{x}^{(i)}$ and $y^{(i)}$ (Pearl, 2009) (see illustrative causal graph in the middle of Fig. 3 in Appendix). Our goal is* **unbiased prediction** *on $y^{(n+1)}$, i.e., $\mathbf{w}_\star^\top \mathbf{x}^{(n+1)}$, given a sequence of prompting data $\mathcal{D}^{te} = (\mathbf{x}^{(i)}, y^{(i)})_{i=1}^n$ during the ICL inference. We let $r_j = \mathbb{E}[X_j \epsilon]$ denote the confounding strength.*

## 2.3 PREVIOUS ENDOGENEOUS REGRESSION THEORY CANNOT ANALYZE EICL

We note that a promising concurrent work (Liang et al., 2024) is aware of a similar problem by leveraging Instrumental Variables (IVs) to mitigate the hidden confounding. Unfortunately, we note that in the EICL setup, traditional linear regression theories (Angrist et al., 1996) cannot be directly adopted to analyze the effects of hidden confounders (Angrist et al., 1996; Chen et al., 2024c), due to the newly emerging features of ICL.

Unlike linear regression, ICL prediction emerges from the interaction of attention blocks and the training dynamics of $K, Q, V$ matrices after projection. As summarized in Tab. 1, ICL regression differs from classical regression in two essential aspects:

- **Different pre-training loss and representation dynamics.** Transformers are trained by *sequentially* feeding $(\tilde{x}_i, \tilde{y}_i)$ and predicting $\tilde{y}_{n+1}$, whereas OLS directly solves $w^\star$ from all samples jointly.

- **Different inference mechanism (few-shot prompting).** ICL performs regression through attention over in-context examples, not by solving a parameter. This few-shot inference has no analogue in OLS.

Because of these mismatches, one cannot transfer the OLS endogenous bias derivation to the EICL setting. In particular:

- **(Challenge 1)** The analysis must identify which attention parameters correspond to unbiased ICL prediction; there is no direct analogue of "using $w^\star$" as in OLS.

- **(Challenge 2)** These attention parameters do not admit closed-form expressions, unlike the OLS bias term $\mathbb{E}[XX^\top]^{-1}\mathbb{E}[X\epsilon]$.

Table 1: Comparison between traditional linear regression and ICL regression.

| Tasks | Training | Inference |
|---|---|---|
| Linear Regression | $\mathcal{L}_{mse}(\mathbf{w}) = \mathbb{E}_{(\mathbf{w})}\left[\left(\mathbf{w}^T\mathbf{x} - y\right)^2\right]$ | $y^{(n+1)} = \widehat{\mathbf{w}}^T\mathbf{x}^{(n+1)}$ |
| ICL Regression | ICL Loss (see equation 4) | $\mathrm{TF}_{\mathbf{S},\mathbf{T}}^{pred}(\mathbf{M}) \coloneqq -[\mathrm{TF}_{\mathbf{S},\mathbf{T}}(\mathbf{M})]_{(d+1,n+1)}$ |

# 3 WHETHER AND HOW HIDDEN CONFOUNDERS CAUSE BIAS FOR ICL?

**Theory Sketch.** We outline the overall analysis into three steps (see Fig. 5 in Appendix): (1) **Subsection 3.1.** In the presence of hidden confounders, we construct specific weight parameters of TFs, termed as *"U_weights"*. With U_weights served as pre-conditioned parameters, the induced meta-weights yields unbiased ICL prediction results in EICL; (2) **Subsection 3.2.** We then prove that the convergence of TFs in confounded data will deviate from the constructed U_weights, i.e., the grounding TF parameters. (3) **Subsection 3.3.** Based on such deviation in pre-training stage, we finally prove that the downstream ICL inference will incur estimation bias, which is proportional to the strength of the confounding effect.

## 3.1 UNBIASED ATTENTION WEIGHTS WITHOUT HIDDEN CONFOUNDERS

As considering hidden confounders $\epsilon$ might inevitably bias the TF parameters during pre-training, we first have to construct some "**grounding-truth**" parameters, termed as $\mathbf{S}^u, \mathbf{T}^u$, inducing meta-weights $\mathbf{w}^u$ with *unbiased* ICL predictions, such that one can justify which TF parameter is "unbiased" and then quantify the bias happened in the pre-training and inference stages. To this end, we find some grounding TF parameters with unbiased ICL prediction, when the **ICL prompting data** is also confounded, i.e., $\epsilon \not\perp \mathbf{x}^{(i)}$ for $\{\mathbf{x}^{(i)}, y^{(i)}\}_{i=1}^n$:

**Lemma 1** (Unbiased Attention Weights (U_weights)). *Constructing the TF with parameterized by* $\mathbf{S}^u, \mathbf{T}^u$ *in below:*

$$\mathbf{S}^u = \begin{bmatrix} 0_{d \times d} & 0 \\ \mathbf{w}_\star^\top & 0 \end{bmatrix} \quad \mathbf{T}^u = -\frac{1}{2} \begin{bmatrix} \mathbf{I}_{d \times d} & 0 \\ 0 & 0 \end{bmatrix}. \tag{5}$$

*Let* $\hat{y}^{(n+1)}$ *be predicted* $y$ *such that* $\hat{y}^{(n+1)} = -TF_{\mathbf{S}^u, \mathbf{T}^u}^{pred}(\mathbf{M})$. *Then it holds that* $\hat{y}^{(n+1)} = \langle \mathbf{x}^{(n+1)}, \mathbf{w}^u \rangle$ *where the **Meta-weights** $(\mathbf{w}^u)$ is optimized by standard gradient descent w.r.t. the loss* $\mathcal{L}(\mathbf{w}^u) := \frac{1}{2n} \sum_{i=1}^n ((\mathbf{w}^u)^\top \mathbf{x}^{(i)} - \mathbf{w}_\star^\top \mathbf{x}^{(i)})^2$.

We also note that pre-trained weights in the **unconfounded** case (Ahn et al., 2023; Mahankali et al., 2023) will induce **biased inference results in the ICL stage** with confounded testing examples (see Appendix E.) By contrast, Lemma 1 informs that in the case of EICL, the equivalence between forwarding process of ICL and implicit unbiased gradient w.r.t. squared loss on learning $\mathbf{w}_\star$.

## 3.2 EFFECT OF ENDOGENOUS BIAS DURING THE PRE-TRAINING PHASE

Since the "grounding parameters" are constructed in Sec 3.1, we then are capable of characterizing *how parameters of TF pre-trained on endogenous data deviate from such grounding parameters*, i.e., offering the theory of the pre-training stage in our ECIL problem (see Fig. 5).

To be specific, we denote the covariance matrix of the input $X$ as $\Sigma = \mathbf{U} \mathrm{diag}(\lambda_1, \ldots, \lambda_d) \mathbf{U}^\top$, with $\mathbf{w}_\star$ randomly sampled from $\mathcal{N}(0, I_d)$. We use $\mathbf{S}^b, \mathbf{T}^b$ to denote the parameters of TFs **pre-trained on the confounded data**. Following previous protocols (Ahn et al., 2023), we first equivalently reduce the form of parameters from $(\mathbf{S}^b, \mathbf{T}^b)$ to a more sparse version as $\{\overline{\mathbf{S}^b}, \mathbf{T}_{:,j}^b\}_{j=1}^d$ (the same as $(\mathbf{S}^u, \mathbf{T}^u))^2$ (see detailed derivation in Appendix F.2). Subsequently, to prepare for our first main result, some extra regularity assumptions are required:

**Assumption 2.** *We assume that* $\mathbf{X}_j = r_j \epsilon + \kappa_j$, *where* $\epsilon$ *refers to the hidden confounder,* $r_j$ *is the confounding strength,* $\kappa_j$ *refers to the noise term with* $\mathbb{E}[\kappa_j] = 0$ *and* $\mathbb{E}[\kappa_j^2] = 1$.

**Assumption 3** (No interference). *For any* $i_1$, $i_2$, $\epsilon^{(i_1)} \perp\!\!\!\perp \mathbf{x}^{(i_2)}$ *and vice versa.*

Our Assumption 2 **aims to simplify our main results**, while removing it *will not* affect our final conclusion of the bias characterization. Meanwhile, the Assumption 3 is commonly adopted in causal inference area, i.e., the Stable Unit Treatment Value Assumption (Zhang & Wang, 2024; Tchetgen & VanderWeele, 2012). Then we present our first main theorem in the pre-training phase in below:

**Theorem 1** (Deviated Parameters During the Pre-training Phase). *Under Assumptions 2, 3 with in our EICL problem, we derive the following result for the single-layer TF with* $\mathcal{P}_\mathcal{X} = \mathcal{N}(0, \Sigma)$:

$$\Delta_{pre}^j (\overline{\mathbf{S}}, \overline{\mathbf{T}}) = \mathbf{U} \left( \underbrace{r_j}_{Conf.Strength} \mathbf{K} + \mathbf{R} \right) \mathbf{U}^\top, \tag{6}$$

*where* $\Delta_{pre}^j (\overline{\mathbf{S}}, \overline{\mathbf{T}})$ *characterizes the bias of the simplified/reduced TF parameters* $\overline{\mathbf{S}^b}, \overline{\mathbf{T}^b}$ *compared to (unbiased) U_weights* $\overline{\mathbf{S}^u}, \overline{\mathbf{T}^u}$ *(see Sec. 3.1) at the $j$-th feature dimension, the matrices $\mathbf{R}$ and $\mathbf{K}$ consists of some constants w.r.t. the moments of $\epsilon$, the underlying weight $w_\star$ and $\lambda$.*

> Our Theorem 1 informs that: (a) The (biased) TF parameters $\overline{\mathbf{S}^b}, \overline{\mathbf{T}^b}$ deviate from the grounding (induced unbiased ICL predictions) parameters $\overline{\mathbf{S}^u}, \overline{\mathbf{T}^u}$ with the gap proportional to the confounding strength $r_j$ (see proof in Appendix F.3.1).

---

[2] $\overline{\mathbf{S}}$ refers to the last row of $\mathbf{S}^b$, and $\mathbf{T}_{:,j}^b$ $(1 \le j \le d)$ refers to the first $j$-th column of $\mathbf{T}^b$

### 3.3 EFFECT OF ENDOGENOUS BIAS DURING THE ICL PREDICTION PHASE

When characterized the effect of endogenous bias in pre-training phase, it is natural to doubt whether such bias will propagate into the ICL prediction stage. To answer this question, we then derive the endogenous prediction bias through the meta-weight induced by deviated pre-training in Theorem 1 (). More specifically, definition 2 in below informs that our ICL Gradient Divergence equivalently quantifies the estimation bias on $y^{(i)}$:

**Definition 2** (ICL Gradient Divergence). *The $j$-th coordinate on the meta-weight (see Def 1) further decides the prediction bias, as shown in below:*

$$\underbrace{\Delta_{est}^{w}[j] := \mathbf{w}^{u}[j] - \mathbf{w}^{b}[j]}_{\text{Difference in Induced Meta-Weights}} \Rightarrow \underbrace{\Delta y^{(i)} := \sum_{j} \left(\mathbf{w}^{u} - \mathbf{w}^{b}\right)[j]\mathbf{x}^{(i)}[j]}_{\text{Prediction Bias}}. \tag{7}$$

We then present our **second main theorem** by utilizing results from Theorem 1:

**Theorem 2** (Estimation Bias). *With regularity assumptions: (1) Bounded sample-covariance matrix $\mathbf{Z} = \mathbf{m}^{(i)}(\mathbf{m}^{(i)})^{\top}$ as $\alpha_1 \leq \min_{1 \leq l,k \leq d} \mathbf{Z}_{kl} \leq \alpha_2$, where $\mathbf{m}^{(i)} = (\mathbf{x}^{(i)}, y^{(i)})$; (2) Finite second-moments of $\mathbf{Z}$: $\max_{kl} Var[\mathbf{Z}] \leq \kappa_Z$, the following inequality holds with probability in $1 - \sum_l q_l$:*

$$\Delta_{est}^{w}[j] \geq \underbrace{r_j}_{\text{Conf. Strength}} \underbrace{\mathcal{O}_n\left(\sum_l r_l \left(\sum_v \mathbf{w}_\star[v]\right)\sigma^2\right)}_{\text{Constant w.r.t. increasing } n} + \underbrace{\mathcal{O}(\kappa_Z \sum_l \sqrt{\frac{\kappa_Z}{q_l}})}_{\text{Constant w.r.t. } q}. \tag{8}$$

**Remark 2** (New Features of Our Bias Characterization). *Notably, in our Theorem 2, $\Delta_{est}^{w}[j]$ caused by hidden confounders exist is proportional to the confounding strength $r_j$, which shares similar expression as in endogenous OLS regression (Angrist et al., 1996). To be specific, under the typical theory of OLS regression, the bias can be derived via a closed-form estimation of $w_\star$:*

$$\mathbb{E}[\hat{w}] - w^{\star} = \mathbb{E}[XX^{\top}]^{-1}\mathbb{E}[X\epsilon],$$

*and its prediction bias*

$$\mathbb{E}[\hat{Y}(x)] - Y^{\star}(x) = x^{\top}\mathbb{E}[XX^{\top}]^{-1}[r_1\sigma_{X_1}\sigma_\epsilon, \ldots, r_d\sigma_{X_d}\sigma_\epsilon]^{\top},$$

*which depends only on the confounding correlations $r_j$. However, these derivations fundamentally rely on properties that do not hold in transformer-based ICL. By contrast, our bias in Theorem 2 differs qualitatively from OLS: (1) **Bias cancellation via** $\sum_l r_l$. The global bias depends on the sum of confounding strengths across dimensions, enabling cancellation when $\sum_l r_l = 0$. Classical OLS regression does not exhibit this property; (2) **Dependence on attention geometry.** Additional bias terms arise from how confounders interact with attention, a behavior entirely absent in OLS/IV regression.*

## 4 DOUBLY-DEBIASING: PROMPTING WITH UNBIASED DATA COLLECTION

Current ICL-based de-confounding methods are limited to instrumental-variable (IV) approaches (Liang et al., 2024). Moreover, data-fusion de-confounding techniques (Kallus et al., 2018; Li et al., 2024) require fine-tuning transformers on unbiased data, which contradicts the inference-only nature of ICL. Motivated by these gaps, we propose a novel **gradient-free debiasing framework, Doubly-Debiasing (DDbias)**, which operates without any additional fine-tuning, auxiliary labels, or IV construction.

> Collecting **a small number of unbiased** prompting data (i.e., samples with independence $\epsilon \perp\!\!\!\perp \mathbf{x}$) as $\mathcal{D}^u = \{\mathbf{x}_{rc}^{(i)}, y_{rc}^{(i)}\}_{i=1}^{n_b+1}$, without **hidden confounders existing** between $y_{rc}^{(i)}$ and $\mathbf{x}_{rc}^{(i)}$;
>
> (1) We first prompt the pre-trained TF with $\mathcal{D}^u = \{\mathbf{x}_{rc}^{(i)}, y_{rc}^{(i)}\}_{i=1}^{n_b}$ with predicted (biased) outcome as $\hat{y}_b^{(i)}$;
>
> (2) We then prompt the TF again with residual prompting $\{\mathbf{x}_{rc}^{(i)}, y_{rc}^{(i)} - \hat{y}_b^{(i)}\}_{i=1}^{n_b}$, and the prediction for $y^{(n+1)}$ will be unbiased.

We first prove that our DDbias method implicitly performs gradient descent w.r.t. learning the residual term $y^{(i)} - \hat{y}_b^{(i)}$ using a squared loss in below:

**Theorem 3** (Implicit Implementation of Debiasing Algorithm). *Consider the TF with parameterized by* **biased** $\mathbf{S}^b, \mathbf{T}^b$, *and let* $\hat{y}^{(n+1)}$ *be predicted* $y$ *such that* $\hat{y}^{(n+1)} = TF_{\mathbf{S}^b, \mathbf{T}^b}^{pred}(\mathbf{M})$. *Then it holds that* $\hat{y}^{(n+1)} = \langle \mathbf{x}_{rc}^{(n+1)}, \mathbf{w}_l^{DEB} \rangle$ *where the* **Meta-weights** $\{\mathbf{w}_l^{DEB}\}$ *is defined as* $\mathbf{w}_0^{DEB}$ *equaling to some constant and as follows for* $l \geq 0$:

$$\mathbf{w}_{l+1}^{DEB} = \mathbf{w}_l^{DEB} - \eta \nabla \mathcal{L}(\mathbf{w}^{DEB}) \ \ where \ \ \mathcal{L}^{deb}(\mathbf{w}) := \frac{1}{2n} \sum_{i=1}^{n} (y^{(i)} - \hat{y}_b^{(i)} - \mathbf{w}^\top \mathbf{x}_{rc}^{(i)})^2, \quad (9)$$

*where* $\hat{y}_b^{(i)}$ *refers to the biased prediction from pre-trained TF equipped with* $\mathbf{S}^b, \mathbf{T}^b$.

We then prove that optimizing equation 9 in Theorem 3 yields unbiased ICL prediction over $y^{(n+1)}$:

**Proposition 1.** *Assuming that: (1) TF equipped with* $\overline{S}^b$ *and* $\overline{T}^b$ *is a consistent estimator on confounded pre-trainined data; (2)* $X, Y, Y_b$ *have finite fourth moments over the unbiased prompting data, optimizing over* $\mathcal{L}^{deb}(\mathbf{w}) := \frac{1}{2n} \sum_{i=1}^{n} (y^{(i)} - \hat{y}_b^{(i)} - \mathbf{w}^\top \mathbf{x}_{rc}^{(i)})^2$ *yields unbiased ICL prediction.*

> Theorem 3 and Proposition 1 informs that with our proposed DDbias, TFs pre-trained on biased data can achieve unbiased ICL predictions without requiring tuning the model parameters or constructing IVs as in (Liang et al., 2024) (see proof in Appendix F.3.3).

## 4.1 ROBUSTNESS OF DDBIAS ON PARTIALLY CONFOUNDED DATA

In real-world in-context learning (ICL) setups, the residual correction set is rarely perfectly unbiased. Confounding may occur due to imperfect interventions, correlated noise, or heterogeneous data sources, resulting in non-zero feature–noise correlation $\mathbb{E}[x_j \epsilon] \neq 0$. We consider two representative forms of such bias: **(1) Weakly confounded samples:** All samples exhibit mild but bounded confounding effects, i.e., $|\mathbb{E}[x_j \epsilon]| \leq \delta$ with a small $\delta$, reflecting a globally weak bias, where each example slightly deviates from the ideal unconfounded assumption; **(2) Partially confounded samples:** Only a fraction $\rho$ of samples are contaminated while the remaining $(1 - \rho)$ are unbiased. This case captures the realistic situation where most context examples are clean, but a small portion introduces systematic bias. Analyzing these two cases is critical for understanding our DDebias:

**Proposition 2** (Weakly Confounded ICL Samples). *Let the confounded ICL samples* $X^{(conf)}$ *contain* $n_b$ *unbiased samples with*

$$\lambda_{\min}\Big( \tfrac{1}{n_b} X^{(conf)}(X^{(conf)})^\top \Big) \geq \lambda_* > 0,$$

*and let the corresponding noise* $\varepsilon$ *satisfy* $\mathbb{E}[x_i \varepsilon_i] = 0$ *and* $\mathbb{E}[\varepsilon_i^2] \leq \sigma^2$. *Then, for the DDbias estimator* $\widehat{\theta}$, *there exists a constant* $C > 0$ *such that*

$$\|\widehat{\theta} - \theta\|_2 \leq C\Big( \tfrac{1}{\sqrt{n_b}\,\lambda_*} \Big), \qquad |\mathbb{E}[y_{GT} - \widehat{y}_{DEB}]| \leq C'\Big( \tfrac{1}{\sqrt{n_b}\,\lambda_*} \Big).$$

**Proposition 3** (Mixed ICL Samples). *Assume a fraction* $\rho$ *of the "unconfounded" batch is contaminated: for those contaminated samples* $\mathbb{E}[x_j \epsilon] = r_j \neq 0$ *(denoted by* $X^{(\mathrm{cont})}$*), while the remaining* $(1 - \rho)n_b$ *samples are unbiased (denoted by* $X^{(\mathrm{clean})}$*). Suppose the clean subset is well-conditioned:* $\lambda_{\min}\Big( \frac{1}{(1-\rho)n_b} X^{(\mathrm{clean})} X^{(\mathrm{clean})\top} \Big) \geq \lambda_* > 0$. *Let the mean confounding strength be* $\bar{r} := \frac{1}{\rho n_b} \sum_{i \in \mathrm{cont}} |r_i|$. *Then there exist constants* $C' > 0$ *such that*

$$\Big| \mathbb{E}[y_{GT} - \widehat{y}_{DEB}] \Big| \leq C'\Big( \frac{1}{\sqrt{(1-\rho)\,n_b\,\lambda_*}} + \rho\,\bar{r} \Big).$$

*Hence, the bias remains asymptotically negligible as* $n_b \to \infty$ *or* $\rho \to 0$, *and a sufficient condition for asymptotic unbiasedness is*

$$\rho\,\bar{r} = o(1), \quad and \quad \frac{1}{\sqrt{(1-\rho)n_b \lambda_*}} = o(1).$$

Table 2: Brief comparison between IV (proxy-based) and DDbias (unbiased-sample correction) in ICL settings.

|  | Proxy variable | Unbiased samples | Structural Assumptions | Suitable scenes | Failure mode |
|---|---|---|---|---|---|
| IV (Proxy) | Valid instrument(s). | Not required. | Additive Formula. | Genetic instruments, policy shocks. | Invalid or weak instruments. |
| DDbias (ours) | Not required. | Few unconfounded examples. | Not required. | Randomized traffic / recommender A/B. | Partially confounded samples. |

Table 3: Experimental results of DDbias on the Yelp Sentiment-Review Prediction tasks.

| Dataset | Vanilla LLaMA | DMCEE Cheng et al. (2022b) | SC Cheng et al. (2022c) | DDbias (15 ) | DDbias (30 ICL) | DDbias (60 ICL) |
|---|---|---|---|---|---|---|
| RPI | 23.8 | 21.2 | 22.5 | **22.8** | **19.4** | **16.7** |
| ROR | 0.46 | 0.28 | 0.24 | **0.36** | **0.18** | **0.16** |

## 5 EXPERIMENTS

**Linear Data Generation.** We follow the function regression setting in (Garg et al., 2022; Ahn et al., 2023; Von Oswald et al., 2023) by considering the ICL loss for linear regression. Our simulation protocols contains two parts, i.e., unconfounded $x, y$ and confounded $x, y$ respectively: (1) **Unconfounded Case.** We simulate $x^{(i)} \sim \mathcal{N}(0, \Sigma)$ and $w_\star \sim \mathcal{N}(0, \Sigma^{-1})$, where $d = 5$, $n_{tr} = 10^7$, $n_{te} = 20$ (context length), $\Sigma = U^\top D U$, $U$ is a uniformly random orthogonal matrix, and $D$ is a diagonal matrix with entries $(1, 1, 0.25, 0.0625, 1)$. Meanwhile, we generate the exogenous $\epsilon \sim N(0, 0.5)$, and then simulate $y^{(i)} = \langle w_\star, \mathbf{x}^{(i)} \rangle + \epsilon^{(i)}$; (2) **Confounded Case.** We first generate the exogenous variable $\epsilon \sim N(0, 0.5)$, and then sample the linear $x - \epsilon$ relationship, i.e., the confounding effect $r_j \sim U(0, 1)$ (uniform). Then the features are simulated as $\mathbf{x}_j^{(i)} = r_j \epsilon + \kappa_j$ with $\kappa_j \sim N(0, 1)$. Finally, $y^{(i)}$ is generated by an analog. By controlling $|r_j|$, the strength of confounding effect can be tuned, i.e., the larger $r_j$ corresponds to larger effect. To check our Theorem 1 and 2 that the deviation and bias increasing w.r.t. increasing $r_j$, we tune the value of $r_j$ by multiplying factos in $[0.5, 1.0, 1.5, 2.0]$, with corresponding data denoted as Conf_$r_j$, e.g., Conf_2.0.

**IV-oriented Data Generation.** To compare DDbias with an IV-based approach, we extend our original confounded DGP by introducing instruments $Z$. We first draw $Z^{(i)} \sim \mathcal{N}(0, I_m)$, a latent confounder $U^{(i)} \sim \mathcal{N}(0, \sigma_u^2)$, and noise $\epsilon^{(i)} \sim \mathcal{N}(0, \sigma_\epsilon^2)$. Features are generated by combining instrument relevance and confounding: $x_j^{(i)} = (\Gamma z^{(i)})_j + \alpha_j U^{(i)} + \kappa_j^{(i)}$, where $\Gamma$ controls instrument strength and $\alpha_j$ matches the heterogeneous confounding coefficients $r_j$. Outcomes follow $y^{(i)} = x^{(i)\top} w_\star + \beta U^{(i)} + \epsilon^{(i)}$, with $w_\star \sim \mathcal{N}(0, \Sigma^{-1})$. By construction, $Z \perp (U, \epsilon)$ (instrument exogeneity) and $Z \to X$ via $\Gamma$ (relevance), yielding a clean comparison: IV exploits large confounded observational data with valid instruments, whereas DDbias requires only a small unconfounded batch.

**Non-linear and Partially Confounded Data Generation.** We further test the robustness of theoretical conclusions on non-linear models or partially confounded ICL samples. We leave the protocols of non-linear DGP in Appendix G.7 and G.8.

**Real-world NLP Datasets.** Finally, we conduct experiments on larger models pre-trained on two NLP datasets, i.e., two sentiment review tasks, collected from the Yelp website (Cheng et al., 2022b;c). We leave detailed setup of compared baselines with in Appendix G.6. We refer to the adopted two datasets as (1) Restaurant Popularity Index (RPI) and (2) Restaurant Overall Rating (ROR).

**Implementation of Transformers.** Throughout our experiments, we set the number of layer of the TF as 3 (TF@3) and 1 (TF@1), respectively, with each weight is initialized as i.i.d. Gaussian matrices. The optimizer is set to the Adam optimizer (Kingma, 2014) w.r.t. the ICL loss in equation 4, with each gradient step computed from a minibatch of size 20,000, each minibatch resampled every 100 steps, and gradients clipped to $0.01$. All results are averaged over five runs, with a different $U$ (and thus $\Sigma$) sampled for each run.

**Justifying the U_Weights.** Following (Von Oswald et al., 2023), we design three metrics to test whether the constructed U_weights in Lemma 1 match GD on $\mathcal{L}'(\mathbf{w}) = \frac{1}{2n} \sum_i (\mathbf{w}^\top \mathbf{x}^{(i)} - w_\star^\top \mathbf{x}^{(i)})^2$. Let $\theta^u$ denote our constructed TF parameters and $\theta^*$ those optimized on $\mathcal{L}'$ (Appendix G.2). We compare: (a) prediction divergence $\|\hat{y}^{\text{te}}(\theta^u) - \hat{y}^{\text{te}}(\theta^*)\|$; (b) L2 sensitivity divergence; (c) cosine sensitivity divergence, averaged over 500 trials. Fig. 1(a–b) confirms that U_weights closely track implicit GD on $\mathcal{L}'$.

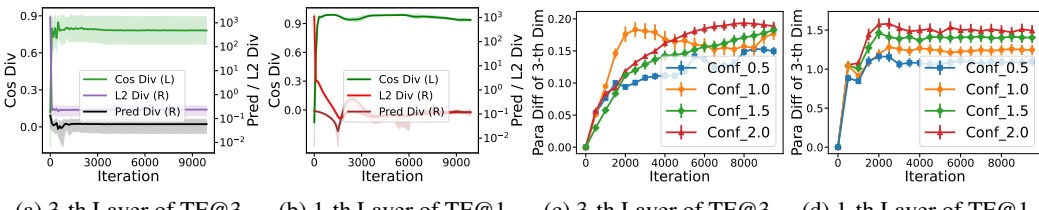

(a) 3-th Layer of TF@3.   (b) 1-th Layer of TF@1.   (c) 3-th Layer of TF@3   (d) 1-th Layer of TF@1

Figure 1: (a-b): Verification of Lemma 1; (c-d): Deviation of Pretrained Weights in Theorem 1.

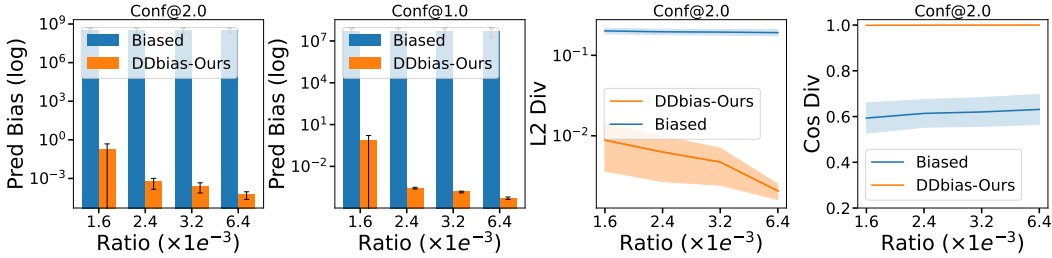

Figure 2: The left two figures unveil the prediction bias comparision between our DDbias method and vanilla pre-trained biased model. The right two figures show the L2 Div and Cosine similarity with grounding $\mathbf{w}_*$ of the regressed $\hat{\mathbf{w}}$ before (red) and after debiasing (blue).

Table 4: $f$ = Ploy, Prediction error of deeper, non-linear Transformers (ReLU, L=3 / 5 / 7, LayerNorm, Non-linear MLPs, Softmax, Ratio=$1.61e^{-3}$)) under different confounding strengths $r$.

| ICL Samples / L | Conf@0.5 | Conf@1.0 | Conf@1.5 | Conf@2.0 |
|---|---|---|---|---|
| L=5 — Biased (Vanilla TF) | 0.280 | 0.370 | 0.475 | 0.600 |
| L=5 — DDbias (Ours) | 0.115 | 0.150 | 0.200 | 0.260 |
| L=7 — Biased (Vanilla TF) | 0.320 | 0.450 | 0.600 | 0.750 |
| L=7 — DDbias (Ours) | 0.135 | 0.175 | 0.225 | 0.290 |

Table 5: Effect on the Prompting Sample Ratio (Weak Conf@0.10)

| ICL prediction error | 0.090 | 0.092 | 0.095 | 0.100 |
|---|---|---|---|---|
| Sample ratio for DDbias (oracle) ($\times10^{-3}$) | 6.4 | 3.2 | 2.4 | 1.6 |
| Sample ratio for DDbias (weak) ($\times10^{-3}$) | 8.6 | 7.2 | 6.3 | 3.5 |

**Deviation of the Pre-training.** We further measure the deviation $\Delta_{pre}^j(\overline{\mathbf{S}}, \overline{\mathbf{T}})$ between confounded pre-trained $(S^b, T^b)$ and U_weights $(S^u, T^u)$ across feature dimensions. Fig. 1(c–d) shows that pre-trained weights diverge from U_weights (Theorem 1), and that larger $r_j$ leads to larger deviation, matching our theoretical prediction. Additional visualizations appear in Appendix Figs. 10–13.

**Estimation Bias during ICL.** We compare ICL estimation using unconfounded $(S^u, T^u)$ and confounded $(S^b, T^b)$ weights by regressing their predictions on confounded $\mathbf{x}$ via OLS (Zdaniuk, 2024). L2 and cosine divergences between $w_\star$ and $\hat{\mathbf{w}}$ are computed. As shown in Fig. 6, $\hat{\mathbf{w}}^u$ aligns with $w_\star$, whereas $\hat{\mathbf{w}}^b$ diverges as $r_j$ increases, validating Theorem 2. Additional comparisons under confounded vs. unconfounded data are provided in Appendix Figs. 7–16.

**Verifying Our DDbias Method.** Fig. 2 demonstrates the effectiveness of our proposed DDbias method, highlighting two advantages: (1) as the prompt/pre-train sample ratio increases, prediction bias decreases and the OLS-regressed $\hat{\mathbf{w}}$ aligns with the ground truth (Appendix G.5); (2) only very few unbiased samples suffice to debias, showing efficiency. Additional experiments (Appendix Figs. 18–20) and tests under non-Gaussian data (Appendix G.10) further confirm robustness.

**Results on Real-world Datasets.** Table 3 reports results on RPI and ROR, whose MAE scales differ due to distinct outcome ranges (RPI: foot-traffic popularity $\sim$0–100; ROR: 1–5 star ratings). DDbias consistently reduces MAE as the number of ICL examples grows ($15 \rightarrow 30 \rightarrow 60$), eventually outperforms strong causal baselines. These results show that DDbias mitigates hidden-confounder effects.

**Generalized Analysis on Non-linear, deeper Models.** We conduct extensive studies to prove that our bias characterization analysis can be generalized into non-linear regime, including the non-linear TFs with non-linear DGP in Tab. 4 (see more detailed results in Appendix G.7).

**Robustness Analysis on partially confounded ICL samples.** Finally, as shown in Tab. 5, we perform robustness to validate our Proposition 2 and 3, where the prompted ICL samples are either weakly confounded or a mixture of unbiased/biased examples. Results further support our theoretical robustness analysis that *our DDbias remains robust even with imperfect prompting samples.* We leave detailed experimental results with analysis in Appendix G.8.

**Comparison with IV-based ICL Method.** We evaluate DDbias against IV-based in-context debiasing (Liang et al., 2024) under three controlled setups: (1) both unbiased ICL samples and valid IVs can be queried (see details in Tab. 26 in Appendix); (2) only valid IVs can be queried and partially confounded ICL samples are accessible (see details in Tab. 27 in Appendix); and (3) only unbiased ICL samples are available but confounded IVs exist (see Tab. 28 in Appendix).

## 6 RELATED WORK

**Intrinsic Mechanisms inside ICL.** Recent works seek to explain ICL through implicit Bayesian inference (Xie et al., 2021; Raventós et al., 2024; Wang et al., 2024), pre-training data properties (Chan et al., 2022), and implicit gradient descent (GD) (Ahn et al., 2023; Akyürek et al., 2022). In function extrapolation, Akyürek et al. (2022) first analyzed ICL dynamics of a pre-trained linear TF, later extended by Von Oswald et al. (2023); Cheng et al. (2023) who proved equivalence between ICL and implicit one-step GD under tailored weights. Further, Ahn et al. (2023); Mahankali et al. (2023) linked pre-training dynamics to converged weights inducing implicit GD. In contrast, Liang et al. (2024) highlight that prior studies neglect hidden confounders, showing TFs realize IV regression for bias correction. Yet, theory for endogeneity in ICL remains undeveloped.

**Exploring Causality in Prompting.** A bunch of causally-inspired prompting methods emerge to enhance the causal inference performance of LLMs from diverse perspectives (Chi et al.; Chen et al., 2024a;b; Liang et al., 2023; 2025), including the Causal Prompt (Zhang et al., 2024a), Intervented Prompt (Tan, 2023), Meta-CausalPrompt (Ohtani et al., 2024), and Casual Chain-of-thought (Jin et al., 2023). In recent, Liang et al. (2024) reveals that the TFs are highly-efficient IV estimator during ICL process. Unfortunately, the above-mentioned work is lack of theoretical understanding and insights whether and how hidden confounders affects the ICL prediction.

**Empowering Transformers for Causal Tasks.** The zero-shot or the few-shot properties of TFs have inspired diverse causal tasks, including the zero-shot treatment effect estimation (Zhang et al., 2023; Mahajan et al., 2024; Nichani et al., 2024), and causal discovery (Nichani et al., 2024). Besides, the LLMs are also directly prompted for complex causal tasks (Liu et al., 2024; Jiang et al.).

## 7 CONCLUSION

In this paper, we propose a pioneering analysis reveals the presence of prediction bias proportional to the strength of the confounding effect, highlighting the importance of addressing hidden confounders in data preparation. Alternatively, our proposed DDbias method demonstrates that unbiased predictions can be achieved using a small number of unconfounded prompting examples. Future work includes: (a) extending the framework to nonlinear settings; and (b) exploring the generalization capabilities of ICL, such as out-of-distribution generalization (Liu et al., 2021), by investigating shifts in $\epsilon$ across training and testing regimes, providing broader insights for the ML community.

ACKNOWLEDGEMENT

This work was jointly supported by the National Natural Science Foundation of China (Nos. 62276004, 62325604, 62525213 and 62372459), the Beijing Natural Science Foundation (No. L257007), the Beijing Major Science and Technology Project (No. Z251100008425006) and the NUDT Youth Independent Innovation Science Fund (No. ZK25-20).

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

## Appendix

We provide supplementary documents to support our research. The details of Large Language Model usage are presented in Section A. We provide a concrete case to illustrate the commonly existence of hidden confounders for ICL prediction in Section C, with detailed notations in Section D. Moreover, we first state the failure of previous analysis in constructing unbiased meta-weights in Section E, and offer the proof details in Section F. Finally, we put experimental details, including the baseline setups, data preparations, with more experimental results in Section G.

## A  Large Language Model Usage

In this paper, we clarify that large language models (LLMs) are employed solely to support and refine the writing process. Specifically, we use LLMs to provide sentence-level suggestions and to enhance the overall fluency of the text.

## B  Framework Illustration of Our Method

## C  Case of Hidden Confounders in MSOR Example

This case study illustrates how the consideration of a hidden confounder (HC) can alter the relationship between sentiment aspects and outcomes like Popularity and Rating. For Popularity, the inclusion of HC led to significant reversals in the effects of both Price Neg and Misc Pos, changing negative associations to positive ones. For Rating, while some effects remained unchanged, the influence of Misc Neg was neutralized, highlighting the critical role of the confounder in adjusting the analysis of sentiment's impact on these two metrics (Cheng et al., 2022a). For readers, we refer further details in (Cheng et al., 2022a).

## D  Notation Table

We summarize various notations used in our problem formulation, method design and algorithm in Tab. 8.

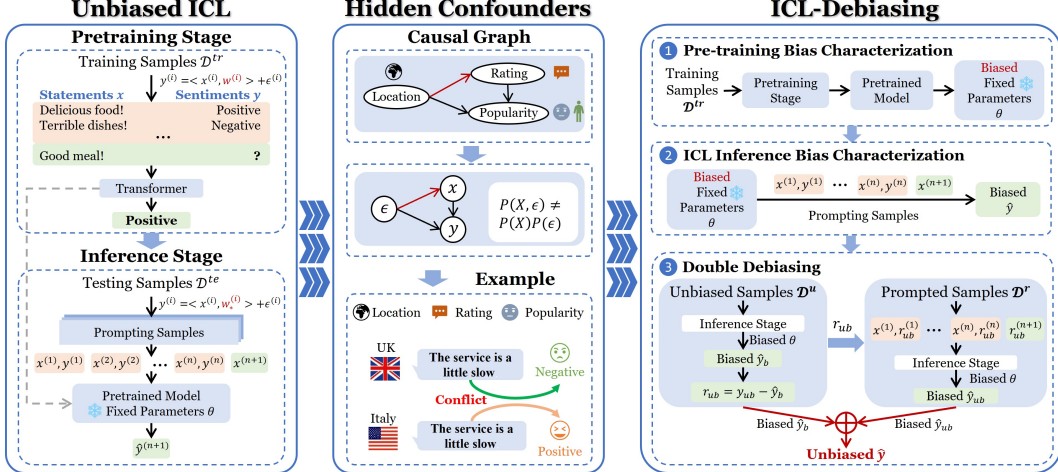

Figure 3: The illustration of (1) Traditional (unbiased) ICL framework; (2) Our proposed EICL problem; (3) The bias characterization contributed in this paper with the proposed DDbias method. As shown in the Figure, the same review 'The service is a little slow' may be perceived as relaxed and positive in Italy, but inefficient and negative in the UK.

Table 6: The table compiles all data related to hidden confounders extracted from (Cheng et al., 2022a). The table illustrates the changes in the correlation coefficients of the effects that MAS has on popularity and rating before and after accounting for hidden confounders. Nearly every sentiment aspect's influence coefficient has changed. However, for several sentiment aspects, the coefficients have reversed signs dramatically, indicating a pronounced effect from hidden confounders, which is shown in Table 7 in below.

| Sentiment Aspect | Popularity | | Rating | |
|---|---|---|---|---|
| | With HC | Without HC | With HC | Without HC |
| Ambience Neg | -0.24 | -0.22 | -0.02 | -0.01 |
| Food Pos | 0.39 | 0.34 | 0.25 | 0.23 |
| Food Neg | 0.09 | 0.05 | -0.06 | -0.08 |
| Price Pos | -0.04 | -0.03 | 0.06 | 0.03 |
| Price Neg | 0.03 | -0.00 | -0.05 | -0.03 |
| Service Pos | 0.10 | 0.03 | 0.22 | 0.20 |
| Service Neg | 0.08 | 0.09 | -0.37 | -0.31 |
| Misc Pos | 0.03 | -0.04 | 0.05 | 0.03 |
| Misc Neg | -0.03 | -0.08 | -0.03 | 0.00 |

Table 7: The table presents the mean values for Popularity and Rating across different Sentiment Aspects, with and without the consideration of a **hidden confounder** (HC). The primary observation from this table is the notable reversal in the influence of certain sentiment aspects on **Popularity** and **Rating** after accounting for the **hidden confounder**.

| | Popularity | | Rating | |
|---|---|---|---|---|
| Sentiment Aspect | With HC | Without HC | With HC | Without HC |
| Price Neg | **0.03** | **-0.00** | -0.05 | -0.03 |
| Misc Pos | **0.03** | **-0.04** | 0.05 | 0.03 |
| Misc Neg | -0.03 | -0.08 | **-0.03** | **0.00** |

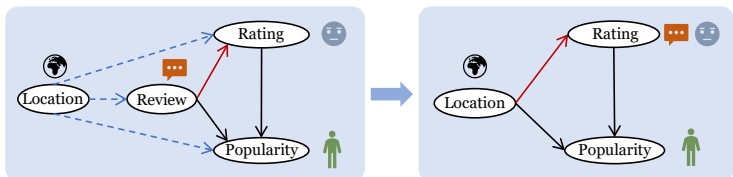

Figure 4: Illustrative Causal Graph of Our Running Example on MSOR.

Table 8: Summary of notions with their definitions.

| Notation | Definition |
|---|---|
| $\mathbf{x}^{(i)}$ | Features of the $i$-th Sample |
| $y^{(i)}$ | Label of the $i$-th Sample |
| $\epsilon^{(i)}$ | Noise of the $i$-th Sample |
| $r_j = \mathbb{E}[X_j \epsilon]$ | The strength of confounding between $j$-th coordinate of $X$ and $\epsilon$. |
| $\kappa_j$ | The exogenous noise of $X$ w.r.t. $\epsilon$. |
| $d$ | Dimension of the features. |
| $\mathcal{D}^{tr}$ and $\mathcal{D}^{te}$ | Training and Testing Datsets. |
| $\mathcal{D}^{tr}$ and $\mathcal{D}^{te}$ | Training and Testing Datsets. |
| $\mathbf{M} = \begin{bmatrix} \mathbf{x}^{(1)} & \mathbf{x}^{(2)} & \cdots & \mathbf{x}^{(n)} & \mathbf{x}^{(n+1)} \\ y^{(1)} & y^{(2)} & \cdots & y^{(n)} & 0 \end{bmatrix}$ | Input format of data when feeding the transformers. |
| $\mathbf{m}^{(i)} = (\mathbf{x}^{(i)}, y^{(i)})$ | Input Pairs. |
| $P_{\mathcal{X}} = \mathcal{N}(0, \Sigma)$ | Distributions of the input features $X$. |
| $\Sigma = \mathbf{U} \mathrm{diag}(\lambda_1, \ldots, \lambda_d) \mathbf{U}^{\top}$ | The covariance matrix of input feature $X$ |
| $\mathbf{w}_{\star}$ | The underlying parameter between $\mathbf{x}^{(i)}$ and $y^{(i)}$. |
| $\mathbf{W_v}, \mathbf{W_k}, \mathbf{W_q} \in \mathbb{R}^{(d+1) \times (d+1)}$ | The projection parameters of transformers when computing the self-attentions. |
| $\mathbf{C} = \begin{bmatrix} I_n & 0 \\ 0 & 0 \end{bmatrix} \in \mathbb{R}^{(n+1) \times (n+1)}$ | The mask matrix to reflect the asymmetry in ICL process. |
| $\mathbf{S} := W_v \in \mathbb{R}^{(d+1) \times (d+1)}$ and $\mathbf{T} := W_k{}^{\top} W_q \in \mathbb{R}^{(d+1) \times (d+1)}$ | The aggregated parameters of TF from $\mathbf{W_v}, \mathbf{W_k}, \mathbf{W_q}$. |
| $\mathbf{S}^u, \mathbf{T}^u$ | The U_weights, i.e., the constructed weights of TFs inducing unbiased predictions. |
| $\mathbf{w}^u$ | Corresponding unbiased $X - Y$ relationship updated by implicit GD steps. |
| $\mathbf{S}^b, \mathbf{T}^b$ | The biased weights, i.e., the converged weights of TFs on confounded training data. |
| $\mathbf{w}^b$ | Corresponding biased $X - Y$ relationship updated by implicit GD steps. |
| $\overline{\mathbf{S}} \in \mathcal{R}^{1 \times (d+1)}$ and $\overline{\mathbf{T}} \in \mathcal{R}^{(d+1) \times d}$ | The last row of $\mathbf{S}$ and the first $d$ columns of $\mathbf{T}$. |
| $\overline{\mathbf{T}}_{:,j} \in \mathcal{R}^{d+1}$ | The $j$-th column of $\overline{\mathbf{T}}$ |
| $\mathbf{G}_{st}^j = \overline{\mathbf{S}}^{\top} \overline{\mathbf{T}_{:,j}}^{\top}$ | The composition of $\overline{\mathbf{T}_{:,j}}$ and $\overline{\mathbf{S}}$. |
| $\Delta_{pre}^j (\overline{\mathbf{S}}, \overline{\mathbf{T}})$ | Derivative Divergence in Def 3. |
| $\mathbf{K}_1, \mathbf{K}_2, K_3$ | Constant Matrices/scalars in Theorem 1. |
| $\Delta_{est}^w[j]$ | Prediction bias of $y^{(i)}$ in Theorem 2. |
| $\mathcal{D}^u, \mathbf{x}_{rc}^{(i)}, y_{rc}^{(i)}$ | Unbiased (unconfounded) Dataset. |
| $\mathbf{w}^{DEB}$ | Induced implicit gradient by our proposed method. |
| $\kappa, \alpha$ | Regularity Constants in Theorem 2. |

# E FAILURE OF PREVIOUS CONSTRUCTED WEIGHTS

**Lemma** [**Biased Pre-trained Attention Weights**] *Attention weights in previous studies (Ahn et al., 2023; Akyürek et al., 2022; Von Oswald et al., 2023) suffers from biased (confounded) prompting*

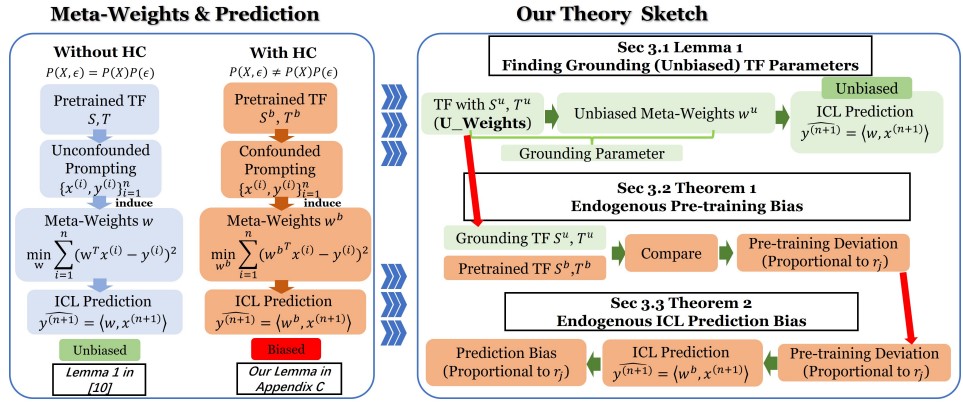

Figure 5: Outline of Our Theory Sketch

examples $\mathcal{D}^{te}$:

$$\mathbf{S} = \begin{bmatrix} 0_{d \times d} & 0 \\ 0 & 1 \end{bmatrix}$$

$$\mathbf{T} = - \begin{bmatrix} \mathbf{U}\text{diag}\left( \left\{ \frac{1}{\frac{n}{n-1}\lambda_i + \frac{1}{n-1}\cdot(\sum_k \lambda_k)} \mathbf{U}^\top \right\}_{i=1}^d \right) & 1 \\ 0 & 0 \end{bmatrix}, \tag{10}$$

*suffers from biased ICL inference, and let $\hat{y}^{(n+1)}$ be predicted $y$ such that $\hat{y}^{(n+1)} = TF_{\mathbf{S},\mathbf{T}}^{pred}(\mathbf{M})$. Then it holds that $\hat{y}^{(n+1)} = \langle \mathbf{x}^{(n+1)}, \mathbf{w}^{GD} \rangle$ where $\{\mathbf{w}_\ell^{GD}\}$ is defined as $\mathbf{w}_0^{GD} = \mathbf{0}$ with either exogenous or endogenous $\epsilon$:*

$$\mathbf{w}_1^{GD} = \mathbf{w}_0^{GD} - \nabla \mathcal{L}(\mathbf{w}^{GD}) \tag{11}$$

$$\text{where} \quad \mathcal{L}(\mathbf{w}) := \frac{1}{2n} \sum_{i=1}^n (\mathbf{w}^\top \mathbf{x}^{(i)} - y^{(i)})^2. \tag{12}$$

*Proof.* We only consider the special case that $U = I$ with $\lambda_1 = \lambda_2 = \cdots = \lambda_d$. Then our derivation follows the same protocols as in (Von Oswald et al., 2023):

$$\hat{y}^{(n+1)} = \mathbf{TF}_{\mathbf{S},\mathbf{T}}^{pred}(\mathbf{M})$$

$$= \mathbf{M} + \frac{1}{n} \mathbf{SMC}(\mathbf{M}^\top \mathbf{TM})$$

$$= \begin{pmatrix} \mathbf{x}^{(n+1)} \\ 0 \end{pmatrix} + \frac{1}{2n} \sum_{i=1}^n \left( \begin{bmatrix} \mathbf{0}_{d \times d} & \mathbf{0} \\ \mathbf{0} & -1 \end{bmatrix} \begin{pmatrix} \mathbf{x}^{(i)} \\ y^{(i)} \end{pmatrix} \right) \otimes \left( \begin{bmatrix} \mathbf{I}_{d \times d} & \mathbf{0} \\ \mathbf{0} & 0 \end{bmatrix} \begin{pmatrix} \mathbf{x}^{(i)} \\ y^{(i)} \end{pmatrix} \right) \begin{bmatrix} \mathbf{I}_{d \times d} & \mathbf{0} \\ 0 & 0 \end{bmatrix} \begin{pmatrix} \mathbf{x}^{(n+1)} \\ y^{(n+1)} \end{pmatrix}$$

$$= \begin{pmatrix} \mathbf{x}^{(n+1)} \\ 0 \end{pmatrix} + \begin{pmatrix} 0 \\ -\nabla \mathcal{L}(\mathbf{w}^{GD})\mathbf{x}^{(n+1)} \end{pmatrix},$$

$$\tag{13}$$

where the optimized loss function is biased, as the $y^{(i)}$ is confounded by $\epsilon^{(i)}$. $\qquad\square$

# F THEORETICAL ANALYSIS

In the following, we perform all theoretical analysis in the case of $P_{\mathcal{X}} = \mathcal{N}(0, \text{diag}(\lambda_1, \ldots, \lambda_d))$, and then turn back to the correlated case that $P_{\mathcal{X}} = \mathcal{N}(0, \Sigma)$.

## F.1 U_WEIGHTS CONSTRUCTION

**Lemma 4 (Construction of Unbiased Implicit Gradient).** *Consider the TF with parameterized by $\mathbf{S}^u, \mathbf{T}^u$ in below:*

$$\mathbf{S}^u = \begin{bmatrix} 0_{d \times d} & 0 \\ \mathbf{w}_\star^\top & 0 \end{bmatrix}$$

$$\mathbf{T}^u = -\frac{1}{2} \begin{bmatrix} \mathbf{I}_{d \times d} & 0 \\ 0 & 0 \end{bmatrix}. \tag{14}$$

*, and let $\hat{y}^{(n+1)}$ be predicted $y$ such that $\hat{y}^{(n+1)} = TF_{\mathbf{S},\mathbf{T}}^{pred}(\mathbf{M})$. Then it holds that $\hat{y}^{(n+1)} = \langle \mathbf{x}^{(n+1)}, \mathbf{w}^{GD} \rangle$ where $\{\mathbf{w}_\ell^{GD}\}$ is defined as $\mathbf{w}_0^{GD} = \mathbf{0}$ **with biased (confounded) prompting examples** $\mathcal{D}^{te}$:*

$$\mathbf{w}_1^{GD} = \mathbf{w}_0^{GD} - \nabla \mathcal{L}(\mathbf{w}^{GD}) \tag{15}$$

$$\text{where} \quad \mathcal{L}(\mathbf{w}) := \frac{1}{2n} \sum_{i=1}^n (\mathbf{w}^\top \mathbf{x}^{(i)} - \mathbf{w}_\star^\top \mathbf{x}^{(i)})^2. \tag{16}$$

*Proof.* The extension follows the same protocols as in (Von Oswald et al., 2023):

$$
\begin{aligned}
\hat{y}^{(n+1)} &= \text{TF}_{\mathbf{S},\mathbf{T}}^{pred}(\mathbf{M}) \\
&= \mathbf{M} + \frac{1}{n}\mathbf{SMC}(\mathbf{M}^\top \mathbf{TM}) \\
&= \begin{pmatrix} \mathbf{x}^{(n+1)} \\ 0 \end{pmatrix} + \frac{1}{2n}\sum_{i=1}^{n}\left(\begin{bmatrix} \mathbf{0}_{d\times d} & \mathbf{0} \\ \mathbf{w}_\star^\top & 0 \end{bmatrix}\begin{pmatrix}\mathbf{x}^{(i)}\\ y^{(i)}\end{pmatrix}\right) \otimes \left(\begin{bmatrix} \mathbf{I}_{d\times d} & \mathbf{0} \\ \mathbf{0} & 0 \end{bmatrix}\begin{pmatrix}\mathbf{x}^{(i)}\\ y^{(i)}\end{pmatrix}\right)\begin{bmatrix} \mathbf{I}_{d\times d} & \mathbf{0} \\ 0 & 0 \end{bmatrix}\begin{pmatrix}\mathbf{x}^{(n+1)}\\ y^{(n+1)}\end{pmatrix} \\
&= \begin{pmatrix} \mathbf{x}^{(n+1)} \\ 0 \end{pmatrix} + \begin{pmatrix} 0 \\ -\nabla\mathcal{L}(\mathbf{w}^{GD})\mathbf{x}^{(n+1)} \end{pmatrix},
\end{aligned}
\tag{17}
$$

where the underlying label is $\mathbf{w}_\star^\top \mathbf{x}^{(i)}$, serving as unbiased guide for optimization during ICL inference. $\square$

## F.2 SIMPLIFICATION ON TRANSFORMER PARAMETERS

### F.2.1 DETAILS ON PARAMETER SIMPLIFICATION

To ease the theoretical analysis, we introduce the Gaussian assumption, which has been commonly adopted (Ahn et al., 2023; Mahankali et al., 2023) when analyzing into the dynamics of pre-training optimization of TFs:

**Assumption 4** (Gaussianity). *We assume that $P_\mathcal{X} = \mathcal{N}(0, \Sigma)$ with $\Sigma = \mathbf{U}\text{diag}(\lambda_1, \ldots, \lambda_d)\mathbf{U}^\top$, and $\mathbf{w}_\star$ is sampled from $\mathcal{N}(0, I_d)$.*

We use $\mathbf{S}^b, \mathbf{T}^b$ to denote the parameters of TFs **pre-trained on the confounded data**. We first show that quantifying the difference between the converged parameters $\mathbf{S}^b, \mathbf{T}^b$ in EICL and the grounding parameters $\mathbf{S}^u, \mathbf{T}^u$ *can be reduced to only a small subset of each parameter matrix*:

**Lemma 2** (Parameter Simplification). *During the pre-training phase, analyzing the variation of $\{\overline{\mathbf{S}^b}, \mathbf{T}_{:,j}^b\}_{j=1}^d$ equals to analyzing that of $\mathbf{S}^b, \mathbf{T}^b$, where $\overline{\mathbf{S}}$ refers to the last row of $\mathbf{S}^b$, and $\mathbf{T}_{:,j}^b$ ($1 \le j \le d$) refers to the first $j$-th column of $\mathbf{T}^b$.*

Unfortunately, we find that directly quantify difference between the unbiased TF parameters $\{\mathbf{S}^b, \mathbf{T}^b\}$ and the converged parameters (potential biased) $\{\mathbf{S}^u, \mathbf{T}^u\}$ exhibits challenges, since the spurious correlation $r_j = \mathbb{E}[X_j\epsilon]$ prevents the derivation of a closed-form expression of $\mathbf{S}^b, \mathbf{T}^b$ as in equation 5 (see (Ahn et al., 2023)). We therefore compare the two groups of TF parameters from the perspective of derivative instead, that is, quantifying *how $\mathbf{S}^b, \mathbf{T}^b$ and $\mathbf{S}^u, \mathbf{T}^u$ contribute to the pre-training objective $\mathcal{L}_{icl}$ differently:*

**Definition 3** (**Derivative Divergence**). *We measure the deviation of $\mathbf{S}^b, \mathbf{T}^b$ from $\mathbf{S}^u, \mathbf{T}^u$ as follows:*

$$
\Delta_{pre}^j\left(\overline{\mathbf{S}}, \overline{\mathbf{T}}\right) = \frac{d\mathcal{L}_{icl}}{d\mathbf{G}_{st}^j}\Bigg|_{\mathbf{G}_{st}^j = \overline{\mathbf{S}^b}, \overline{\mathbf{T}_{:,j}^b}} - \frac{d\mathcal{L}_{icl}}{d\mathbf{G}_{st}^j}\Bigg|_{\mathbf{G}_{st}^j = \overline{\mathbf{S}^u}, \overline{\mathbf{T}_{:,j}^u}}.
$$

Intuitively, $\Delta_{pre}^j$ quantifies the divergence between effects of $\overline{\mathbf{S}^b}, \overline{\mathbf{T}^b}$ and $\overline{\mathbf{S}^u}, \overline{\mathbf{T}^u}$ in the $j$-th feature-dim on $\mathcal{L}_{icl}$ during pre-training. We note that the derivatives of $\mathcal{L}_{icl}$ w.r.t. $\overline{\mathbf{S}^b}, \overline{\mathbf{T}^b}$ equals to 0, as $\overline{\mathbf{S}^b}, \overline{\mathbf{T}^b}$ refers to the global minima points with endogenous $\epsilon$.

### F.2.2 PROOFS OF PARAMETER SIMPLIFICATION

To simplify our proof of Lemma 2, we first introduce several useful lemmas:

**Lemma 3** (Ahn et al., 2023)*During the optimization of $\mathcal{L}_{icl}$, only $\overline{\mathbf{S}} \in \mathcal{R}^{1\times(d+1)}$ and $\overline{\mathbf{T}} \in \mathcal{R}^{(d+1)\times d}$ are updated, where $\overline{\mathbf{S}} = \mathbf{S}_{d+1,:}$ refers to the last row of $\mathbf{S}$ and $\overline{\mathbf{T}} = \mathbf{T}_{:,:d}$ refers to the first $d$ columns of $\mathbf{T}$.*

We use $\mathbf{S}, \mathbf{T}$ interleavely with notations $\overline{\mathbf{S}}, \overline{\mathbf{T}}$ without ambiguity. We then present another lemma for decomposition of $\overline{\mathbf{S}}$ and $\overline{\mathbf{T}}$ into $d$ pairs of weight parameters: $\{\overline{\mathbf{S}}, \overline{\mathbf{T}_{:,j}}\}_{j=1}^d$, where $\overline{\mathbf{T}_{:,j}} \in \mathcal{R}^{d+1}$ refers to the $j$-th column of $\overline{\mathbf{T}}$:

**Lemma 3** (Decomposition along $\overline{\mathbf{T}}$, $P_{\mathcal{X}} = \mathcal{N}(0, \mathrm{diag}(\lambda_1, \dots, \lambda_d))$) *Finding global minimum, i.e.,* $\overline{\mathbf{S}}, \overline{\mathbf{T}}$ *of* $\mathcal{L}_{icl}$ *can be equivalently turned into finding the global minimum of each component* $\mathcal{L}_{icl,j}$ *w.r.t.* $\overline{\mathbf{S}}^\top \overline{\mathbf{T}_{:,j}}^\top$:

$$\mathcal{L}_{icl,j} = \sum_{j=1}^d \mathbb{E}_{\mathbf{M}, \mathbf{w}_\star, \epsilon} \left[ \left\langle \mathbf{Z}, \mathbf{G}_{st}^j \right\rangle + w_\star[j] \right]^2 \mathbb{E}[\mathbf{x}^{(N)}[j]^2] + \sum_{j=1}^d \mathbb{E}_{\mathbf{M}, \mathbf{w}_\star, \epsilon} \left[ \left\langle \mathbf{Z}, \mathbf{G}_{st}^j \right\rangle \right] \beta_j, \quad (18)$$

*where* $\mathbf{G}_{st}^j = \overline{\mathbf{S}}^\top \overline{\mathbf{T}_{:,j}}^\top$, $\beta_j$ *refers to the correlation between* $X_j$ *and* $\epsilon$: $r_j = \mathbb{E}\left[\epsilon X_j\right]$, *and* $\langle \mathbf{A}, \mathbf{B} \rangle :=$ $\mathrm{Tr}(\mathbf{A}\mathbf{B}^\top)$ *for two matrices* $\mathbf{A}$ *and* $\mathbf{B}$ *here and below.*

*Proof.* We first simplify the original ICL objective:

$$\mathcal{L}_{icl}(\mathbf{S}, \mathbf{T}) = \mathbb{E}_{(\mathbf{M}, \mathbf{w}_\star)} \left[ \left( \mathrm{TF}_{\mathbf{S}, \mathbf{T}}^{pred}(\mathbf{M}) + y^{(n+1)} \right)^2 \right]$$

$$= \mathbb{E}_{(\mathbf{M}, \mathbf{w}_\star)} \left[ \left( \mathrm{TF}_{\mathbf{S}, \mathbf{T}}^{pred}(\mathbf{M}) + \mathbf{w}_\star^\top \mathbf{x}^{(n+1)} + \epsilon^{(n+1)} \right)^2 \right]$$

$$= \mathbb{E}_{(\mathbf{M}, \mathbf{w}_\star)} \left[ \left( \left( \mathbf{M} + \frac{1}{N_{tr}} \mathbf{SMC}(\mathbf{M}^\top \mathbf{TM}) \right)_{(d+1, n+1)} + \mathbf{w}_\star^\top \mathbf{x}^{(n+1)} + \epsilon^{(n+1)} \right)^2 \right]$$

$$= \mathbb{E}_{(\mathbf{M}, \mathbf{w}_\star)} \left[ \left( \left( \begin{bmatrix} \mathbf{x}^{(n+1)} \\ 0 \end{bmatrix} + \mathbf{S} \left( \underbrace{\frac{1}{n} \sum_{i=1}^n \mathbf{m}^{(i)}(\mathbf{m}^{(i)})^\top}_{:=\mathbf{Z}} \right) \mathbf{T} \begin{bmatrix} \mathbf{x}^{(n+1)} \\ 0 \end{bmatrix} \right)_{(d+1)} + \mathbf{w}_\star^\top \mathbf{x}^{(n+1)} + \epsilon^{(n+1)} \right)^2 \right]$$

$$= \mathbb{E}_{(\mathbf{M}, \mathbf{w}_\star)} \left[ \left( \overline{\mathbf{S}} \mathbf{Z} \overline{\mathbf{T}} \mathbf{x}^{(n+1)} + \mathbf{w}_\star^\top \mathbf{x}^{(n+1)} + \epsilon^{(n+1)} \right)^2 \right]$$

$$= \underbrace{\mathbb{E}\left[ \left( \overline{\mathbf{S}} \mathbf{Z} \overline{\mathbf{T}} + \mathbf{w}_\star^\top \right) \mathbf{x}^{(n+1)} \right]^2 + 2 \mathbb{E}\left[ \mathbf{x}^{(n+1)} \epsilon^{(n+1)} \left( \overline{\mathbf{S}} \mathbf{Z} \overline{\mathbf{T}} + \mathbf{w}_\star^\top \right) \right]}_{\textbf{Optimizable}} + \mathbb{E}\left[ \epsilon^{(n+1)} \right]^2,$$

$$(19)$$

where we further expand the first term as follows:

$$\mathbb{E}\left[ \left( \overline{\mathbf{S}} \mathbf{Z} \overline{\mathbf{T}} + \mathbf{w}_\star^\top \right) \mathbf{x}^{(n+1)} \right]^2 = \sum_{j=1}^d \lambda_j \mathbb{E}\left[ \left\langle \mathbf{Z}, \overline{\mathbf{S}}^\top \overline{\mathbf{T}_{:,j}}^\top \right\rangle + \mathbf{w}_\star[j] \right]^2, \quad (20)$$

as $\mathbb{E}[\mathbf{x}^{(n+1)}[j]\mathbf{x}^{(n+1)}[j']] = 0$ for $j \neq j'$ and $\mathbb{E}[\mathbf{x}^{(n+1)}[j]^2] = \lambda_j$. The second term is expanded as follows in an analog:

$$\mathbb{E}\left[ \epsilon^{(n+1)} \left( \overline{\mathbf{S}} \mathbf{Z} \overline{\mathbf{T}} + \mathbf{w}_\star^\top \right) \mathbf{x}^{(n+1)} \right] = \sum_{j=1}^d \beta_j \mathbb{E}\left[ \left\langle \mathbf{Z}, \overline{\mathbf{S}}^\top \overline{\mathbf{T}_{:,j}}^\top \right\rangle \right]. \quad (21)$$

Hence, the overall expression of $\mathcal{L}_{icl}$ can be written as follows:

$$\mathcal{L}_{icl} = \sum_{j=1}^d \lambda_j \mathbb{E}\left[ \left\langle \mathbf{Z}, \overline{\mathbf{S}}^\top \overline{\mathbf{T}_{:,j}}^\top \right\rangle + \mathbf{w}_\star[j] \right]^2 + 2\beta_j \mathbb{E}\left[ \left\langle \mathbf{Z}, \overline{\mathbf{S}}^\top \overline{\mathbf{T}_{:,j}}^\top \right\rangle \right]. \quad (22)$$

With the decomposed expression shown above, we further prove our lemma by performing induction on $d$. When $d = 1$, the objective becomes:

$$\mathcal{L}_{icl,1} = \lambda_1 \mathbb{E}_{\mathbf{M}, \mathbf{w}_\star, \epsilon} \left[ \left\langle \mathbf{Z}, \mathbf{G}_{st}^1 \right\rangle + \mathbf{w}_\star \right]^2 + 2 \mathbb{E}_{\mathbf{M}, \mathbf{w}_\star, \epsilon} \left[ \left\langle \mathbf{Z}, \mathbf{G}_{st}^1 \right\rangle \right] \eta_1. \quad (23)$$

By taking $\mathbf{G}^1_{st}$ as an integration, we observe that $\mathcal{L}_{icl,1}$ exhibits the quadratic-form w.r.t. $\mathbf{G}^1_{st}$, i.e., a convex formulation w.r.t. $\mathbf{G}^1_{st}$. Furthermore, when taking derivative of $\mathcal{L}_{icl,1}$ w.r.t. $\mathbf{G}^1_{st}$, we obtain the following expression:

$$\frac{\nabla \mathcal{L}_{icl,1}}{\nabla \mathbf{G}^1_{st}} = 2\lambda_1 \mathbb{E}_{\mathbf{M}, w_\star, \epsilon} \left[ \left( \langle \mathbf{Z}, \mathbf{G}^1_{st} \rangle + w_\star \right) \mathbf{Z} \right] + \eta_1 \mathbb{E}[\mathbf{Z}] = 0 \tag{24}$$

To achieve vanished derivative, we just have to show that there exists $\mathbf{G}^1_{st}$ such that:

$$\lambda_1 \overline{\mathbf{S}}^\top \mathbf{Z} \overline{\mathbf{T}_{:,j}} = -w_\star - \eta_1, \tag{25}$$

We now decompose $\mathbf{Z}$ due to its symmetry:

$$\mathbf{Z} = \mathbf{U}^{\mathbf{z}} \text{diag} (\lambda^z_1, \lambda^z_2) (\mathbf{U}^{\mathbf{z}})^{-1}, \tag{26}$$

and turn the above expression into:

$$\overline{\mathbf{S}}^\top \mathbf{U}^{\mathbf{z}} \text{diag} (\lambda^z_1, \lambda^z_2) (\mathbf{U}^{\mathbf{z}})^{-1} \overline{\mathbf{T}_{:,j}} \tag{27}$$

$$= \left( \overline{\mathbf{S}}^\top \mathbf{U}^{\mathbf{z}} \right) \text{diag} (\lambda^z_1, \lambda^z_2) \left( (\mathbf{U}^{\mathbf{z}})^{-1} \overline{\mathbf{T}_{:,j}} \right), \tag{28}$$

and we construct $\overline{\mathbf{S}}$ and $\overline{\mathbf{T}_{:,1}}$ such that $\left( \overline{\mathbf{S}}^\top \mathbf{U}^{\mathbf{z}} \right) = \left( \frac{-(w_\star + \eta_1)}{d\lambda_1 (\lambda^z_1)^2}, \frac{-(w_\star + \eta_1)}{d\lambda_1 (\lambda^z_2)^2} \right)$ and $\left( (\mathbf{U}^{\mathbf{z}})^{-1} \overline{\mathbf{T}_{:,j}} \right) = (\lambda^z_1, \lambda^z_2)$.

We assume that the our lemma holds for the case $d = d'$. For the case that $d = d' + 1$, given the the existence of $\overline{\mathbf{S}}^*, \overline{\mathbf{T}^*_{:,j}}_{j=1}^{d'}$ that achieves global minima of $\sum_{j=1}^{d'} \mathcal{L}_{icl,j}$, we only have to show that there exists $\overline{\mathbf{T}_{:,d'+1}}$ such that the pair $\overline{\mathbf{T}_{:,d'+1}}, \overline{\mathbf{S}}$ can achieves the global minima of $\mathcal{L}_{icl,d'+1}$. Be awaring of the fact that $\mathcal{L}_{icl,d'+1}$ is also convex w.r.t. $\mathbf{G}^{d'+1}_{st}$, we just have to construct $\overline{\mathbf{T}_{:,d'+1}}$ with given $\overline{\mathbf{S}}$ such that:

$$2\lambda^2_{d'+1} \mathbb{E}_{\mathbf{M}, \mathbf{w}_\star[d'+1], \epsilon} \left[ \left( \langle \mathbf{Z}, \mathbf{G}^{d'+1}_{st} \rangle + \mathbf{w}_\star[d'+1] \right) \mathbf{Z} \right] + \eta_{d'+1} \mathbb{E}[\mathbf{Z}] = 0, \tag{29}$$

where we further have to construct the vector $\overline{\mathbf{T}_{:,d'+1}}$ such that:

$$\left\langle \overline{\mathbf{S}}^*, \overline{\mathbf{T}_{:,d'+1}} \right\rangle = -\frac{\mathbf{w}_\star[d'+1] + \eta_{d'+1}}{2\lambda_{d'+1}}, \tag{30}$$

which is obviously feasible. Then our claims follows by induction. □

**Remark 3.** *Previous studies has been explored similar conclusion in (Ahn et al., 2023). However, the conclusion relies on the explicit closed-form expressions of $\mathbf{S}, \mathbf{T}$ with a shared $\overline{\mathbf{S}}$ across each $\overline{\mathbf{T}_{:,j}}$ by coincidence when hidden confounder is missing.*

### F.3 PROOF

#### F.3.1 PROOF OF THEOREM 1

**Theorem 1.** (Deviated Weights, $P_\mathcal{X} = \mathcal{N}(0, \text{diag}(\lambda_1, \ldots, \lambda_d))$) *Under Assumption 4 with* **endogenous** $\epsilon$, *we derive that:*

$$\Delta^j_{pre} (\overline{\mathbf{S}}, \overline{\mathbf{T}}) = r_j \begin{bmatrix} \mathbf{K}_1 & \mathbf{K}_2 \\ \mathbf{K}_2^\top & K_3 \end{bmatrix} + \frac{1}{n} \begin{bmatrix} \mathbf{w}_\star[j] Diag \left( \{ \frac{n-1}{n} \lambda_k \lambda_j + \mathbb{1}(k=j) \lambda^4_k \}_{k=1}^d \right) & \lambda_j \mathbf{e}_j \\ \lambda_j \mathbf{e}_j^\top & \mathcal{O}_n \left( \{ \lambda_j \}_{j=1}^d, \sigma^2 \right) \end{bmatrix}, \tag{31}$$

*where each sub-matrix are expressed as follows:*

$$\mathbf{K}_1 = \mathbf{w}_\star[j] \sigma^2 \Lambda_d,$$

$$\mathbf{K}_2 = \mathcal{O}_n \left( \left( \sum_v \mathbf{w}_\star[v] \right) \sigma^2 \right) \mathbf{r},$$

$$K_3 = \mathcal{O}_n(\sigma^2 + \sum_v \lambda_v),$$

*where $\mathbf{e}_j$ denotes the base vectors with $j$-th element as 1 and others as 0, $\Lambda_d = \text{diag}(\lambda_1, \ldots, \lambda_d)$, $\lambda \in \mathcal{R}^d$ and $\mathbf{r} \in \mathcal{R}^d$ are vectors consisting of $\lambda^d_{j=1}$ and $r^d_{j=1}$, respectively, $\sigma^2, \sigma^4$ denotes the 2-th and 4-th moments of $\epsilon$, respectively.*

*Proof.* Before our formal proof, we further clarify the decomposition of unbiased weight matrices, i.e., $\overline{\mathbf{S}}^u, \overline{\mathbf{T}}^u$ in our Lemma 4 as follows:

$$\overline{\mathbf{S}}^u = \left(\mathbf{w}_\star^\top, 0\right)$$
$$\overline{\mathbf{T}}^u_{:,j} = \mathbf{e}_j, \tag{32}$$

where we remove the scaling factor $1/2$ without loss of generality. Recalling the expression of $\mathcal{L}_{icl}$ w.r.t. $\mathbf{G}^j_{st}$ as follows:

$$\mathcal{L}_{icl,j} = \lambda_j \, \mathbb{E}_{\mathbf{M},\mathbf{w}_\star,\epsilon} \left[\left\langle \mathbf{Z}, \mathbf{G}^j_{st}\right\rangle + \mathbf{w}_\star[j]\right]^2 + \mathbb{E}_{\mathbf{M},\mathbf{w}_\star,\epsilon}\left[\left\langle \mathbf{Z}, \mathbf{G}^j_{st}\right\rangle\right]\eta_1, \tag{33}$$

with derivative as follows:

$$\frac{\nabla\mathcal{L}_{icl,j}}{\nabla\mathbf{G}^j_{st}} = \underbrace{2\lambda_j\,\mathbb{E}_{\mathbf{M},\mathbf{w}_\star,\epsilon}\left[\left\langle \mathbf{Z},\mathbf{G}^j_{st}\right\rangle\mathbf{Z}\right]}_{(1)} + \underbrace{\mathbb{E}_{\mathbf{M},\mathbf{w}_\star,\epsilon}\left[\mathbf{w}_\star[j]\mathbf{Z}\right]}_{(2)} + \underbrace{\beta_j\,\mathbb{E}\left[\mathbf{Z}\right]}_{(3)}. \tag{34}$$

Meanwhile, we also recall the expression of decomposed derivative as follows:

$$\Delta^j_{pre}\left(\overline{\mathbf{S}},\overline{\mathbf{T}}\right) = \left.\frac{\nabla\mathcal{L}_{icl}}{\nabla\mathbf{G}^j_{st}}\right|_{\overline{\mathbf{S}}^b,\overline{\mathbf{T}}^b_{:,j}} - \left.\frac{\nabla\mathcal{L}_{icl}}{\nabla\mathbf{G}^j_{st}}\right|_{\overline{\mathbf{S}}^u,\overline{\mathbf{T}}^u_{:,j}}, \tag{35}$$

and we compute two terms one by one. To be first, as $\overline{\mathbf{S}}^b, \overline{\mathbf{T}}^b_{:,j}$ corresponds to the global minimum of $\mathcal{L}_{icl,j}$ with endogenous $\epsilon$, we have:

$$\left.\frac{\nabla\mathcal{L}_{icl}}{\nabla\mathbf{G}^j_{st}}\right|_{\overline{\mathbf{S}}^b,\overline{\mathbf{T}}^b_{:,j}} = \mathbf{0}. \tag{36}$$

We then calculate $\left.\frac{\nabla\mathcal{L}_{icl}}{\nabla\mathbf{G}^j_{st}}\right|_{\overline{\mathbf{S}}^u,\overline{\mathbf{T}}^u_{:,j}}$ by expanding $\mathbf{Z}$ as follows:

$$\mathbf{Z} = \begin{bmatrix} \frac{1}{n}\sum_{i=1}^n \mathbf{x}^{(i)}(\mathbf{x}^{(i)})^\top & \frac{1}{n}\sum_{i=1}^n y^{(i)}\mathbf{x}^{(i)} \\ \frac{1}{n}\sum_{i=1}^n y^{(i)}(\mathbf{x}^{(i)})^\top & \frac{1}{n}\sum_{i=1}^n (y^{(i)})^2 \end{bmatrix}, \tag{37}$$

We then derive term (2) in equation 35 by further decomposition:

$$\mathbb{E}_{\mathbf{M},\mathbf{w}_\star,\epsilon}\left[\mathbf{w}_\star[j]\mathbf{Z}\right] = \mathbb{E}_{\mathbf{M},\mathbf{w}_\star,\epsilon}\left[\mathbf{w}_\star[j]\begin{bmatrix} \frac{1}{n}\sum_{i=1}^n \mathbf{x}^{(i)}(\mathbf{x}^{(i)})^\top & \frac{1}{n}\sum_{i=1}^n y^{(i)}\mathbf{x}^{(i)} \\ \frac{1}{n}\sum_{i=1}^n y^{(i)}(\mathbf{x}^{(i)})^\top & \frac{1}{n}\sum_{i=1}^n (y^{(i)})^2 \end{bmatrix}\right]$$

$$= \mathbb{E}_{\mathbf{M},\mathbf{w}_\star,\epsilon}\left[\mathbf{w}_\star[j]\begin{bmatrix} \mathbf{0}_{d\times d} & \frac{1}{n}\sum_{i=1}^n y^{(i)}\mathbf{x}^{(i)} \\ \frac{1}{n}\sum_{i=1}^n y^{(i)}(\mathbf{x}^{(i)})^\top & \frac{1}{n}\sum_{i=1}^n (y^{(i)})^2 \end{bmatrix}\right]$$

$$= \mathbb{E}_{\mathbf{M},\mathbf{w}_\star,\epsilon}\left[\mathbf{w}_\star[j]\begin{bmatrix} \mathbf{0}_{d\times d} & \frac{1}{n}\sum_{i=1}^n \left(\mathbf{w}_\star^\top\mathbf{x}^{(i)}+\epsilon^{(i)}\right)\mathbf{x}^{(i)} \\ \frac{1}{n}\sum_{i=1}^n \left(\mathbf{w}_\star^\top\mathbf{x}^{(i)}+\epsilon^{(i)}\right)(\mathbf{x}^{(i)})^\top & \frac{1}{n}\sum_{i=1}^n (y^{(i)})^2 \end{bmatrix}\right]$$

$$= \begin{bmatrix} \mathbf{0}_{d\times d} & \lambda_j\mathbf{e}_j \\ \lambda_j\mathbf{e}_j^\top & \frac{1}{n}\mathbb{E}_{\mathbf{M},\mathbf{w}_\star,\epsilon}\left[\mathbf{w}_\star[j]\sum_{i=1}^n \left(\mathbf{w}_\star^\top\mathbf{x}^{(i)}+\epsilon^{(i)}\right)^2\right] \end{bmatrix}$$

$$= \begin{bmatrix} \mathbf{0}_{d\times d} & \lambda_j\mathbf{e}_j \\ \lambda_j\mathbf{e}_j^\top & 2\beta_j, \end{bmatrix} \tag{38}$$

where the second equality is due to the fact that $\mathbf{w}_\star$ is symmetric, and the third equality is due to $\mathbb{E}\left[\mathbf{w}_\star[j]^2\,\mathbf{x}^{(i)}[j]\,\mathbf{x}^{(i)}[k]\right] = \lambda_j\mathbb{1}_{[j=k]}$, and the final equality is due to the derivation that:

$$\mathbb{E}_{\mathbf{M},\mathbf{w}_\star,\epsilon}\left[\mathbf{w}_\star[j]\sum_{i=1}^n \left(\mathbf{w}_\star^\top\mathbf{x}^{(i)}+\epsilon^{(i)}\right)^2\right] = \sum_{i=1}^n \mathbb{E}\left[\mathbf{w}_\star[j]\left(\mathbf{w}_\star^\top\mathbf{x}^{(i)}\right)^2 + \mathbf{w}_\star[j]\left(\epsilon^{(i)}\right)^2 + 2\mathbf{w}_\star[j]\left(\mathbf{w}_\star^\top\mathbf{x}^{(i)}\right)\epsilon^{(i)}\right]$$

$$= 2\sum_{i=1}^n \mathbb{E}\left[\mathbf{w}_\star[j]\left(\mathbf{w}_\star^\top\mathbf{x}^{(i)}\right)\epsilon^{(i)}\right]$$

$$= 2\mathbb{E}\left[\mathbf{w}_\star[j]\mathbf{w}_\star[j]\mathbf{x}^{(i)}[j]\epsilon^{(i)}\right]$$

$$= 2\beta_j \tag{39}$$

We then derive term (3) in equation 35 by further decomposition:

$$
\beta_j \, \mathbb{E}\left[\mathbf{Z}\right] = \beta_j \, \mathbb{E}\left[\begin{matrix} \frac{1}{n}\sum_{i=1}^{n}\mathbf{x}^{(i)}(\mathbf{x}^{(i)})^{\top} & \frac{1}{n}\sum_{i=1}^{n} y^{(i)}\mathbf{x}^{(i)} \\ \frac{1}{n}\sum_{i=1}^{n} y^{(i)}(\mathbf{x}^{(i)})^{\top} & \frac{1}{n}\sum_{i=1}^{n}(y^{(i)})^2 \end{matrix}\right]
$$

$$
= \beta_j \left[\begin{matrix} \mathbf{\Lambda}_d & \mathbb{E}\left[\frac{1}{n}\sum_{i=1}^{n}\left(\mathbf{w_\star}^{\top}\mathbf{x}^{(i)} + \epsilon^{(i)}\right)\mathbf{x}^{(i)}\right] \\ \mathbb{E}\left[\frac{1}{n}\sum_{i=1}^{n}\left(\mathbf{w_\star}^{\top}\mathbf{x}^{(i)} + \epsilon^{(i)}\right)(\mathbf{x}^{(i)})^{\top}\right] & \mathbb{E}\left[\frac{1}{n}\sum_{i=1}^{n}\left(\mathbf{w_\star}^{\top}\mathbf{x}^{(i)} + \epsilon^{(i)}\right)^2\right] \end{matrix}\right]
$$

$$
= \beta_j \left[\begin{matrix} \mathbf{\Lambda}_d & \mathbf{r} \\ \mathbf{r}^{\top} & \mathbb{E}\left[\frac{1}{n}\sum_{i=1}^{n}\left(\mathbf{w_\star}^{\top}\mathbf{x}^{(i)} + \epsilon^{(i)}\right)^2\right] \end{matrix}\right]
$$

$$
= \beta_j \left[\begin{matrix} \mathbf{\Lambda}_d & \mathbf{r} \\ \mathbf{r}^{\top} & \sigma^2 + \sum_l \lambda_l \end{matrix}\right],
$$

(40)

where the final equality is due to the fact that:

$$
\mathbb{E}\left[\left(\mathbf{w_\star}^{\top}\mathbf{x}^{(i)} + \epsilon^{(i)}\right)^2\right] = \mathbb{E}\left[\left(\mathbf{w_\star}^{\top}\mathbf{x}^{(i)}\right)^2 + \left(\epsilon^{(i)}\right)^2 + 2\epsilon^{(i)}\left(\mathbf{w_\star}^{\top}\mathbf{x}^{(i)}\right)\right]
$$

$$
= \sigma^2 + \mathbb{E}\left[\left(\mathbf{w_\star}^{\top}\mathbf{x}^{(i)}\right)^2\right]
$$

$$
= \sigma^2 + \sum_{j=1}^{d}\mathbb{E}\left[\mathbf{w_\star}[j]^2\left(\mathbf{x}^{(i)}[j]\right)^2\right]
$$

$$
= \sigma^2 + \sum_{l=1}^{d}\lambda_l.
$$

(41)

Finally, we derive term (1) in equation 35 in three sub-blocks:

$$
\mathbf{Z} = \left[\begin{matrix} \underbrace{\frac{1}{n}\sum_{i=1}^{n}\mathbf{x}^{(i)}(\mathbf{x}^{(i)})^{\top}}_{\mathbf{Z}^{(a)}} & \underbrace{\frac{1}{n}\sum_{i=1}^{n} y^{(i)}\mathbf{x}^{(i)}}_{\mathbf{Z}^{(b)}} \\ \underbrace{\frac{1}{n}\sum_{i=1}^{n} y^{(i)}(\mathbf{x}^{(i)})^{\top}}_{\mathbf{Z}^{(b)}} & \underbrace{\frac{1}{n}\sum_{i=1}^{n}(y^{(i)})^2}_{\mathbf{Z}^{(c)}} \end{matrix}\right]
$$

(42)

Then we derive the following equations for $v \leq d$:

$$
\mathbb{E}_{\mathbf{M},\mathbf{w_\star},\epsilon}\left[\left\langle \mathbf{Z}^{(a)}, \mathbf{E}_{v,j}\right\rangle \mathbf{Z}^{(a)}\right]_{lk} = \frac{1}{n^2}\sum_{i_1=1}^{n}\sum_{i_2=1}^{n}\mathbb{E}\left[\mathbf{x}^{(i_1)}[v]\mathbf{x}^{(i_1)}[j]\mathbf{x}^{(i_2)}[l]\mathbf{x}^{(i_2)}[k]\right]
$$

$$
= \frac{1}{n^2}\sum_{i}^{n}\mathbb{E}\left[\mathbf{x}^{(i_1)}[v]\mathbf{x}^{(i_1)}[j]\mathbf{x}^{(i)}[l]\mathbf{x}^{(i)}[k]\right]
$$

$$
+ \frac{1}{n^2}\sum_{i_1 \neq i_2}^{n}\mathbb{E}\left[\mathbf{x}^{(i_1)}[v]\mathbf{x}^{(i_1)}[j]\right]\mathbb{E}\left[\mathbf{x}^{(i_2)}[l]\mathbf{x}^{(i_2)}[k]\right]
$$

$$
= \frac{1}{n^2}\sum_{i}^{n}\mathbb{1}(v = j = l = k)\lambda_j^4 + \frac{1}{n^2}\sum_{i_1 \neq i_2}^{n}\mathbb{1}(v = j)\lambda_j\mathbb{1}(l = k)\lambda_l
$$

(43)

where we utilize the fact that zero correlation in joint Gaussian implies independence. Hence, we conclude the expression as follows for $v = j$:

$$
\mathbb{E}_{\mathbf{M},\mathbf{w_\star},\epsilon}\left[\left\langle \mathbf{Z}^{(a)}, \mathbf{E}_{j,j}\right\rangle \mathbf{Z}^{(a)}\right] = \frac{1}{n^2}\left[\begin{matrix} n(n-1)\lambda_j\lambda_1 & 0 & 0 & 0 & 0 \\ 0 & n(n-1)\lambda_j\lambda_2, & 0 & 0 & 0 \\ 0 & 0 & \cdots & \cdots, n(n-1)\lambda_j\lambda_j + n\lambda_j^4 & \cdots \\ 0 & 0 & \cdots & 0 & n(n-1)\lambda_j\lambda_d \end{matrix}\right],
$$

(44)

where $\mathbb{E}_{\mathbf{M},\mathbf{w}_\star,\epsilon}\left[\left\langle \mathbf{Z}^{(a)}, \mathbf{E}_{j,j}\right\rangle \mathbf{Z}^{(a)}\right] = \mathbf{0}$ when $v \neq j$. We then calculate the second term as follows:

$$
\mathbb{E}_{\mathbf{M},\mathbf{w}_\star,\epsilon}\left[\left\langle \mathbf{Z}^{(b)}, \mathbf{E}_{v,j}\right\rangle \mathbf{Z}^{(b)}\right]_l = \frac{1}{n^2}\sum_{i_1=1}^n\sum_{i_2=1}^n \mathbb{E}\left[\mathbf{x}^{(i_1)}[v]\mathbf{x}^{(i_1)}[j]\left(\mathbf{w}_\star^\top\mathbf{x}^{(i_2)} + \epsilon^{(i_2)}\right)\mathbf{x}^{(i_2)}[l]\right]
$$

$$
= \frac{1}{n^2}\sum_{i_1=1}^n\sum_{i_2=1}^n \mathbb{E}\left[\mathbf{x}^{(i_1)}[v]\mathbf{x}^{(i_1)}[j]\epsilon^{(i_2)}\mathbf{x}^{(i_2)}[l]\right]
$$

$$
= \underbrace{\frac{1}{n^2}\sum_{i_1\neq i_2} \mathbb{E}\left[\mathbf{x}^{(i_1)}[v]\mathbf{x}^{(i_1)}[j]\right]\mathbb{E}\left[\epsilon^{(i_2)}\mathbf{x}^{(i_2)}[l]\right]}_{\text{①}}
$$

$$
+ \underbrace{\frac{1}{n^2}\sum_i \mathbb{E}\left[\mathbf{x}^{(i)}[v]\mathbf{x}^{(i)}[j]\epsilon^{(i)}\mathbf{x}^{(i)}[l]\right]}_{\text{②}}
$$

$$(45)$$

where ① can be derived as follows:

$$
\frac{1}{n^2}\sum_{i_1\neq i_2} \mathbb{E}\left[\mathbf{x}^{(i_1)}[v]\mathbf{x}^{(i_1)}[j]\right]\mathbb{E}\left[\epsilon^{(i_2)}\mathbf{x}^{(i_2)}[l]\right] = \mathbb{1}(v = j)\frac{1}{n^2}\left(n(n-1)r_l\sigma^2 r_j\right) \tag{46}
$$

and ② can be derived as follows:

$$
\frac{1}{n^2}\sum_i \mathbb{E}\left[\mathbf{x}^{(i_1)}[v]\mathbf{x}^{(i)}[j]\epsilon^{(i)}\mathbf{x}^{(i)}[l]\right] = \frac{1}{n^2}\sum_i \mathbb{E}\left[\epsilon^{(i)}(r_j\epsilon^{(i)} + \delta_j^{(i)})(r_v\epsilon^{(i)} + \delta_v^{(i)})(r_l\epsilon^{(i)} + \delta_l^{(i)})\right]
$$

$$
= \frac{1}{n}\left(r_j r_v r_l \sigma^4 + \mathbb{E}\left[r_v\delta_l\delta_j + r_j\delta_l\delta_v + r_l\delta_j\delta_v\right]\right)
$$

$$
= \frac{1}{n}\left(r_j r_v r_l \sigma^4 + \sigma^2\left(\mathbb{1}(l=j)r_v + \mathbb{1}(v=j)r_l + \mathbb{1}(v=l)r_j\right)\right), \tag{47}
$$

where we invokes linear confounding effect to derive the expectation. Finally, we calculate the third term, i.e., $\mathbb{E}_{\mathbf{M},\mathbf{w}_\star,\epsilon}\left[\left\langle \mathbf{Z}^{(c)}, \mathbf{E}_{v,j}\right\rangle \mathbf{Z}^{(c)}\right]$ as follows:

$$
\mathbb{E}_{\mathbf{M},\mathbf{w}_\star,\epsilon}\left[\left\langle \mathbf{Z}^{(c)}, \mathbf{E}_{v,j}\right\rangle \mathbf{Z}^{(c)}\right] = \frac{1}{n^2}\sum_{i_1=1}^n\sum_{i_2=1}^n\left(\underbrace{\mathbb{E}\left[\mathbf{x}^{(i_1)}[v]\mathbf{x}^{(i_1)}[j]\left(\epsilon^{(i_2)}\right)^2\right]}_{\text{①}} + \underbrace{\mathbb{E}\left[\mathbf{x}^{(i_1)}[v]\mathbf{x}^{(i_1)}[j]\left(\mathbf{w}_\star^\top\mathbf{x}^{(i_2)}\mathbf{x}^{(i_2)\top}\mathbf{w}_\star\right)\right]}_{\text{②}}\right)
$$

$$(48)$$

where we further expand each term in below:

For Term ① $\quad \frac{1}{n^2}\sum_{i_1=1}^n\sum_{i_2=1}^n \mathbb{E}\left[\mathbf{x}^{(i_1)}[v]\mathbf{x}^{(i_1)}[j]\left(\epsilon^{(i_2)}\right)^2\right] = \begin{cases} \frac{n-1}{n}\mathbb{1}(v=j)\lambda_j\sigma^2 & \text{if } i_1 \neq i_2, \\ \frac{1}{n}\left(r_v r_j\sigma^4 + \mathbb{1}(v=j)\sigma^2\right) & \text{if } i_1 = i_2, \end{cases}$

$$(49)$$

where we invokes linear confounding effect to derive the expectation. For the term ②, we perform the following derivation:

$$
\frac{1}{n^2}\sum_{i_1=1}^n\sum_{i_2=1}^n \mathbb{E}\left[\mathbf{x}^{(i_1)}[v]\mathbf{x}^{(i_1)}[j]\left(\mathbf{w}_\star^\top\mathbf{x}^{(i_2)}\mathbf{x}^{(i_2)\top}\mathbf{w}_\star\right)\right]
$$

$$
= \frac{1}{n^2}\sum_{i_1=1}^n\sum_{i_2=1}^n \mathbb{E}\left[\mathbf{w}_\star^\top\left(\mathbf{x}^{(i_1)}[v]\mathbf{x}^{(i_1)}[j]\mathbf{x}^{(i_2)}\mathbf{x}^{(i_2)\top}\right)\mathbf{w}_\star\right]
$$

$$
= \frac{1}{n^2}\sum_{i_1=1}^n\sum_{i_2=1}^n\sum_{k=1}^d \mathbb{E}\left[\mathbf{x}^{(i_1)}[v]\mathbf{x}^{(i_1)}[j](\mathbf{x}^{(i_2)}[k])^2\right]
$$

$$
= \mathbb{1}(v=j)\frac{1}{n}\left(\left(\lambda_j^4 + \lambda_j\sum_{k\neq j}\lambda_k\right) + (n-1)\lambda_j\left(\sum_k\lambda_k\right)\right) = \mathcal{O}\left(\{\lambda_j\}_{j=1}^d\right)
$$

$$(50)$$

Now every components are calculated, and we are ready to summarize our main results by summing over $v \in [d]$ with weight $\mathbf{w}_\star[v]$:

- For the upper-left area of $\Delta_{pre}^j$, i.e., $(\Delta_{pre}^j)_{d \times d}$, we have the following equations by combining equation 40 and equation 44:

$$
\mathbf{w}_\star[j]\text{Diag}\left(\frac{n-1}{n}\lambda_j\lambda_1, \frac{n-1}{n}\lambda_j\lambda_2, \cdots, \frac{n-1}{n}\lambda_j\lambda_j + \frac{1}{n}\lambda_j^4, \cdots, \frac{n-1}{n}\lambda_d\lambda_1\right) + \beta_j\Lambda_d
$$
$$
= \mathbf{w}_\star[j]\text{Diag}\left(\frac{n-1}{n}\lambda_j\lambda_1, \frac{n-1}{n}\lambda_j\lambda_2, \cdots, \frac{n-1}{n}\lambda_j\lambda_j + \frac{1}{n}\lambda_j^4, \cdots, \frac{n-1}{n}\lambda_d\lambda_1\right) + \mathbf{w}_\star[j]r_j\sigma^2\Lambda_d
$$
(51)

- The upper-right area $(\Delta_{pre}^j)_{d \times 1}$ can be derived as follows by combining equation 46, equation 47, equation 38 and equation 40:

$$
(\Delta_{pre}^j)_{d \times 1} = \frac{1}{n}\left(r_j(\sum_v \mathbf{w}_\star[v]r_v)\mathbf{r} + \mathbf{r} + \mathcal{O}_n(1) + (\sum_v \mathbf{w}_\star[v])(2n-1)r_j\mathbf{r}\sigma^2\right) + \lambda_j\mathbf{e}_j
$$
$$
= r_j\mathcal{O}_n\left(\left(\sum_v \mathbf{w}_\star[v]\right)\sigma^2\right)\mathbf{r} + \lambda_j\mathbf{e}_j.
$$
(52)

- The lower-right area can be derived as follows by combining equation 50, equation 49,

$$
(\Delta_{pre}^j)_{(d+1) \times (d+1)} = \mathcal{O}_n\left(\{\lambda_j\}_{j=1}^d, \sigma^2\right) + r_j\mathcal{O}_n(3\sigma^2 + \sum_v \lambda_v)
$$
$$
= \mathcal{O}_n\left(\{\lambda_j\}_{j=1}^d, \sigma^2\right) + r_j\mathcal{O}_n(\sigma^2 + \sum_v \lambda_v).
$$
(53)

$\square$

**Prediction Weight $\mathbf{w}^b$ with endogenous noise $\epsilon$.** We show that inside the forward pass of ICL inference, deviated parameters $\mathbf{S}^b$, $\mathbf{T}^b$ still induces implicit gradient weights $\mathbf{w}^b$. Consider the fact that $\mathbf{G}_{st}^{b,j} = \mathbf{G}_{st}^{u,j} + \delta_{st}^j$, and we have:

$$
\hat{y}^{(n+1)}[j] = \text{TF}_{\mathbf{S},\mathbf{T}}^{pred}(\mathbf{M})[j] = \left\langle \mathsf{Z}, \mathbf{G}_{st}^{b,j} \right\rangle \mathbf{x}^{(n+1)}[j]
$$
$$
= \left\langle \mathsf{Z}, \mathbf{G}_{st}^{u,j} + \delta_{st}^j \right\rangle \mathbf{x}^{(n+1)}[j]
$$
(54)
$$
= \mathbf{x}^{(n+1)}[j]\mathbf{w}^{GD}[j] + \left\langle \mathsf{Z}, \delta_{st}^j \right\rangle \mathbf{x}^{(n+1)}[j],
$$

where we let $\mathbf{w}^b[j] = \mathbf{w}^{GD}[j] + \left\langle \mathsf{Z}, \delta_{st}^j \right\rangle$.

### F.3.2 PROOF OF THEOREM 2

**Theorem 2. (Estimation Bias)** *By assuming the lower-boundness of the sample-covariance matrix $\mathbf{Z} = \mathbf{m}^{(i)}(\mathbf{m}^{(i)})^\top$, i.e., $\alpha_1 \leq \min_{1 \leq l,k \leq d} \mathbf{Z}_{kl} \leq \alpha_2$, and the finite second-moment of $\mathbf{Z}$: $\max_{kl} Var[\mathbf{Z}] \leq \kappa_Z$, we have the following inequality holds with probability in $1 - \sum_l q_l$:*

$$
\Delta_{est}^w[j] \geq \min\left(\frac{K(n,\lambda,\mathbf{r},\sigma^2,\sigma^4)}{d\alpha_2} + \sum_l \sqrt{\frac{\kappa_Z}{q_{d+1,l}d\alpha_2}}, \frac{K(n,\lambda,\mathbf{r},\sigma^2,\sigma^4)}{d\alpha_1} - \sum_l \sqrt{\frac{\kappa_Z}{q_{d+1,l}d\alpha_1}}\right),
$$
(55)

*Proof.* We observe the following equations during the forward process:

$$
\widehat{y_b^{(n+1)}}[j] = \text{TF}_{\mathbf{S}^b,\mathbf{T}^b}^{pred}(\mathbf{M})[j] = \left\langle \mathsf{Z}, \mathbf{G}_{st}^{b,j} \right\rangle \mathbf{x}^{(n+1)}[j] \quad \textbf{Biased}
$$
(56)

$$\widehat{y_u^{(n+1)}}[j] = \mathrm{TF}^{pred}_{\mathbf{S}^u, \mathbf{T}^u}(\mathbf{M})[j] = \left\langle \mathsf{Z}, \mathbf{G}^{u,j}_{st} \right\rangle \mathbf{x}^{(n+1)}[j] \quad \textbf{Unbiased} \tag{57}$$

Then we only have to compare $\left\langle \mathsf{Z}, \mathbf{G}^{u,j}_{st} \right\rangle$ and $\left\langle \mathsf{Z}, \mathbf{G}^{b,j}_{st} \right\rangle$, where their difference are already shown in equation 34 the proof of Theorem 1:

$$\mathbb{E}_{\mathbf{M},\mathbf{w}_\star,\epsilon}\left[\left\langle \mathsf{Z}, \mathbf{G}^{j,b}_{st} \right\rangle \mathbf{Z}\right] - \mathbb{E}_{\mathbf{M},\mathbf{w}_\star,\epsilon}\left[\left\langle \mathsf{Z}, \mathbf{G}^{j,b}_{st} \right\rangle \mathbf{Z}\right] = \mathbb{E}_{\mathbf{M},\mathbf{w}_\star,\epsilon}\left[\Delta^w_{est}[j]\mathbf{Z}\right], \tag{58}$$

where the quantity $\mathbb{E}_{\mathbf{M},\mathbf{w}_\star,\epsilon}\left[\Delta^w_{est}[j]\mathbf{Z}\right]$ has already been computed by combining. Hence, by invoking Chebyshev's Inequality on the region of $\mathbf{Z}^b$ (lower-left area in equation 42), we have the following inequality holds with probability $1 - q_{d+1,l}$ for $1 \le l \le d$:

$$\left|\Delta^w_{est}[j]\mathbf{Z}_{d+1,l} - \mathbb{E}_{\mathbf{M},\mathbf{w}_\star,\epsilon}\left[\left\langle \mathbf{Z}^{(b)}, B(n,\lambda)\mathbf{E}_{d+1,j} \right\rangle \mathbf{Z}^{(b)}\right]_l\right| \le \sqrt{\frac{\mathrm{Var}_{\mathbf{M},\mathbf{w}_\star,\epsilon}\left[\Delta^w_{est}[j]\mathbf{Z}\right]_{d+1,j}}{q_{d+1,l}}} \le \sqrt{\frac{\kappa_Z}{q_{d+1,l}}} \tag{59}$$

where we further recall the expression of the lower-left region (in equation 52) as follows:

$$\mathbb{E}_{\mathbf{M},\mathbf{w}_\star,\epsilon}\left[\left\langle \mathbf{Z}^{(b)}, \mathbf{E}_{d+1,j} \right\rangle \mathbf{Z}^{(b)}\right] = (\Delta^j_{pre})_{d\times 1} - \lambda_j \mathbf{e}_j = r_j \mathcal{O}_n\left(\left(\sum_v \mathbf{w}_\star[v]\right)\sigma^2\right)\mathbf{r}.$$

we further perform the union bound of equation 59 over $l$, we further have the following inequality holding with probability $1 - \sum_j q_{d+1,l}$:

$$\sum_j \left|\Delta^w_{est}[j]\mathbf{Z}_{d+1,j} - \mathbb{E}_{\mathbf{M},\mathbf{w}_\star,\epsilon}\left[\left\langle \mathbf{Z}^{(b)}, B(n,\lambda)\mathbf{E}_{d+1,j} \right\rangle \mathbf{Z}^{(b)}\right]_l\right| \le \sum_l \sqrt{\frac{\kappa_Z}{q_{d+1,l}}}, \tag{60}$$

where the left side can be derived as follows:

$$\begin{aligned}
&\left|\sum_l \left(\Delta^w_{est}[j]\mathbf{Z}_{d+1,l} - \mathbb{E}_{\mathbf{M},\mathbf{w}_\star,\epsilon}\left[\left\langle \mathbf{Z}^{(b)}, B(n,\lambda)\mathbf{E}_{d+1,j} \right\rangle \mathbf{Z}^{(b)}\right]_l\right)\right| \\
&\le \sum_l \left|\Delta^w_{est}[j]\mathbf{Z}_{d+1,l} - \mathbb{E}_{\mathbf{M},\mathbf{w}_\star,\epsilon}\left[\left\langle \mathbf{Z}^{(b)}, B(n,\lambda)\mathbf{E}_{d+1,j} \right\rangle \mathbf{Z}^{(b)}\right]_l\right|,
\end{aligned} \tag{61}$$

with:

$$\begin{aligned}
&\left|\sum_l \left(\Delta^w_{est}[j]\mathbf{Z}_{d+1,l} - \mathbb{E}_{\mathbf{M},\mathbf{w}_\star,\epsilon}\left[\left\langle \mathbf{Z}^{(b)}, B(n,\lambda)\mathbf{E}_{d+1,j} \right\rangle \mathbf{Z}^{(b)}\right]_l\right)\right| \\
&= \left|\Delta^w_{est}[j]\sum_l \mathbf{Z}_{d+1,l} - r_j \mathcal{O}_n\left(\left(\sum_v \mathbf{w}_\star[v]\right)\sigma^2\right)\sum_l r_l\right|,
\end{aligned} \tag{62}$$

Then we substitute the above quantity back to equation 60, we obtain:

$$\Delta^w_{est}[j] \ge \min\left(\frac{r_j \mathcal{O}_n\left(\left(\sum_v \mathbf{w}_\star[v]\right)\sigma^2\right)\sum_l r_l}{d\alpha_2} + \sum_l \sqrt{\frac{\kappa_Z}{q_{d+1,l}d\alpha_2}}, \frac{r_j \mathcal{O}_n\left(\left(\sum_v \mathbf{w}_\star[v]\right)\sigma^2\right)\sum_l r_l}{d\alpha_1} - \sum_l \sqrt{\frac{\kappa_Z}{q_{d+1,l}d\alpha_1}}\right), \tag{63}$$

$\square$

We then turn everything back to the correlated case, i.e., $\mathbf{X}$ exhibits covariance matrix $\Sigma = \mathbf{U}\mathrm{diag}(\lambda_1, \dots, \lambda_d)\mathbf{U}^\top$ using the same trick as in (Ahn et al., 2023).

*Proof for Theorem 1 when $P_\mathcal{X} = \mathcal{N}(0, \Sigma)$.* We deduce the case with $P_\mathbf{X} = \mathcal{N}(0, \Sigma)$ to the previous case. By defining $\widetilde{x}^{(i)} := \mathbf{U}^\top \mathbf{x}^{(i)}$, we conclude that $\mathbb{E}[\widetilde{x}^{(i)}(\widetilde{x}^{(i)})^\top] = \Lambda_d$ due to the orthogonal properties of $\mathbf{U}$. With $\mathbf{x}^{(i)} = \mathbf{U}\widetilde{x}^{(i)}$ back to the loss decomposition in equation 18, we have:

$$\begin{aligned}
\mathbb{E}\left[\left(\overline{\mathbf{S}}\mathbf{Z}\overline{\mathbf{T}} + \mathbf{w}_\star^\top\right)\mathbf{x}^{(n+1)}\right]^2 &= \mathbb{E}\left[\left(\overline{\mathbf{S}}\mathbf{Z}\overline{\mathbf{T}} + \mathbf{w}_\star^\top\right)\mathbf{U}\widetilde{x}^{(n+1)}\right]^2 \\
&= \sum_{j=1}^d \lambda_j \mathbb{E}\left[\left(\overline{\mathbf{S}}\mathbf{Z}\overline{\mathbf{T}} + \mathbf{w}_\star^\top\right)\mathbf{U}[j]\right]^2,
\end{aligned} \tag{64}$$

and the vector $\left(\overline{\mathbf{S}}\mathbf{Z}\overline{\mathbf{T}} + \mathbf{w}_\star{}^\top\right)\mathbf{U}$ can be transformed as follows:

$$
\begin{aligned}
\left(\overline{\mathbf{S}}\mathbf{Z}\overline{\mathbf{T}} + \mathbf{w}_\star{}^\top\right)\mathbf{U} &= \frac{1}{n}\sum_i \overline{\mathbf{S}}^\top \begin{bmatrix} \mathbf{x}^{(i)} \\ \left\langle \mathbf{x}^{(i)}, \mathbf{w}_\star\right\rangle + \epsilon^{(i)} \end{bmatrix}^{\otimes 2} \overline{\mathbf{T}}\mathbf{U} + \mathbf{w}_\star{}^\top \mathbf{U} \\
&= \frac{1}{n}\sum_i \overline{\mathbf{S}}^\top \begin{bmatrix} \mathbf{U}\widetilde{x}^{(i)} \\ \left\langle \mathbf{U}\widetilde{x}^{(i)}, \mathbf{w}_\star\right\rangle + \mathbf{U}\epsilon^{(i)}\mathbf{U}^\top \end{bmatrix}^{\otimes 2} \overline{\mathbf{T}}\mathbf{U} + \mathbf{w}_\star{}^\top \mathbf{U} \qquad (65) \\
&= \frac{1}{n}\sum_i \overline{\mathbf{S}}^\top \begin{bmatrix} \mathbf{U} & 0 \\ 0 & 1 \end{bmatrix} \begin{bmatrix} \widetilde{x}^{(i)} \\ \left\langle \widetilde{x}^{(i)}, \mathbf{w}_\star\right\rangle + \epsilon^{(i)} \end{bmatrix}^{\otimes 2} \begin{bmatrix} \mathbf{U}^\top & 0 \\ 0 & 1 \end{bmatrix} \overline{\mathbf{T}}\mathbf{U} + \mathbf{w}_\star{}^\top \mathbf{U}
\end{aligned}
$$

where we use $\mathbf{x}^{\otimes 2}$ to denote the operation $\mathbf{x}^{\otimes 2} = \mathbf{x}\mathbf{x}^\top$. Letting $\overline{\mathbf{S}}^\top \begin{bmatrix} \mathbf{U} & 0 \\ 0 & 1 \end{bmatrix}$ and $\begin{bmatrix} \mathbf{U}^\top & 0 \\ 0 & 1 \end{bmatrix} \overline{\mathbf{T}}\mathbf{U}$ be the new weight parameters, respectively, with the fact that $\mathbf{w}_\star{}^\top \mathbf{U}$ stills follows $\mathcal{N}(0, I_d)$, the case is deduced to previous cases, and our final claim of Theorem 1 follows. $\qquad\square$

*Proof for Theorem 2 when $P_\mathcal{X} = \mathcal{N}(0, \Sigma)$.* We note that $\left\langle \mathsf{Z}, \mathbf{G}_{st}^{u,j}\right\rangle$ and $\left\langle \mathsf{Z}, \mathbf{G}_{st}^{b,j}\right\rangle$ remain invariant under the transformation $\widetilde{x}^{(i)} := \mathbf{U}^\top \mathbf{x}^{(i)}$ due to the fact that $\mathbf{U}\mathbf{U}^T = I$, and our claim on $\left\langle \mathsf{Z}, \mathbf{G}_{st}^{u,j}\right\rangle - \left\langle \mathsf{Z}, \mathbf{G}_{st}^{b,j}\right\rangle$ follows the same. $\qquad\square$

### F.3.3 PROOF OF THEOREM 3

**Theorem 3. (Implicit Implementation of Debiasing Algorithm)** *Consider the TF with parameterized by* **biased** $\mathbf{S}^b, \mathbf{T}^b$, *and let* $\hat{y}^{(n+1)}$ *be predicted $y$ such that* $\hat{y}^{(n+1)} = TF_{\mathbf{S}^b, \mathbf{T}^b}^{pred}(\mathbf{M})$. *Then it holds that* $\hat{y}^{(n+1)} = \langle \mathbf{x}^{(n+1)}, \mathbf{w}^{DEB}\rangle$ *where* $\{\mathbf{w}_\ell^{GD}\}$ *is defined as* $\mathbf{w}_0^{DEB}$ *equaling to some constant w.r.t. input data and pre-trained weights:*

$$
\mathbf{w}_1^{DEB} = \mathbf{w}_0^{DEB} - \eta\nabla\mathcal{L}(\mathbf{w}^{DEB}) \tag{66}
$$

$$
\text{where} \quad \mathcal{L}^{deb}(\mathbf{w}) := \frac{1}{2n}\sum_{i=1}^n (y^{(i)} - \hat{y^{(i)}}_b - \mathbf{w}^\top \mathbf{x}^{(i)})^2, \tag{67}
$$

*where* $\hat{y^{(i)}}_b$ *refers to the biased prediction from pre-trained TF equipped with* $\mathbf{S}^b, \mathbf{T}^b$.

*Proof.* We first note two observations as follows:

---

1. The parameters $\overline{S}, \overline{T}$ can be decomposed using the feature-wise expression:
$$
\mathbf{S}^u, \mathbf{T}^u \Leftrightarrow \{\mathbf{G}_{st}^{u,j}\}_{j=1}^d \quad \mathbf{S}^b, \mathbf{T}^b \Leftrightarrow \{\mathbf{G}_{st}^{b,j}\}_{j=1}^d \tag{68}
$$
and we note that $\mathbf{G}_{st}^{u,j} = \delta_{st}^j + \mathbf{G}_{st}^{b,j}$.

---

2. The pre-trained parameters of TF is fixed, while inputing different data matrix will invoke different unbiased implicit gradient for the same unbiased parameters $\mathbf{G}_{st}^{u,j}$, as shown in Lemma 4.

---

With the above observations, we first derive the behaviors of the biased parameters $\{\mathbf{G}_{st}^{b,j}\}_{j=1}^d$ when meeting the residual input matrix $\mathbf{Z}^r$. By treating the residual term $y^{(i)} - \hat{y^{(i)}}_b$ as the new label $y^{(i)}$, we invoke Lemma 4 and conclude that $\mathbf{S}^u, \mathbf{T}^u$ achieves unbiased implicit gradient w.r.t.

the loss $\mathcal{L}^{deb}(\mathbf{w})$ implemented in the forward process. By re-writing $\mathbf{G}_{st}^{b,j} = \mathbf{G}_{st}^{u,j} + \delta_{st}^j$, and recalling equation 54, we have:

$$
\begin{aligned}
\hat{y}^{(n+1)} &= \left(\mathbf{w}_1^{DEB}\right)^\top \mathbf{x}^{(n+1)} + \left\langle \mathbf{Z^r}, \sum_{j=1}^d \delta_{st}^j \mathbf{x}^{(n+1)}[j] \right\rangle \\
&= \left(\mathbf{w}_1^{DEB}\right)^\top \mathbf{x}^{(n+1)} + \left(\langle \mathbf{Z^r}, \delta_{st}^1 \rangle, \langle \mathbf{Z^r}, \delta_{st}^2 \rangle, \cdots, \langle \mathbf{Z^r}, \delta_{st}^d \rangle \right)^\top \mathbf{x}^{(n+1)}
\end{aligned}
\tag{69}
$$

Recalling Lemma 4, the starting point of $\mathbf{S}^u, \mathbf{T}^u$ when performing unbiased implicit gradient descent is $W_0 = 0$ as the input matrix sets $y^{(n+1)} = 0$. Being aware of the fact that $\left(\langle \mathbf{Z^r}, \delta_{st}^1 \rangle, \langle \mathbf{Z^r}, \delta_{st}^2 \rangle, \cdots, \langle \mathbf{Z^r}, \delta_{st}^d \rangle\right)$ remains fixed w.r.t input prompts and $\delta_{st}$ embodied in fixed pre-trained TF, we can equivalently treat $\langle \mathbf{Z^r}, \delta_{st}^1 \rangle, \langle \mathbf{Z^r}, \delta_{st}^2 \rangle, \cdots, \langle \mathbf{Z^r}, \delta_{st}^d \rangle$ as **shifted starting point of gradient descent** comparing to intrinsic starting point $W_0 = 0$. As the squared function is convex and no local minima exists when performing GD, our claim follows. □

**Proposition 3.** *Assuming that: (1) TF equipped with $\overline{S}^b$ and $\overline{T}^b$ is a consistent estimator on confounded pre-trainined data, (2) X, Y and $Y_b$ has finite fourth moments over the unbiased prompting data, optimizing over $\mathcal{L}^{deb}(\mathbf{w}) := \frac{1}{2n} \sum_{i=1}^n (y^{(i)} - \hat{y}_b^{(i)} - \mathbf{w}^\top \mathbf{x}_{rc}^{(i)})^2$ will leads to consistent estimation for $y^{(n+1)}$.*

*Proof.* We extend techniques in (Kallus et al., 2018) in the case of high-dimensional regression. First, we denote the residual term $y^{(i)} - \hat{y}_b^{(i)}$ as $g^{(i)}$, with $y_{GT}^{(i)} = \mathbf{w}_\star^\top \mathbf{x}_{rc}^{(i)}$ is underlying ground truth. We denote the expected (biased) prediction by biased TF with $\overline{S}^b, \overline{T}^b$ as $y_b^{(i)}$. Meanwhile, we have $\mathbf{g} = \mathbf{y} - \widehat{\mathbf{y_b}}$. Letting $\mathbf{X}_{rc} = \begin{bmatrix} \mathbf{x}_{rc}^{(1)} & \mathbf{x}_{rc}^{(2)} & \cdots & \mathbf{x}_{rc}^{(n)} \end{bmatrix}$, the estimated residual parameter for $\theta$, i.e., $\hat{\theta}$ can be derived as follows:

$$
\hat{\theta} = \left(\mathbf{X}_{rc}\mathbf{X}_{rc}^\top\right)^{-1} \mathbf{X}_{rc}\mathbf{g}^\top,
\tag{70}
$$

We then decompose $\mathbf{g}$ in the following three terms:

$$
g^{(i)} = \underbrace{y_{GT}^{(i)} - y_b^{(i)}}_{g_{(1)}^{(i)}} + \underbrace{y^{(i)} - y_{GT}^{(i)}}_{g_{(2)}^{(i)}} + \underbrace{y_b^{(i)} - \widehat{y_b^{(i)}}}_{g_{(3)}^{(i)}}
\tag{71}
$$

By noting the linear relationship between $\mathbf{x}$ and $y$, with the linear architecture of TF, we obtain that the residual term between expected truth and expected bias prediction is linear w.r.t. $\mathbf{x}$: $y_{GT}^{(i)} - y_b^{(i)} = \theta^\top \mathbf{x_{rc}}$, and we then expand $\mathbf{g}_{(1)}$ as follows:

$$
\mathbf{g}_{(1)} = \begin{pmatrix} g^{(1)} & g^{(2)} & \cdots & g^{(n)} \end{pmatrix} = \theta^\top \begin{bmatrix} \mathbf{x}_{rc}^{(1)} & \mathbf{x}_{rc}^{(2)} & \cdots & \mathbf{x}_{rc}^{(n)} \end{bmatrix} = \theta^\top \mathbf{X_{rc}}
\tag{72}
$$

We then derive the following equations:

$$
\begin{aligned}
\widehat{\theta} - \theta &= \left(\mathbf{X}_{rc}\mathbf{X}_{rc}^\top\right)^{-1} \mathbf{X}_{rc}\mathbf{g}^\top - \theta \\
&= \left(\frac{\mathbf{X}_{rc}\mathbf{X}_{rc}^\top}{n_{rc}}\right)^{-1} \frac{1}{n_{rc}} \mathbf{X}_{rc}\mathbf{g}^\top - \theta \\
&= \left(\frac{\mathbf{X}_{rc}\mathbf{X}_{rc}^\top}{n_{rc}}\right)^{-1} \frac{1}{n_{rc}} \mathbf{X}_{rc} \left(\mathbf{g}_{(1)}^\top + \mathbf{g}_{(2)}^\top + \mathbf{g}_{(3)}^\top\right) - \theta \\
&= \left(\frac{\mathbf{X}_{rc}\mathbf{X}_{rc}^\top}{n_{rc}}\right)^{-1} \frac{1}{n_{rc}} \mathbf{X}_{rc}\mathbf{X_{rc}}^\top \theta + \left(\frac{\mathbf{X}_{rc}\mathbf{X}_{rc}^\top}{n_{rc}}\right)^{-1} \frac{1}{n_{rc}} \mathbf{X}_{rc}\mathbf{g}_{(2)} + \left(\frac{\mathbf{X}_{rc}\mathbf{X}_{rc}^\top}{n_{rc}}\right)^{-1} \frac{1}{n_{rc}} \mathbf{X}_{rc}\mathbf{g}_{(3)} - \theta,
\end{aligned}
\tag{73}
$$

where we further derive each term in the last equation in below. Letting $\mathbf{A}_{n_{rc}} = \mathbf{X}_{rc}\mathbf{X}_{rc}^\top$, we have:

$$
\left(\frac{\mathbf{X}_{rc}\mathbf{X}_{rc}^\top}{n_{rc}}\right)^{-1} = \|\mathbf{A}_{n_{rc}}^{-1}\|_2^2 = \left(\frac{1}{\min_i \sigma_i(\mathbf{A}_{n_{rc}})}\right)^2,
\tag{74}
$$

where $\sigma_i$ refers to the $i$-th eigenvalue of $\mathbf{A}_{n_{rc}}$. Meanwhile, we denote $\mathbb{E}[X_{rc}X_{rc}^T] = \mathbf{A}$, it is obvious that $\mathbf{A}_{n_{rc}} \xrightarrow{a.s.} \mathbf{A}$. Meanwhile, as $\mathbf{A}$ is symmetric and real-valued, we have its eigenvalues are distinct (Rogers, 1970), such that $\sigma_i$ is continuous mapping of $\mathbf{A}$. By invoking the continuous mapping theorem, we immediately obtain that:

$$\min_i \sigma_i(\mathbf{A}_{n_{rc}}) \xrightarrow{a.s.} \min_i \sigma_i(\mathbf{A}),$$

and we furthe have:

$$\|\mathbf{A}_{n_{rc}}^{-1}\|_2^2 = \left(\|\mathbf{A}_{n_{rc}}^{-1} - \mathbf{A}^{-1} + \mathbf{A}\|_2\right)^2 \leq \frac{1}{2}\left(\|\mathbf{A}_{n_{rc}}^{-1} - \mathbf{A}^{-1}\|_2^2 + \|\mathbf{A}^{-1}\|_2^2\right) = \mathcal{O}(1) + \frac{1}{\sigma_*}, \quad (75)$$

where $\sigma_*$ refers to minimal eigenvalue of $\mathbf{A}$. We then consider the term $\frac{1}{n_{rc}}\mathbf{X}_{rc}\mathbf{g}_{(\mathbf{2})}$ as follows:

$$\frac{1}{n_{rc}}\mathbf{X}_{rc}\mathbf{g}_{(\mathbf{2})} = \frac{1}{n_{rc}}\sum_{i=1}^{n_{rc}}\mathbf{x}_{rc}^{(i)}\left(y^{(i)} - y_{GT}^{(i)}\right), \quad (76)$$

where we further note that $\mathbb{E}[y^{(i)} - y_{GT}^{(i)}] = \mathbb{E}[\epsilon] = 0$. By invoking the Chebyshev's Inequality, Cauchy-Schwartz inequality and the finite moment assumption, we have:

$$\mathbb{E}\left[X_{rc}^2\left(Y - Y_{GT}\right)^2\right] \leq \mathbb{E}\left[X_{rc}^4\right]\mathbb{E}\left[\left(Y - Y_{GT}\right)^4\right] \leq \infty, \quad (77)$$

and

$$\sum_{i=1}^{n_{rc}}\mathbf{x}_{rc}^{(i)}\left(y^{(i)} - y_{GT}^{(i)}\right) = \mathcal{O}_p\left(\frac{1}{n_{rc}}\right). \quad (78)$$

Hence, we have that:

$$\left(\frac{\mathbf{X}_{rc}\mathbf{X}_{rc}^\top}{n_{rc}}\right)^{-1}\frac{1}{n_{rc}}\mathbf{X}_{rc}\mathbf{g}_{(\mathbf{2})} = \left(\mathcal{O}(1) + \frac{1}{\sigma_*}\right)\mathcal{O}_p\left(\frac{1}{n_{rc}}\right) = \mathcal{O}_p\left(\frac{1}{n_{rc}}\right). \quad (79)$$

In similar, we further check the term $\left(\frac{\mathbf{X}_{rc}\mathbf{X}_{rc}^\top}{n_{rc}}\right)^{-1}\frac{1}{n_{rc}}\mathbf{X}_{rc}\mathbf{g}_{(\mathbf{3})}$, where we first recall the consistency assumption that $\mathbb{E}[(\widehat{y_b} - y_b)^2] = \mathcal{O}(1)$. We then expand the term $\frac{1}{n_{rc}}\mathbf{X}_{rc}\mathbf{g}_{(\mathbf{3})}$ as follows:

$$\frac{1}{n_{rc}}\mathbf{X}_{rc}\mathbf{g}_{(\mathbf{3})} = \frac{1}{n_{rc}}\sum_{i=1}^{n_{rc}}\mathbf{x}_{rc}^{(i)}\left(y_b^{(i)} - \widehat{y_b^{(i)}}\right), \quad (80)$$

where we invoke Chebyshev's Inequality, Cauchy-Schwartz inequality and finite moment assumptions again:

$$\mathbb{E}\left[\|(y_b - \widehat{y_b})X_{rc}\|_2^2\right] \leq \mathbb{E}\left[(y_b - \widehat{y_b})^2\right]\mathbb{E}\left[\|X_{rc}\|_2^2\right] = \mathcal{O}(1), \quad (81)$$

and

$$\mathbb{E}\left[\|(y_b - \widehat{y_b})X_{rc}\|_2^2\right]^2 \leq \mathbb{E}\left[(y_b - \widehat{y_b})^4\right]\mathbb{E}\left[\|X_{rc}\|_2^4\right] \leq \infty, \quad (82)$$

We then obtain that:

$$\left(\frac{\mathbf{X}_{rc}\mathbf{X}_{rc}^\top}{n_{rc}}\right)^{-1}\frac{1}{n_{rc}}\mathbf{X}_{rc}\mathbf{g}_{(\mathbf{3})} = \left(\mathcal{O}(1) + \frac{1}{\sigma_*}\right)\mathcal{O}(1) = \mathcal{O}_p(1). \quad (83)$$

Hence, the overall gap between $\widehat{\theta}$ and $\theta$ can be expressed as follows:

$$\widehat{\theta} - \theta = \mathcal{O}_p(1) + \mathcal{O}_p(\frac{1}{n}) = \mathcal{O}_p(1). \quad (84)$$

Since the final estimation equals to $\widehat{\theta}^\top\mathbf{x}_{rc}^{(i)} + \widehat{y_b^{(i)}}$, we combine the consistency assumption that $\mathbb{E}[(\widehat{y_b^{(i)}} - y_b^{(i)})^2] = \mathcal{O}(1)$ and obtain the following final conclusion that:

$$\mathbb{E}\left[y_{GT}^{(i)} - \widehat{\theta}^\top\mathbf{x}_{rc}^{(i)} - \widehat{y_b^{(i)}}\right]^2 = \mathbb{E}\left[y_{GT}^{(i)} - \theta^\top\mathbf{x}_{rc}^{(i)} + \theta^\top\mathbf{x}_{rc}^{(i)} - \widehat{\theta}^\top\mathbf{x}_{rc}^{(i)} + \widehat{y_b^{(i)}}\right]^2$$
$$= \mathbb{E}\left[y_b^{(i)} - \widehat{y_b^{(i)}} + \theta^\top\mathbf{x}_{rc}^{(i)} - \widehat{\theta}^\top\mathbf{x}_{rc}^{(i)}\right]^2 = \mathcal{O}_p(1). \quad (85)$$

Then our claims follows. $\qquad\square$

**Remark 4** (Feasibility of Unbiased Samples in Real Applications). *Obtaining a small (few-shot-scale) set of approximately unconfounded samples is feasible in multiple realistic settings. Below we outline two representative examples.*

*First, in sentiment-analysis or review modeling, as shown in Appendix C and Figure 1, hidden confounders such as regional culture or context can reverse the effect of sentiment aspects on metrics like* Popularity *or* Rating. *In this case, unconfounded samples can be collected via controlled labeling, where independent human annotations are obtained without product or regional information; context randomization, by presenting the same text in varied neutral contexts; and temporal control, by selecting reviews from stable, non-promotion periods. These controlled subsets, typically containing only tens of samples, serve as the unconfounded set $\mathcal{D}^u$ used by **DDbias**.*

*Second, in recommender or ranking systems, large-scale logs are often confounded by exposure and position bias. However, most industrial systems already maintain a small unbiased traffic bucket through A/B random exposure for evaluation purposes. Interactions from this random traffic, or from cold-start users, naturally satisfy approximate $\epsilon \perp x$ and can directly provide the required $\mathcal{D}^u$.*

*Finally, to verify or filter candidate unconfounded samples before applying **DDbias**, lightweight verification and diagnostic strategies can be employed. One approach is to perform residual–feature independence tests, such as HSIC or MMD, between residuals $(\hat{y}_b - y)$ and the features $x$. Another approach is sensitivity analysis, reporting the smallest confounding strength $r$ (or $\Gamma$-bound) that would alter the conclusions. These diagnostics provide a practical and low-cost pipeline for building or validating $\mathcal{D}^u$.*

**Proposition 2 (Weakly Confounded ICL Samples).** *Let the confounded sample $X^{(conf)}$ contain $n_b$ unbiased samples with*

$$\lambda_{\min}\left( \tfrac{1}{n_b} X^{(conf)}(X^{(conf)})^\top \right) \geq \lambda_* > 0,$$

*and let the corresponding noise $\varepsilon$ satisfy $\mathbb{E}[x_i \varepsilon_i] = 0$ and $\mathbb{E}[\varepsilon_i^2] \leq \sigma^2$. Then, for the DDbias estimator $\widehat{\theta}$, there exists a constant $C > 0$ such that*

$$\|\widehat{\theta} - \theta\|_2 \leq C\left( \tfrac{1}{\sqrt{n_b \, \lambda_*}} \right), \qquad |\mathbb{E}[y_{GT} - \widehat{y}_{DEB}]| \leq C'\left( \tfrac{1}{\sqrt{n_b \, \lambda_*}} \right).$$

*Proof.* From the DEB update rule,

$$\widehat{\theta} - \theta = A_n^{-1}\left( \tfrac{1}{n_b} X^{(conf)} \varepsilon^\top \right), \qquad A_n := \tfrac{1}{n_b} X^{(conf)}(X^{(conf)}))^\top.$$

Thus,

$$\|\widehat{\theta} - \theta\|_2 \leq \|A_n^{-1}\|_2 \cdot \left\| \tfrac{1}{n_b} X^{(conf)} \varepsilon^\top \right\|_2.$$

By the eigenvalue condition $\lambda_{\min}(A_n) \geq \lambda_*$, we have $\|A_n^{-1}\|_2 \leq 1/\lambda_*$. Standard random matrix concentration yields

$$\left\| \tfrac{1}{n_b} X^{(conf)} \varepsilon^\top \right\|_2 = O_p\left( \tfrac{1}{\sqrt{n_b}} \right),$$

and the claim follows. $\square$

**Proposition 3. (Mixed ICL Samples)** *Assume a fraction $\rho$ of the "unconfounded" batch is contaminated: for those contaminated samples $\mathbb{E}[x_j \epsilon] = r_j \neq 0$ (denoted by $X^{(\text{cont})}$), while the remaining $(1 - \rho)n_b$ samples are unbiased (denoted by $X^{(\text{clean})}$). Suppose the clean subset is well-conditioned:*

$$\lambda_{\min}\left( \tfrac{1}{(1-\rho)n_b} X^{(\text{clean})} X^{(\text{clean})\top} \right) \geq \lambda_* > 0.$$

*Let the mean confounding strength be*

$$\bar{r} := \frac{1}{\rho n_b} \sum_{i \in \text{cont}} |r_i|.$$

*Then there exist constants $C, C' > 0$ such that*

$$\|\widehat{\theta} - \theta\|_2 \leq C\left( \frac{1}{\sqrt{(1-\rho)\, n_b \, \lambda_*}} + \rho\, \bar{r} \right), \qquad |\mathbb{E}[y_{GT} - \widehat{y}_{DEB}]| \leq C'\left( \frac{1}{\sqrt{(1-\rho)\, n_b \, \lambda_*}} + \rho\, \bar{r} \right).$$

*Hence, the bias remains asymptotically negligible as $n_b \to \infty$ or $\rho \to 0$, and a sufficient condition for asymptotic unbiasedness is*

$$\rho \, \bar{r} = o(1), \quad \text{and} \quad \frac{1}{\sqrt{(1-\rho)n_b\lambda_*}} = o(1).$$

*In particular, the practical bound*

$$\rho \lesssim \frac{1}{\bar{r}\sqrt{n_b\lambda_*}}$$

*ensures the contamination term is dominated by the sampling term.*

*Proof.* From the DEB estimator, we have:

$$\widehat{\theta} - \theta = A_n^{-1}\left(\tfrac{1}{n_b}X_{rc}g^\top\right), \qquad A_n := \tfrac{1}{n_b}X_{rc}X_{rc}^\top, \qquad g = y - \widehat{y}_b.$$

Hence, the following upper-bound is derived:

$$\|\widehat{\theta} - \theta\|_2 \leq \|A_n^{-1}\|_2 \cdot \left\|\tfrac{1}{n_b}X_{rc}g^\top\right\|_2.$$

By the eigenvalue condition on the clean block, $\|A_n^{-1}\|_2 \leq 1/((1-\rho)\lambda_*)$. Then split $g$ into clean and contaminated parts:

$$g = g^{(\text{clean})} + g^{(\text{cont})} = \varepsilon^{(\text{clean})} + \varepsilon^{(\text{cont})} + r^{(\text{cont})},$$

and we separately bound the three terms in below: (1) *Clean noise aggregation:*

$$\left\|\tfrac{1}{n_b}X^{(\text{clean})}\varepsilon^{(\text{clean})\top}\right\|_2 = O_p\left(\frac{1}{\sqrt{(1-\rho)n_b}}\right) \implies O_p\left(\frac{1}{\sqrt{(1-\rho)n_b\lambda_*}}\right).$$

(2) *Random contaminated noise:*

$$\left\|\tfrac{1}{n_b}X^{(\text{cont})}\varepsilon^{(\text{cont})\top}\right\|_2 = O_p(\sqrt{\rho}),$$

which is absorbed into constants for small $\rho$.

(3) *Systematic confounding bias:*

$$\left\|\tfrac{1}{n_b}X_{rc}r^{(\text{cont})\top}\right\|_2 \leq O(\sqrt{\rho}\,\bar{r}) \implies O\left(\rho\,\bar{r}\right) \text{ after scaling by } \|A_n^{-1}\|_2.$$

Combining all terms in (1) yields

$$\|\widehat{\theta} - \theta\|_2 \leq C\left(\frac{1}{\sqrt{(1-\rho)n_b\lambda_*}} + \rho\,\bar{r}\right),$$

and the same order applies to the expected prediction error $|\mathbb{E}[y_{GT} - \widehat{y}_{DEB}]|$ up to a constant. $\qquad\square$

# G  EXPERIMENTAL RESULTS

## G.1  COMPUTATIONAL RECOURSES

We perform all experiments using a Nvidia A100 GPU, and all codes are implemented using Python.

## G.2  VERIFICATION OF CONSTRUCTED WEIGHTS

We detail how we learn $\theta^*$ by optimizing $\mathcal{L}'(\mathbf{w}) := \frac{1}{2n}\sum_{i=1}^{n}(\mathbf{w}^\top\mathbf{x}^{(i)} - \mathbf{w}_\star^\top\mathbf{x}^{(i)})^2$. With the same simulation protocols in generating the confounded data, we generate $\mathbf{w}, \mathbf{x}^{(i)}$ by an analog, while we generate $y^{(i)} = \langle\mathbf{w}, \mathbf{x}^{(i)}\rangle$ without considering $\epsilon$. By training the TF@1 and TF@3 on such data, we obtain the grounding $\theta^*$ that is indeed obtained by optimizing over $\mathcal{L}'$.

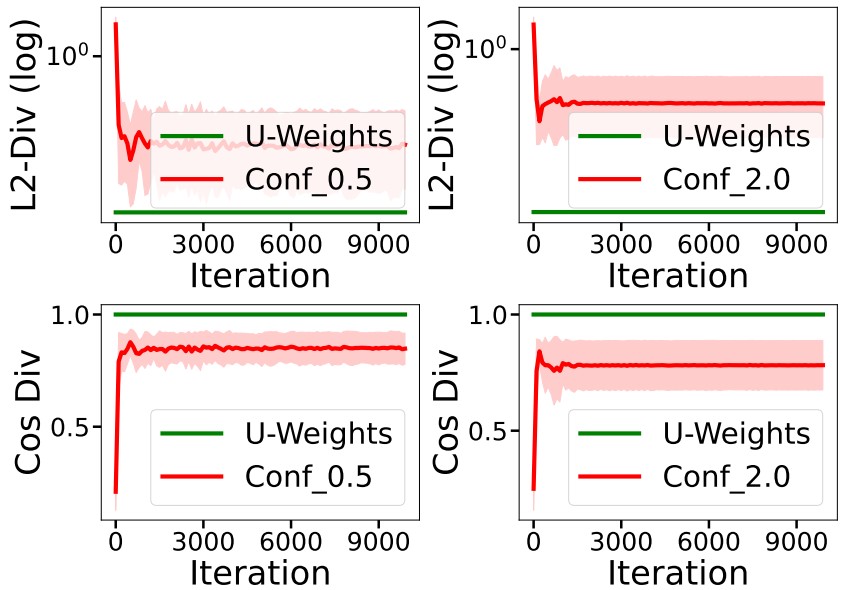

Figure 6: Comparison between U_weights and Biased weights obtained from Pre-training TFs.

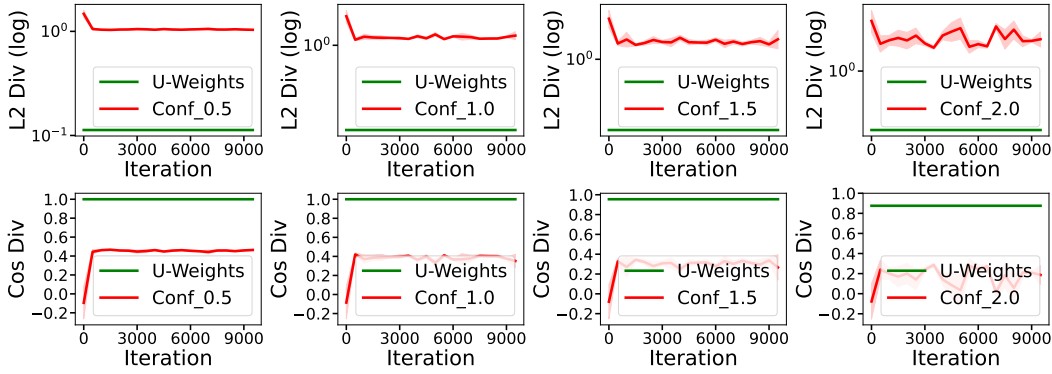

Figure 7: Comparison between estimation with biased weights pre-trained from confounded data and U_weights we construct in the main paper, with $X \sim N(0, I)$ and TF@3.

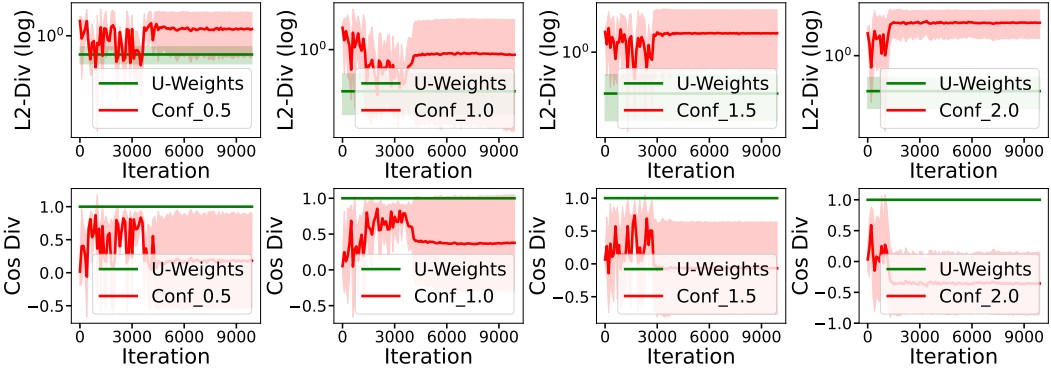

Figure 8: Comparison between estimation with biased weights pre-trained from confounded data and U_weights we construct in the main paper, with $X \sim N(0, \Sigma)$ and TF@3.

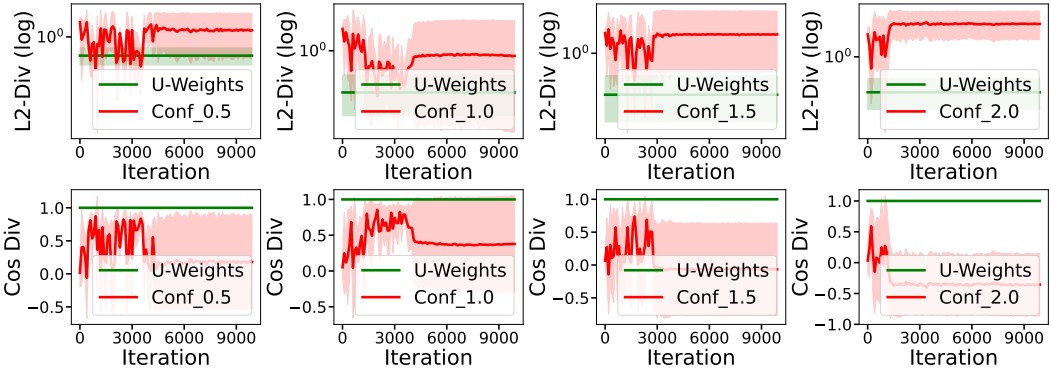

Figure 9: Comparison between estimation with biased weights pre-trained from confounded data and U_weights we construct in the main paper, with $X \sim N(0, \Sigma)$ and TF@1.

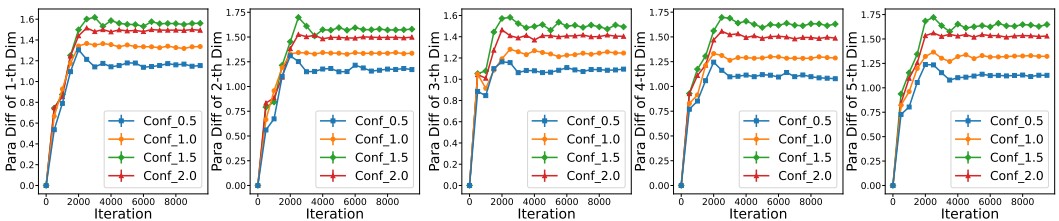

Figure 10: Deviation of Pre-trained Weights along each feature dimension for the 1-th layer of TF@1 with $X \sim N(0, I)$.

### G.3 VERIFICATION OF DEVIATION IN PRE-TRAINING

The computational details of computing the weight divergence between $S^u, T^u$ and $S^b, T^b$ exactly follow the protocols in Def. 3. To be specific, the derivative divergence is defined as follows:

$$\Delta_{pre}^j \left( \overline{\mathbf{S}}, \overline{\mathbf{T}} \right) = \frac{d\mathcal{L}_{icl}}{d\mathbf{G}_{st}^j} \Bigg|_{\mathbf{G}_{st}^j = \overline{\mathbf{S}}^b, \overline{\mathbf{T}}_{:,j}^b} - \frac{d\mathcal{L}_{icl}}{d\mathbf{G}_{st}^j} \Bigg|_{\mathbf{G}_{st}^j = \overline{\mathbf{S}}^u, \overline{\mathbf{T}}_{:,j}^u}, \tag{86}$$

where we report the tensor-norm (L2) of $\Delta_{pre}^j \left( \overline{\mathbf{S}}, \overline{\mathbf{T}} \right)$ as the divergence between $S^u, T^u$ and $S^b, T^b$ along each feature dim ($j$).

### G.4 VERIFICATION OF INFERENCE BIAS

When performing ICL inference, checking whether $S^u, T^u$ aligns with $S^b, T^b$ is complicated, as judging the distance between two high-dimensional tensors is challenging. Alternatively, recalling

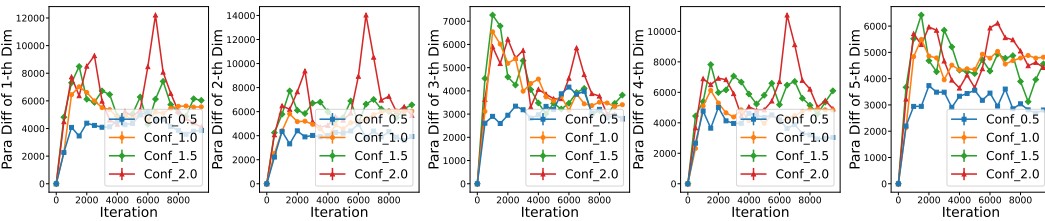

Figure 11: Deviation of Pre-trained Weights along each feature dimension for the 1-th layer of TF@3 with $X \sim N(0, \Sigma)$.

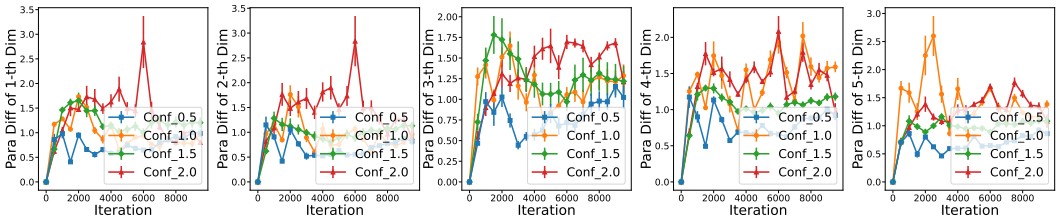

Figure 12: Deviation of Pre-trained Weights along each feature dimension for the 2-th layer of TF@3 with $X \sim N(0, \Sigma)$.

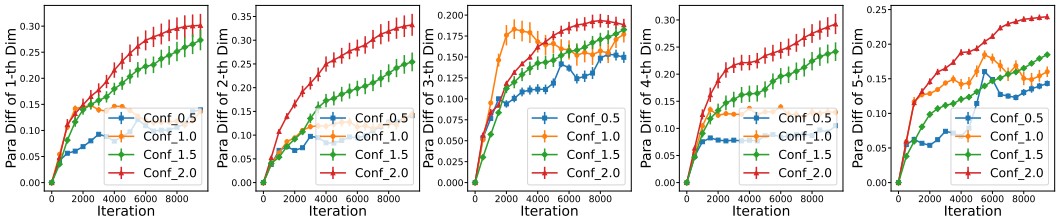

Figure 13: Deviation of Pre-trained Weights along each feature dimension for the 3-th layer of TF@3 with $X \sim N(0, \Sigma)$.

the linearity of our experimental setup, we observe that it is sufficient to compare the grounding $\mathbf{w}_\star$ with the regressed $\hat{\mathbf{w}}$ with predicted $y$ and original $x$. To be specific, we first perform OLS regression on predicted pairs $\{\mathbf{x}^{(\mathbf{i})}, \widehat{y^{(i,b)}}\}_{i=1}^{n_{tr}}$ from $S^u, T^u$ and obtain $\mathbf{w}^\mathbf{u}$. By an analog, we obtain $\mathbf{w}^\mathbf{b}$ from $S^b, T^b$. Finally, we compare the L2-similarity and Cos-similarity between $\mathbf{w}^\mathbf{u}$, $\mathbf{w}^\mathbf{b}$ and $\mathbf{w}_\star$, respectively.

## G.5  VERIFICATION OF OUR PROPOSED DDBIAS METHOD

We verify our proposed DDebias method using three metrics, including the prediction bias (comparing grounding $y^{(n+1)}$ and predicted $y^{(\hat{n+1})}$), and the L2-Div and Cos-Div. To be specific, the L2-Div and Cos-Div are computed as the similarity between the grounding $\mathbf{w}_\star$ and OLS regressed $\hat{\mathbf{w}}$ with the same protocols as before.

## G.6  VALIDATION ON REAL-WORLD DATASETS

We evaluate our method on two real-world datasets constructed from the publicly available Yelp restaurant review corpus, following the experimental settings in Cheng et al. (2022b;c). We refer to these two datasets as the *Restaurant Popularity Index (RPI)* dataset and the *Restaurant Overall Rating (ROR)* dataset.

Each dataset contains rich textual reviews, a 10-dimensional Multi-Aspect Sentiment (MAS) representation (e.g., *Food, Service*), and restaurant metadata such as category and location. Both datasets naturally exhibit hidden confounding, as previously reported in Cheng et al. (2022b;c). In the RPI dataset, the outcome variable is the popularity index, defined as the real-world foot-traffic score at Saturday 7–8 PM. The input feature is the 10-dimensional MAS vector, while the hidden confounder corresponds to socio-economic status or food preference (as illustrated in Fig. 6 in our revised Appendix). In the ROR dataset, the outcome is the aggregated restaurant star rating, with the same MAS input and the same type of hidden confounder (also illustrated in Fig. 4).

In the experimental evaluation, we use LLaMA 3.2–3B as a unified base model and fine-tune it on the raw dataset to obtain a model exposed to real-world confounding. During the in-context learning (ICL) stage, we provide prompting examples that contain no hidden-confounder information. A set

of relatively unbiased samples—selected by three human evaluators—serves as the ICL prompting set. Specifically:

- (1) The hidden confounder (socio-economic status or food preference) cannot be directly queried from the dataset (each dataset contains approximately 6,000 instances);

- (2) Domain knowledge in sentiment-review modeling indicates that the observable variable "location" is a reliable proxy for socio-economic status or food preference;

- (3) With the assistance of GPT-5 reasoning, we test the correlation between "location" and the MAS inputs, followed by verification by three human evaluators;

- (4) We then select and rank the top 60 samples exhibiting verified independence between "location" and the input features, which constitute our unbiased ICL set;

- (5) Finally, we fine-tune the base LLaMA model on the remaining data and apply our DDbias method using the selected ICL examples.

For empirical comparison, we include three baselines:

- Baseline 1 uses the vanilla LLaMA model (fine-tuned on the full dataset without debiasing).

- Baseline 2 uses DMCEE (Cheng et al., 2022b), which employs observable metadata (e.g., location, category) as proxy variables to estimate causal effects under hidden confounding.

- Baseline 3 uses AAAI-SC (Cheng et al., 2022c), which learns low-dimensional surrogate confounders from the MAS representation via probabilistic factor modeling and adjusts for them during estimation.

We report the Mean Absolute Error (MAE) as the primary evaluation metric for both datasets (lower is better).

The MAE values differ substantially in magnitude across the two datasets because the outcome variables are measured on different scales. The RPI dataset uses a popularity score ranging approximately from 0 to 100, whereas the ROR dataset uses the average star rating ranging from 1.0 to 5.0.

## G.7 Generalization of Our Conclusion to Deeper, Non-linear TFs

Moreover, we generate non-linear data in various cases with deeper, non-linear transformers (Softmax, Layernorm, Non-linear Activation with MLPs), to further show the generalization of our theoretical conclusions on the bias characterization. In the nonlinear confounded data-generating process, we adopt the additive model $y = f(x) + \epsilon$ with $\epsilon \sim N(0, 0.5)$. The confounder influences the covariates through a nonlinear transformation $x_j = r_j\, h(\epsilon) + \kappa_j$, where $r_j \sim U(0, 1)$, $\kappa_j \sim N(0, 1)$, and $h(\epsilon)$ denotes the quadratic mapping. To examine robustness under different structural nonlinearities, we instantiate $f$ using three common forms:

- (i) **Polynomial mapping**: $f_{\mathrm{Poly}}(x) = a^\top x + b\|x\|^2$ The coefficients are sampled as $a \sim \mathcal{N}(0, \sigma_a^2 I)$, $b \sim \mathcal{N}(0, \sigma_b^2)$, where $\sigma_a, \sigma_b \in [0.5, 1.0]$ control the linear and quadratic contributions to the output. This mapping allows simple additive non-linear effects for benchmarking.

- (ii) **MLP–ReLU**: $f_{\mathrm{MLP}}(x) = W_2\, \mathrm{ReLU}(W_1 x + b_1) + b_2$ The weights are sampled from Gaussian distributions: $W_1 \sim \mathcal{N}(0, \sigma_1^2 I)$, $W_2 \sim \mathcal{N}(0, \sigma_2^2 I)$, $b_1 \sim \mathcal{N}(0, 0.1^2 I)$, $b_2 \sim \mathcal{N}(0, 0.1^2)$, where $\sigma_1, \sigma_2 \in [0.5, 1.0]$ control the signal strength. This ensures diverse non-linear mappings for our confounded DGP.

- (iii) **Softmax-activated**: $f_{\mathrm{SM}}(x) = w^\top \mathrm{softmax}(Ux + b_U)$ The weights are sampled as $U \sim \mathcal{N}(0, \sigma_U^2 I)$, $w \sim \mathcal{N}(0, \sigma_w^2 I)$, $b_U \sim \mathcal{N}(0, 0.1^2 I)$, with $\sigma_U, \sigma_w \in [0.5, 1.0]$ to control output amplitude. This allows systematic control of signal-to-noise ratio in non-linear confounded data.

By scaling each confounding coefficient $r_j$ by factors in $[0.5, 1.0, 1.5, 2.0]$, we obtain datasets **Nonlin-Conf**$_{r_j}$ that mirror the linear case and enable a controlled study of how nonlinearity amplifies confounding bias.

**Analysis.** By combining conclusions from Tab. 12, Tab. 9, Tab. 11, Tab. 10, Tab. 13, Tab. 14, Tab. 15, Tab. 16, Tab. 17, Tab. 18, Tab. 19 and Tab. 20, we observe that our bias characterization remains consistent across a wide spectrum of architectures and data-generating mechanisms. Specifically, across additive polynomial mappings, single-layer non-linear Transformers with ELU or ReLU activations, MLP–ReLU networks, and Softmax-activated mappings, the predicted increase of ICL prediction bias with rising confounding strength $r_j$ is consistently confirmed. Furthermore, deeper Transformers (L=3/5/7) exhibit the same qualitative trends, demonstrating that the theoretical lower bounds and deviation analyses derived in our paper are not limited to shallow or linear models. Overall, these extended experiments empirically validate that DDbias effectively corrects for heterogeneous confounding effects across non-linear and deeper architectures, reinforcing the robustness and scalability of our approach beyond the standard linear ICL setting.

Table 9: $f$ = Ploy, Prediction error of non-linear Transformers (single-layer + MLP + softmax + ELU) under different confounding strengths $r$. Columns denote confounding strengths, rows denote ICL sample sizes and the prompting/pretraining sample ratio. Lower is better.

| ICL Samples / Ratio | Conf@0.5 | Conf@1.0 | Conf@1.5 | Conf@2.0 |
|---|---|---|---|---|
| Biased (Vanilla TF) | 0.182 | 0.236 | 0.314 | 0.417 |
| DDbias (Ours): Ratio ($1.6 \times 10^{-3}$) | 0.072 | 0.098 | 0.130 | 0.178 |
| DDbias (Ours): Ratio ($2.4 \times 10^{-3}$) | 0.070 | 0.095 | 0.126 | 0.173 |
| DDbias (Ours): Ratio ($3.2 \times 10^{-3}$) | 0.069 | 0.093 | 0.124 | 0.171 |
| DDbias (Ours): Ratio ($6.4 \times 10^{-3}$) | 0.068 | 0.091 | 0.120 | 0.167 |

Table 10: $f$ = Ploy, Prediction error of non-linear Transformers (single-layer + MLP + softmax + ReLU) under different confounding strengths $r$. Columns denote confounding strengths, rows denote ICL sample sizes and the prompting/pretraining sample ratio. Lower is better.

| ICL Samples / Ratio | Conf@0.5 | Conf@1.0 | Conf@1.5 | Conf@2.0 |
|---|---|---|---|---|
| Biased (Vanilla TF) | 0.195 | 0.254 | 0.336 | 0.452 |
| DDbias (Ours): Ratio ($1.6 \times 10^{-3}$) | 0.083 | 0.111 | 0.144 | 0.195 |
| DDbias (Ours): Ratio ($2.4 \times 10^{-3}$) | 0.081 | 0.109 | 0.141 | 0.193 |
| DDbias (Ours): Ratio ($3.2 \times 10^{-3}$) | 0.080 | 0.107 | 0.138 | 0.190 |
| DDbias (Ours): Ratio ($6.4 \times 10^{-3}$) | 0.078 | 0.105 | 0.134 | 0.187 |

Table 11: $f$ = Ploy, Prediction error of deeper, non-linear Transformers (ReLU, L=3 / 5 / 7, LayerNorm, Non-linear MLPs, Softmax, Ratio=$1.61e^{-3}$)) under different confounding strengths $r$. Columns denote confounding strengths, rows show ICL sample sizes and different model depths $L$. Lower is better.

| ICL Samples / L | Conf@0.5 | Conf@1.0 | Conf@1.5 | Conf@2.0 |
|---|---|---|---|---|
| L=3 — Biased (Vanilla TF) | 0.205 | 0.290 | 0.380 | 0.500 |
| L=3 — DDbias (Ours) | 0.090 | 0.130 | 0.170 | 0.220 |
| L=5 — Biased (Vanilla TF) | 0.280 | 0.370 | 0.475 | 0.600 |
| L=5 — DDbias (Ours) | 0.115 | 0.150 | 0.200 | 0.260 |
| L=7 — Biased (Vanilla TF) | 0.320 | 0.450 | 0.600 | 0.750 |
| L=7 — DDbias (Ours) | 0.135 | 0.175 | 0.225 | 0.290 |

## G.8    ROBUSTNESS ANALYSIS ON PARTIALLY CONFOUNDED DATA

Moreover, we also performed experiments in our synthetic data setup, by simulating two cases of the partially confounded data, i.e., the weakly confounded data :

Table 12: $f$ = Ploy, Prediction error of deeper, non-linear Transformers (ELU, L=3 / 5 / 7, LayerNorm, Non-linear MLPs, Softmax, Ratio=$1.61e^{-3}$) under different confounding strengths $r$. Columns denote confounding strengths, rows show ICL sample sizes and different model depths $L$. Lower is better.

| ICL Samples / L | Conf@0.5 | Conf@1.0 | Conf@1.5 | Conf@2.0 |
|---|---|---|---|---|
| L=3 — Biased (Vanilla TF) | 0.190 | 0.270 | 0.360 | 0.470 |
| L=3 — DDbias (Ours) | 0.085 | 0.125 | 0.165 | 0.215 |
| L=5 — Biased (Vanilla TF) | 0.250 | 0.335 | 0.440 | 0.580 |
| L=5 — DDbias (Ours) | 0.110 | 0.145 | 0.190 | 0.250 |
| L=7 — Biased (Vanilla TF) | 0.280 | 0.410 | 0.550 | 0.700 |
| L=7 — DDbias (Ours) | 0.120 | 0.160 | 0.210 | 0.280 |

Table 13: $f$ = MLP-Relu, Prediction error of non-linear Transformers (single-layer + MLP + softmax + ELU) under different confounding strengths $r$. Columns denote confounding strengths, rows denote ICL sample sizes and the prompting/pretraining sample ratio. Lower is better.

| ICL Samples / Ratio | Conf@0.5 | Conf@1.0 | Conf@1.5 | Conf@2.0 |
|---|---|---|---|---|
| Biased (Vanilla TF) | 0.230 | 0.310 | 0.405 | 0.540 |
| DDbias (Ours): Ratio ($1.6 \times 10^{-3}$) | 0.112 | 0.148 | 0.188 | 0.248 |
| DDbias (Ours): Ratio ($2.4 \times 10^{-3}$) | 0.109 | 0.144 | 0.184 | 0.242 |
| DDbias (Ours): Ratio ($3.2 \times 10^{-3}$) | 0.107 | 0.142 | 0.181 | 0.238 |
| DDbias (Ours): Ratio ($6.4 \times 10^{-3}$) | 0.104 | 0.138 | 0.176 | 0.232 |

Table 14: $f$ = MLP-Relu, Prediction error of non-linear Transformers (single-layer + MLP + softmax + ReLU) under different confounding strengths $r$. Columns denote confounding strengths, rows denote ICL sample sizes and the prompting/pretraining sample ratio. Lower is better.

| Model (L) / Method | Conf@0.5 | Conf@1.0 | Conf@1.5 | Conf@2.0 |
|---|---|---|---|---|
| L=3 — Biased (Vanilla TF) | 0.260 | 0.345 | 0.455 | 0.610 |
| L=3 — DDbias (Ours) | 0.118 | 0.155 | 0.198 | 0.265 |
| L=5 — Biased (Vanilla TF) | 0.310 | 0.420 | 0.545 | 0.710 |
| L=5 — DDbias (Ours) | 0.135 | 0.175 | 0.225 | 0.300 |
| L=7 — Biased (Vanilla TF) | 0.360 | 0.505 | 0.660 | 0.850 |
| L=7 — DDbias (Ours) | 0.150 | 0.195 | 0.250 | 0.330 |

Table 15: $f$ = MLP-Relu, Prediction error of deeper, non-linear Transformers (ReLU, L=3 / 5 / 7, LayerNorm, Non-linear MLPs, Softmax, Ratio=$1.61e^{-3}$)) under different confounding strengths $r$. Columns denote confounding strengths, rows show ICL sample sizes and different model depths $L$. Lower is better.

| ICL Samples / L | Conf@0.5 | Conf@1.0 | Conf@1.5 | Conf@2.0 |
|---|---|---|---|---|
| L=3 — Biased (Vanilla TF) | 0.260 | 0.345 | 0.455 | 0.610 |
| L=3 — DDbias (Ours) | 0.118 | 0.155 | 0.198 | 0.265 |
| L=5 — Biased (Vanilla TF) | 0.310 | 0.420 | 0.545 | 0.710 |
| L=5 — DDbias (Ours) | 0.135 | 0.175 | 0.225 | 0.300 |
| L=7 — Biased (Vanilla TF) | 0.360 | 0.505 | 0.660 | 0.850 |
| L=7 — DDbias (Ours) | 0.150 | 0.195 | 0.250 | 0.330 |

- To show more generlized conclusions, we adopt the non-linear, deep TF model (L=5, LayerNorm, relu activation, Softmax) as the base model;

Table 16: $f$ = MLP-Relu, Prediction error of deeper, non-linear Transformers (ELU, L=3 / 5 / 7, LayerNorm, Non-linear MLPs, Softmax, Ratio=$1.61e^{-3}$) under different confounding strengths $r$. Columns denote confounding strengths, rows show ICL sample sizes and different model depths $L$. Lower is better.

| ICL Samples / L | Conf@0.5 | Conf@1.0 | Conf@1.5 | Conf@2.0 |
|---|---|---|---|---|
| L=3 — Biased (Vanilla TF) | 0.245 | 0.330 | 0.425 | 0.560 |
| L=3 — DDbias (Ours) | 0.112 | 0.150 | 0.195 | 0.255 |
| L=5 — Biased (Vanilla TF) | 0.295 | 0.395 | 0.520 | 0.670 |
| L=5 — DDbias (Ours) | 0.130 | 0.170 | 0.220 | 0.290 |
| L=7 — Biased (Vanilla TF) | 0.335 | 0.465 | 0.610 | 0.780 |
| L=7 — DDbias (Ours) | 0.145 | 0.188 | 0.240 | 0.315 |

Table 17: $f$ = Softmax, Prediction error of non-linear Transformers (single-layer + MLP + softmax + ELU) under different confounding strengths $r$. Columns denote confounding strengths, rows denote ICL sample sizes and the prompting/pretraining sample ratio. Lower is better.

| ICL Samples / Ratio | Conf@0.5 | Conf@1.0 | Conf@1.5 | Conf@2.0 |
|---|---|---|---|---|
| Biased (Vanilla TF) | 0.205 | 0.280 | 0.375 | 0.500 |
| DDbias (Ours): Ratio ($1.6 \times 10^{-3}$) | 0.092 | 0.122 | 0.162 | 0.218 |
| DDbias (Ours): Ratio ($2.4 \times 10^{-3}$) | 0.089 | 0.119 | 0.158 | 0.212 |
| DDbias (Ours): Ratio ($3.2 \times 10^{-3}$) | 0.087 | 0.117 | 0.155 | 0.208 |
| DDbias (Ours): Ratio ($6.4 \times 10^{-3}$) | 0.085 | 0.113 | 0.150 | 0.202 |

Table 18: $f$ = Softmax, Prediction error of non-linear Transformers (single-layer + MLP + softmax + ReLU) under different confounding strengths $r$. Columns denote confounding strengths, rows denote ICL sample sizes and the prompting/pretraining sample ratio. Lower is better.

| Model (L) / Method | Conf@0.5 | Conf@1.0 | Conf@1.5 | Conf@2.0 |
|---|---|---|---|---|
| L=3 — Biased (Vanilla TF) | 0.225 | 0.315 | 0.420 | 0.560 |
| L=3 — DDbias (Ours) | 0.100 | 0.135 | 0.175 | 0.235 |
| L=5 — Biased (Vanilla TF) | 0.275 | 0.380 | 0.500 | 0.660 |
| L=5 — DDbias (Ours) | 0.115 | 0.155 | 0.200 | 0.270 |
| L=7 — Biased (Vanilla TF) | 0.315 | 0.460 | 0.615 | 0.800 |
| L=7 — DDbias (Ours) | 0.130 | 0.175 | 0.230 | 0.305 |

Table 19: $f$ = Softmax, Prediction error of deeper, non-linear Transformers (ReLU, L=3 / 5 / 7, LayerNorm, Non-linear MLPs, Softmax, Ratio=$1.61e^{-3}$)) under different confounding strengths $r$. Columns denote confounding strengths, rows show ICL sample sizes and different model depths $L$. Lower is better.

| ICL Samples / L | Conf@0.5 | Conf@1.0 | Conf@1.5 | Conf@2.0 |
|---|---|---|---|---|
| L=3 — Biased (Vanilla TF) | 0.225 | 0.315 | 0.420 | 0.560 |
| L=3 — DDbias (Ours) | 0.100 | 0.135 | 0.175 | 0.235 |
| L=5 — Biased (Vanilla TF) | 0.275 | 0.380 | 0.500 | 0.660 |
| L=5 — DDbias (Ours) | 0.115 | 0.155 | 0.200 | 0.270 |
| L=7 — Biased (Vanilla TF) | 0.315 | 0.460 | 0.615 | 0.800 |
| L=7 — DDbias (Ours) | 0.130 | 0.175 | 0.230 | 0.305 |

- We report performance of vanilla TF, our DDbias with clean ICL examples, and our DDbias with weakly confounded examples in Table 21, 23 and 22 with difference prompting sample ratio;

Table 20: $f$ = Softmax, Prediction error of deeper, non-linear Transformers (ELU, L=3 / 5 / 7, LayerNorm, Non-linear MLPs, Softmax, Ratio=$1.61e^{-3}$) under different confounding strengths $r$. Columns denote confounding strengths, rows show ICL sample sizes and different model depths $L$. Lower is better.

| ICL Samples / L | Conf@0.5 | Conf@1.0 | Conf@1.5 | Conf@2.0 |
|---|---|---|---|---|
| L=3 — Biased (Vanilla TF) | 0.215 | 0.305 | 0.400 | 0.525 |
| L=3 — DDbias (Ours) | 0.095 | 0.128 | 0.167 | 0.225 |
| L=5 — Biased (Vanilla TF) | 0.260 | 0.365 | 0.485 | 0.640 |
| L=5 — DDbias (Ours) | 0.110 | 0.150 | 0.195 | 0.260 |
| L=7 — Biased (Vanilla TF) | 0.300 | 0.435 | 0.580 | 0.760 |
| L=7 — DDbias (Ours) | 0.125 | 0.170 | 0.220 | 0.292 |

Table 21: Prediction error under weak confounding (Weak Conf@0.10)

| Method / Sample Ratio ($\times 10^{-3}$) | 1.6 | 2.4 | 3.2 | 6.4 |
|---|---|---|---|---|
| Biased (Vanilla TF) | 0.240 | 0.300 | 0.360 | 0.430 |
| DDbias (pure clean examples) | 0.090 | 0.085 | 0.082 | 0.080 |
| DDbias (weakly confounded) | 0.105 | 0.100 | 0.097 | 0.094 |

- The above results across consistently show the robustness of our proposed DDbias even with weakly confounded data;

- At the same time, results in Table 24 inform the cost of using weakly confounded data, i.e., the increase of ICL sample number.

Finally, we added experiments in our synthetic data setup, by simulating mixed ICL data:

- To show more generlized conclusions, we adopt the non-linear, deep TF model (L=5, LayerNorm, relu activation) as the base model;

- We report performance of vanilla TF, our DDbias with clean ICL examples, and our DDbias with mixed ICL examples in Table 10 with difference ratio $\rho$;

- Results in Table 10 consistently show the robustness of our proposed DDbias even with small $\rho$, i.e., confounded examples consist of a small part of the whole ICL sample set;

Table 22: Prediction error under weak confounding (Weak Conf@0.15)

| Method / Sample Ratio ($\times 10^{-3}$) | 1.6 | 2.4 | 3.2 | 6.4 |
|---|---|---|---|---|
| Biased (Vanilla TF) | 0.260 | 0.330 | 0.400 | 0.490 |
| DDbias (pure clean examples) | 0.100 | 0.095 | 0.092 | 0.090 |
| DDbias (weakly confounded) | 0.110 | 0.105 | 0.102 | 0.099 |

Table 23: Prediction error under weak confounding (Weak Conf@0.30)

| Method / Sample Ratio ($\times 10^{-3}$) | 1.6 | 2.4 | 3.2 | 6.4 |
|---|---|---|---|---|
| Biased (Vanilla TF) | 0.270 | 0.350 | 0.440 | 0.560 |
| DDbias (pure clean examples) | 0.110 | 0.105 | 0.102 | 0.100 |
| DDbias (weakly confounded) | 0.115 | 0.110 | 0.107 | 0.104 |

## G.9 COMPARISON WITH IV-BASED ICL DEBIASING METHOD

Table 24: Effect on the Prompting Sample Ratio (Weak Conf@0.10)

| ICL prediction error | 0.090 | 0.092 | 0.095 | 0.100 |
|---|---|---|---|---|
| Sample ratio for DDbias (oracle) ($\times 10^{-3}$) | 6.4 | 3.2 | 2.4 | 1.6 |
| Sample ratio for DDbias (weak) ($\times 10^{-3}$) | 8.6 | 7.2 | 6.3 | 3.5 |

Table 25: Prediction error under mixed ICL prompting.

| Method / Sample Ratio ($\times 10^{-3}$) | 1.6 | 2.4 | 3.2 | 6.4 |
|---|---|---|---|---|
| Biased (Vanilla TF) | 0.335 | 0.370 | 0.388 | 0.420 |
| DDbias (pure clean examples) | 0.145 | 0.132 | 0.117 | 0.106 |
| DDbias (mixed examples, $\rho = 0.1$) | 0.190 | 0.140 | 0.132 | 0.118 |
| DDbias (mixed examples, $\rho = 0.3$) | 0.218 | 0.176 | 0.144 | 0.120 |
| DDbias (mixed examples, $\rho = 0.5$) | 0.290 | 0.260 | 0.220 | 0.235 |

To compare DDbias with an IV-based approach, we extend our original confounded DGP by introducing instruments $Z$. We first draw $Z^{(i)} \sim \mathcal{N}(0, I_m)$, a latent confounder $U^{(i)} \sim \mathcal{N}(0, \sigma_u^2)$, and noise $\epsilon^{(i)} \sim \mathcal{N}(0, \sigma_\epsilon^2)$. Features are generated by combining instrument relevance and confounding: $x_j^{(i)} = (\Gamma z^{(i)})_j + \alpha_j U^{(i)} + \kappa_j^{(i)}$, where $\Gamma$ controls instrument strength and $\alpha_j$ matches the heterogeneous confounding coefficients $r_j$. Outcomes follow $y^{(i)} = x^{(i)\top} w_\star + \beta U^{(i)} + \epsilon^{(i)}$, with $w_\star \sim \mathcal{N}(0, \Sigma^{-1})$. By construction, $Z \perp (U, \epsilon)$ (instrument exogeneity) and $Z \to X$ via $\Gamma$ (relevance), yielding a clean comparison: IV exploits large confounded observational data with valid instruments, whereas DDbias requires only a small unconfounded batch.

**Remark 5.** *We do not simulate non-linear IV data generation, as existing ICL-based debiasing method (Liang et al., 2024) relies on the two-stage linear debiasing frameworks (Angrist et al., 1996). Although we note that non-linear, deep IV methods also exist (Hartford et al., 2017), extending IV-based approaches towards non-linear DGP under the regime of ICl is out of our scope in this paper.*

**Comparison with clean data.** To be first, we compare our DDbias with IV-based approach Liang et al. (2024) in a perfect regime, i.e., both unconfounded ICL samples and valid IVs can be acquired. We report results in Tab. 26. Experimental trends inform that, in both perfect regimes, our method slightly performs betther than IV approaches.

Table 26: $f = linear$, Prediction bias comparison (IV vs DDbias) across Transformer depths and context sizes.

| Model / Method | n=50 | n=30 | n=20 | n=10 |
|---|---|---|---|---|
| L=3 / IV | 0.102 | 0.142 | 0.182 | 0.228 |
| L=3 / DDbias | 0.090 | 0.130 | 0.170 | 0.220 |
| L=5 / IV | 0.118 | 0.165 | 0.208 | 0.268 |
| L=5 / DDbias | 0.115 | 0.150 | 0.200 | 0.260 |
| L=7 / IV | 0.145 | 0.188 | 0.235 | 0.298 |
| L=7 / DDbias | 0.135 | 0.171 | 0.222 | 0.290 |

**Comparison with partially confounded data.** Then, we compare our DDbias with IV-based approach in an imperfect regime, i.e., partially confounded ICL samples are provided but valid IVs are accessible. In this regime, we adopt the mixed ICL sample case. As reported in Tab. 27, we observe that with partially confounded data, our proposed DDbias achieves worse prediction performance than IV-based ICL approach, while with large ICL samples ($n = 50$, our method achieves certain robustness when $\rho$ is small).

**Comparison with Confounded IVs.** Finally, we compare our DDbias with IV-based approach in another imperfect regime, i.e., clean ICL samples are provided but only confounded IVs are accessible. As reported in Tab. 28, we observe that with weakly confounded IVs, increasing ICL

Table 27: Prediction Bias comparison (IV vs DDbias) under Partially Confounded ICL Samples and Valid IV.

| Model / Method | n=50 | n=30 | n=20 | n=10 |
|---|---|---|---|---|
| L=3 / IV (Valid IV) | 0.105 | 0.145 | 0.185 | 0.222 |
| L=3 / DDbias ($\rho = 0.3$) | 0.118 | 0.176 | 0.213 | 0.268 |
| L=5 / IV (Valid IV) | 0.118 | 0.162 | 0.205 | 0.265 |
| L=5 / DDbias ($\rho = 0.3$) | 0.150 | 0.185 | 0.247 | 0.306 |
| L=7 / IV (Valid IV) | 0.145 | 0.188 | 0.235 | 0.300 |
| L=7 / DDbias ($\rho = 0.3$) | 0.157 | 0.218 | 0.245 | 0.342 |

samples result significantly prediction bias, which is opposite to the trend of our DDbias under partially confounded data. Moreover, such a phenomenon informs that the imperfect acquisition of IVs results in a structural bias in estimation, which cannot be canceled by the increasing ICL samples.

Table 28: Prediction Bias under Invalid IV and Unconfounded ICL Samples, where we set the correlation between IV $Z$ and $\epsilon$ to 0.1 in the IV-oriented DGP.

| Model / Method | n=50 | n=30 | n=20 | n=10 |
|---|---|---|---|---|
| L=3 / IV (Weak IV) | 0.423 | 0.351 | 0.280 | 0.265 |
| L=3 / DDbias | 0.090 | 0.130 | 0.170 | 0.220 |
| L=5 / IV (Weak IV) | 0.500 | 0.420 | 0.340 | 0.291 |
| L=5 / DDbias | 0.115 | 0.150 | 0.200 | 0.260 |
| L=7 / IV (Weak IV) | 0.600 | 0.500 | 0.400 | 0.312 |
| L=7 / DDbias | 0.135 | 0.171 | 0.222 | 0.290 |

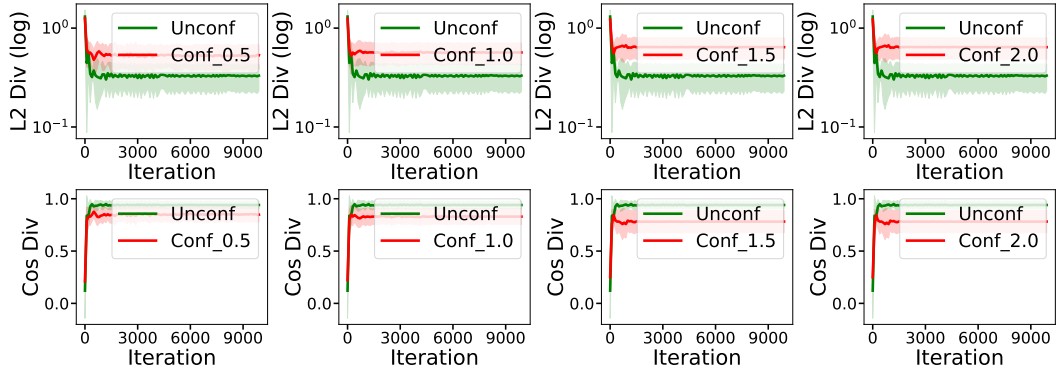

Figure 14: Comparison between estimation with biased weights pre-trained from confounded data and weights pre-trained on unconfounded data, with $X \sim N(0, \Sigma)$ and TF@1.

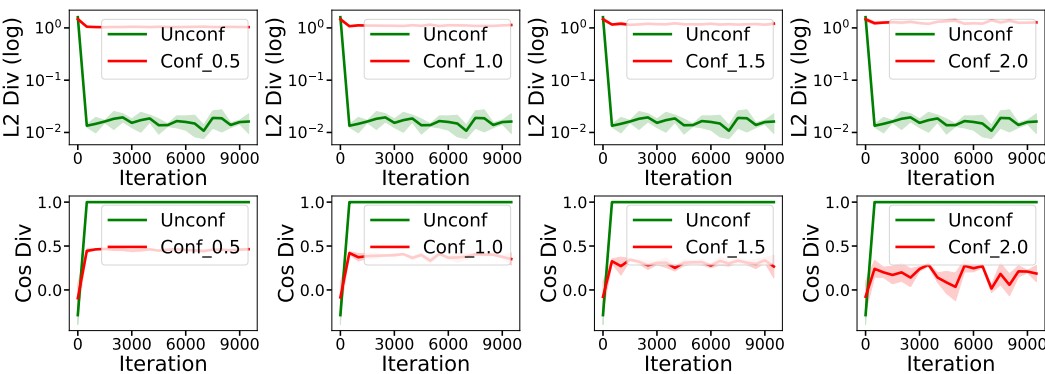

Figure 15: Comparison between estimation with biased weights pre-trained from confounded data and weights pre-trained on unconfounded data, with $X \sim N(0, I)$ and TF@3.

## G.10 NON-GAUSSIAN ANALYSIS

We show the properties of the constructed U_weights, deviation of pre-trained weights, inference bias of the ICL stage and the effectiveness of our proposed DDbias method in the regime of Gamma

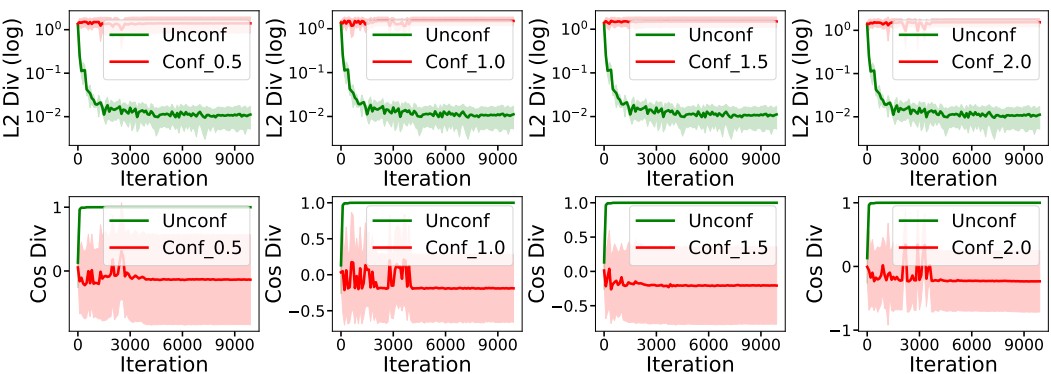

Figure 16: Comparison between estimation with biased weights pre-trained from confounded data and weights pre-trained on unconfounded data, with $X \sim N(0, \Sigma)$ and TF@3.

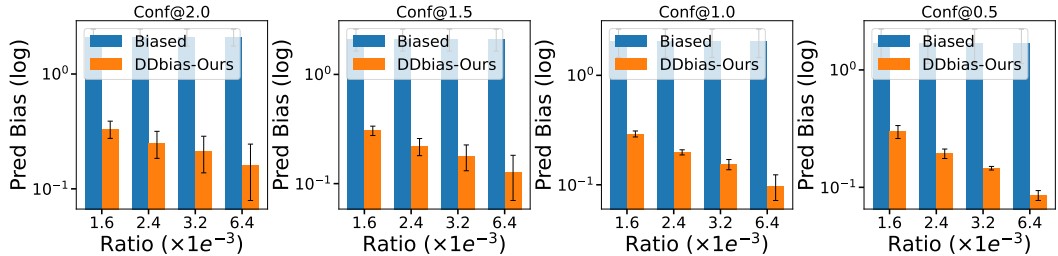

Figure 17: Prediction comparison of our proposed Double-Debiasing method with increasing prompting/pre-training sample ratio with Tf@1, where $X \sim N(0, \Sigma)$.

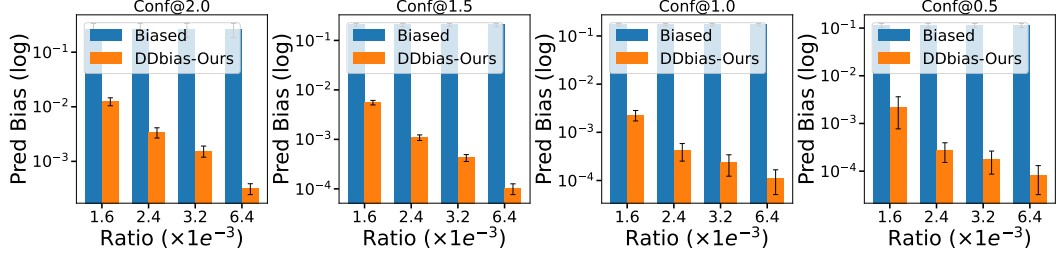

Figure 18: Prediction comparison of our proposed Double-Debiasing method with increasing prompting/pre-training sample ratio with Tf@3, where $X \sim N(0, I)$.

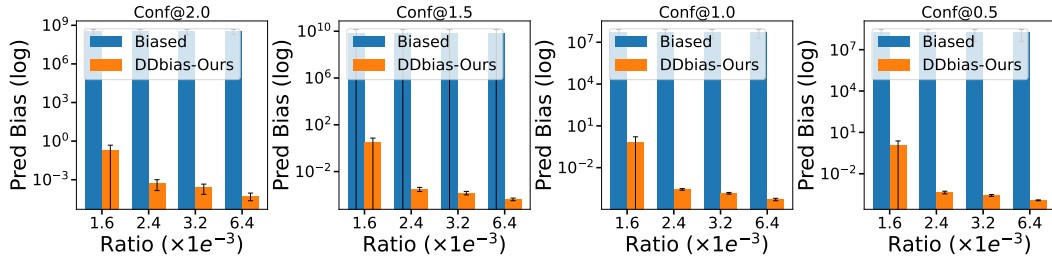

Figure 19: Prediction comparison of our proposed Double-Debiasing method with increasing prompting/pre-training sample ratio with Tf@3, where $X \sim N(0, \Sigma)$.

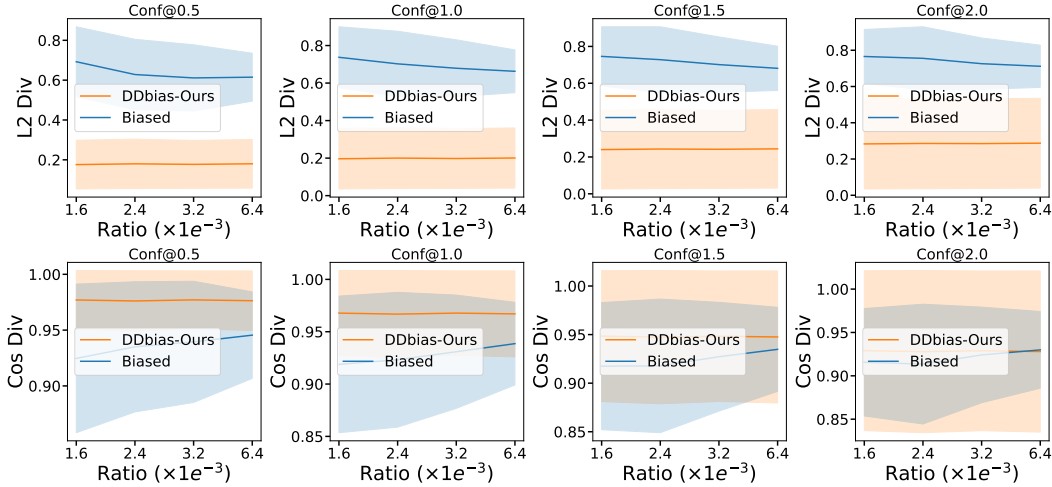

Figure 20: Comparision of Cos Divergence and L2 Divergence of our proposed Double-Debiasing method with increasing prompting/pre-training sample ratio with Tf@1, where $X \sim N(0, \Sigma)$

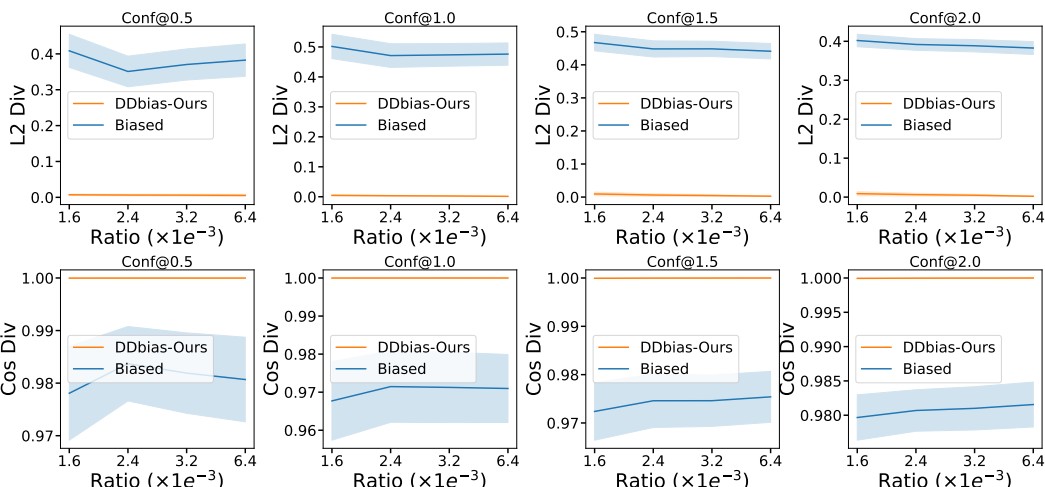

Figure 21: Comparision of Cos Divergence and L2 Divergence of our proposed Double-Debiasing method with increasing prompting/pre-training sample ratio with Tf@3, where $X \sim N(0, \Sigma)$

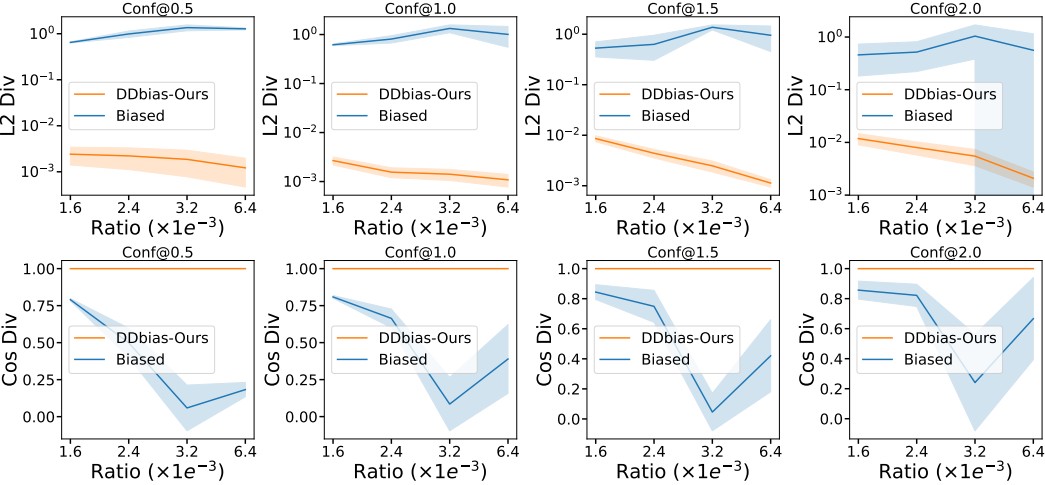

Figure 22: Comparision of Cos Divergence and L2 Divergence of our proposed Double-Debiasing method with increasing prompting/pre-training sample ratio with Tf@3, where $X \sim N(0, I)$.

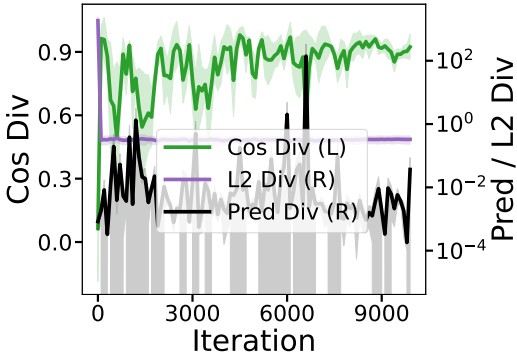

Figure 23: Checking U_weights on Non-Gaussian Data, with $X \sim Gamma(0.1, 10)$ and TF@1.

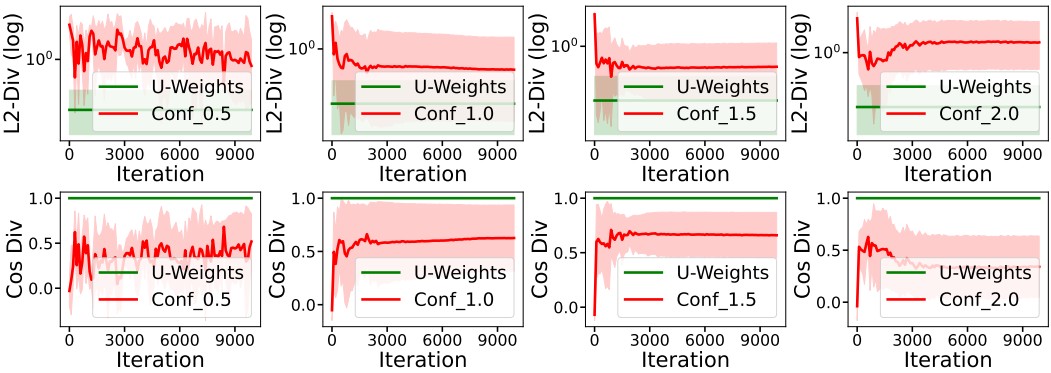

Figure 24: Comparison between estimation with biased weights pre-trained from confounded data and U_weights we construct in the main paper, with $X \sim Gamma(0.1, 10)$ and TF@1.

distribution, where $X$ is sampled from $Gamma(0.1, 10)$, together with the noise $\epsilon$ in Fig 23, 24, 25 and 26.

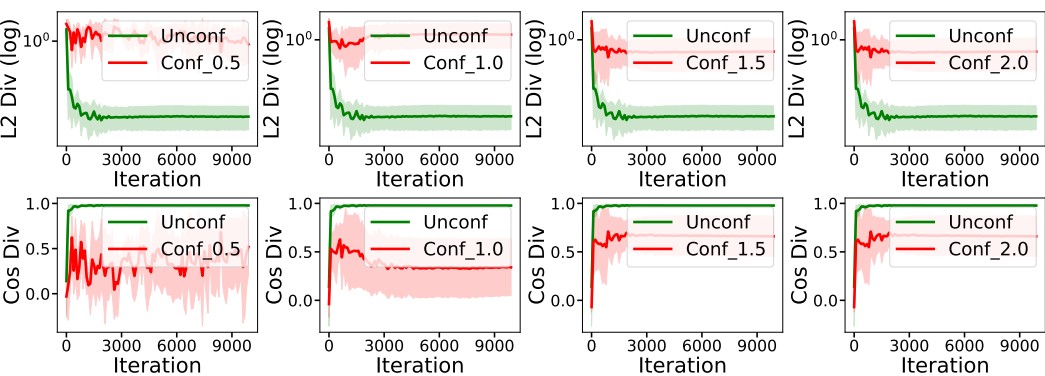

Figure 25: Comparison between estimation with biased weights pre-trained from confounded data and weights pre-trained on unconfounded data, with $X \sim Gamma$ and TF@3.

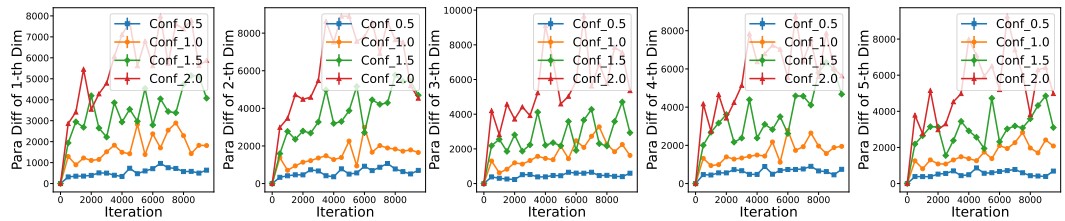

Figure 26: Deviation of Pre-trained Weights along each feature dimension for the 1-th layer of TF@3.

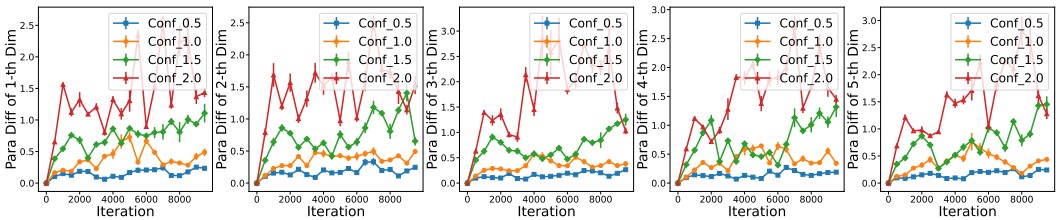

Figure 27: Deviation of Pre-trained Weights along each feature dimension for the 2-th layer of TF@3.

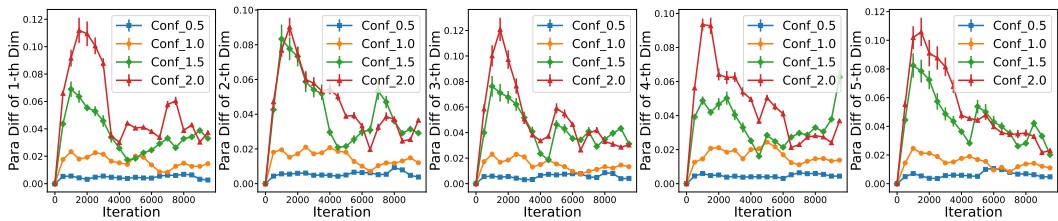

Figure 28: Deviation of Pre-trained Weights along each feature dimension for the 3-th layer of TF@3.

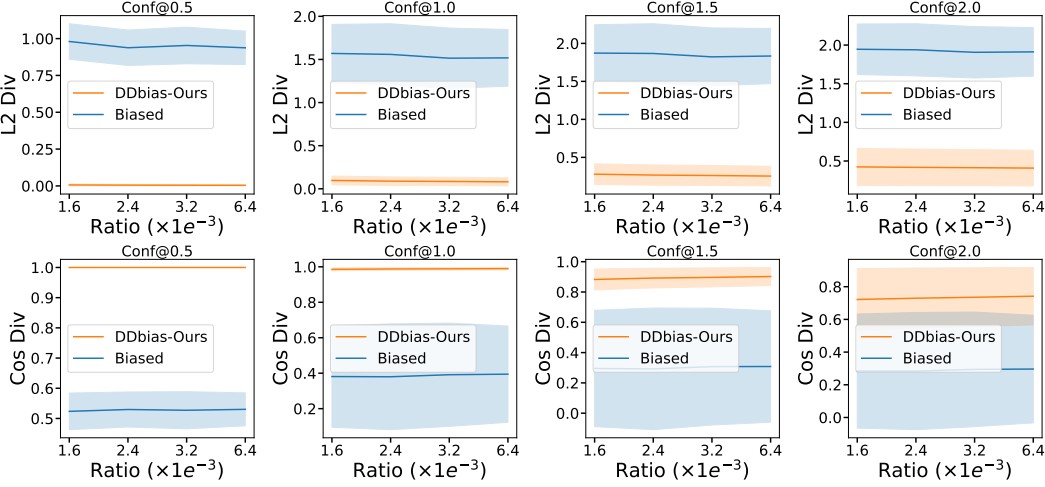

Figure 29: Comparision of Cos Divergence and L2 Divergence of our proposed Double-Debiasing method with increasing prompting/pre-training sample ratio with Tf@1, where $X \sim Gamma(0.1, 10)$

