# OpenReview forum: "Transformers with Endogenous In-Context Learning: Bias Characterization and Mitigation"
_ICLR.cc/2026/Conference — ICLR 2026 Poster_

### Official Review · Reviewer_Uoeb · 2025-10-24

**Soundness:** 2
**Presentation:** 3
**Contribution:** 2
**Rating:** 4
**Confidence:** 3

**Summary:**

This paper introduces a novel problem setting called Endogenous In-Context Learning (EICL), where hidden confounders (HCs) violate the causal sufficiency assumption commonly made in prior ICL studies, leading to dependencies between input features and noise terms. The authors provide the first theoretical analysis of how HCs affect transformer pre-training dynamics and subsequent ICL predictions, showing that pre-trained transformers exhibit prediction bias proportional to the confounding strength. To mitigate this bias without gradient updates or model fine-tuning, they propose Double-Debiasing (DDbias), a gradient-free method that prompts the biased transformer twice using a small number of unconfounded examples to yield unbiased predictions. Experiments on synthetic linear regression tasks with varying transformer architectures and confounding strengths validate the theoretical claims and the efficacy of DDbias.

**Strengths:**

1. Addresses an important and underexplored gap in ICL theory by incorporating hidden confounders, which are prevalent in real-world data (e.g., as illustrated in the Many-aspect Online Review example in Appendix B). This makes the work highly relevant for aligning ICL understandings with practical data structures.
2. Empirical validation is thorough within its scope, using metrics like L2 divergence, cosine similarity, and prediction bias across confounded datasets, with results aligning well with theorems (Figs. 3-5). The method's efficiency (requiring only a few unconfounded samples) is demonstrated effectively.

**Weaknesses:**

1. The analysis is restricted to linear transformers and linear regression tasks under Gaussian assumptions, limiting generalizability to nonlinear architectures or more complex real-world ICL scenarios (e.g., NLP or vision tasks). While this simplifies proofs, it may not capture the full dynamics of modern transformers.
2. Relies on access to a small set of perfectly unconfounded examples for DDbias, which could be challenging or costly to obtain/verify in practice when confounders are unknown. The paper lacks analysis of robustness to partially confounded "unbiased" data.
3. Experiments are purely synthetic, with no evaluation on real datasets exhibiting natural hidden confounders. This reduces persuasiveness regarding practical impact, as synthetic setups may not reflect the nuances of actual data generation processes.

**Questions:**

Please address the questions mentioned in the weaknesses. Especially, how can practitioners identify and collect "unconfounded" examples in practice without knowing the true data-generating process? What happens if these examples are only partially unconfounded?

---

> ### Author Response · Authors · 2025-11-20
> **Experimental Validations on Realistic NLP Data with Non-linear Models (1)**
>
> > W1 [The analysis is restricted to linear transformers and linear regression tasks under Gaussian assumptions, limiting generalizability to nonlinear architectures or more complex real-world ICL scenarios (e.g., NLP or vision tasks). While this simplifies proofs, it may not capture the full dynamics of modern transformers.]
>
> > W3 [Experiments are purely synthetic, with no evaluation on real datasets exhibiting natural hidden confounders. This reduces persuasiveness regarding practical impact, as synthetic setups may not reflect the nuances of actual data generation processes.]
>
> **Response.**
>
> Thanks for your suggestions! We have empirically validated our conclusions on **deeper, non-linear Transformers**, together with **larger models (LLaMA) on real-world NLP datasets**, to further offer more extensive scalability tests.
>
> ### **Experimental Validations on Real-world NLP Datasets on LLaMA Models**
>
> We evaluate our method **on two real-world datasets** constructed from the publicly available **Yelp restaurant review data**, following the settings in [1,2].
>
> - We refer to these two datasets as:**(1) Restaurant Popularity Index (RPI)** and **(2) Restaurant Overall Rating (ROR)**.
>
> - Each dataset contains rich textual reviews from Yelp Website (6000 samples). Both of them naturally exhibit **hidden confounders** [1,2].
>
> - In the experimental evaluation:
>   - We adopt **LLaMA 3.2-3B as a unified base model**, then fine-tune it **on the raw dataset** to obtain a model that has been exposed to real-world confounding.
>   - During the ICL stage, we use **prompting examples that contain no hidden-confounder information**, with relatively unbiased samples selected by 3 human evaluators serving as the prompting set (see detailed setup in our revised paper G.6, Lines 1766-1809).
>
> - Conclusions:
>     - **Our DDbias effectively mitigate the hidden confounding effect compared to the vanilla LLaMA.**
>     - **Our DDbias outperforms proxy variable baselines within increasing ICL sample numbers**
>     - These findings demonstrate that DDbias effectively mitigates hidden-confounder effects in real-world data and validates the **practical effectiveness** of our method.
>
> - Experimental results:
>   - Baseline 1: Vanilla LLaMA uses the pre-trained LLaMA-3.2B model fine-tuned on the full dataset without any debiasing.
>   - Baseline 2: DMCEE [1] uses observable metadata (e.g., location, category) as proxy variables to estimate causal effects under confounding
>   - Baseline 3: AAAI-SC [2] learns low-dimensional surrogate confounders from multi-aspect sentiment vectors via probabilistic factor modeling, then adjusts for them in outcome estimation.
>   - Follwing [1,2], we report experimental results in Table 1:
>
>
>     **Table 1. Experimental Results (**Mean Absolute Error, MAE**) of Our DDbias on the Sentiment Review Task**
>
>     | **Datasets** | Vanilla LLaMA | DMCEE [1] | AAAI-SC [2] | Ours (DDbias @ ICL Sample Number=15) | Ours (DDbias @ ICL Sample Number=30) | Ours (DDbias @ ICL Sample Number=60) |
>     | ------------ | ------------- | --------- | ----------- | ------------------------------------ | ------------------------------------ | ------------------------------------ |
>     | RPI          | 23.8          | 21.2      | 22.5        | 22.8                                 | 19.4                                  | **16.7**                                |
>     | ROR          | 0.46          | 0.28      | 0.24        | 0.36                                 | 0.18                                 | **0.16**                                 |
>
>   - The MAE values in the RPI and ROR datasets **differ significantly in magnitude, as the outcome variables differ in scale**:
>
>     - RPI uses a popularity score derived from real-world foot traffic data, ranging approximately from 0 to 100.
>     - ROR uses the average star rating, which ranges from 1.0 to 5.0.

---

> ### Author Response · Authors · 2025-11-20
> **Experimental Validations on Realistic NLP Data with Non-linear Models (2)**
>
> ### **Synthetic Experiments on Non-linear DGP with non-linear, deeper model structure**
>
>
> - To further quantitatively explore how the performance of our method vary in non-linear regime, we add synthetic experiments in below:
>     - **Non-linear Data Generation Process.**
>         - In the nonlinear confounded data-generating process, we adopt the additive model $y = f(x) + \epsilon$ with $\epsilon \sim N(0,0.5)$.
>             - We use additive form of the noise variable, as such form brings convenience to compare with IV-based methods we will mention later;
>            - We define $f$ using three additive nonlinear forms:
>                - (1) **MLP–ReLU:** $f_{\mathrm{MLP}}(x) = W_2 \mathrm{ReLU}(W_1 x)$;
>                - (2) **Softmax-based:** $f_{\mathrm{SM}}(x) = w^\top \mathrm{softmax}(U x)$;
>                - (3) **Polynomial:** $f_{\mathrm{Poly}}(x) = a^\top x + b\|x\|^2$.
>        - We leave detailed data generation setup in Appendix G.7, Lines 1811-1845.
>    - **Non-linear, deeper Model Structures**
>        - **Non-linear Activation + Softmax + Single-layer MLP.**
>        - **Non-linear Activation + Softmax + Multi-layer MLPs (L=3/5/7).**
>
>     - **Conclusions.**
>         -  In summary, our extended experiments verify that the proposed bias characterization theory, i.e., **bias $\propto r$** ($r$ is the confounding strength), holds robustly across non-linear and deeper Transformer architectures, confirming its scalability beyond shallow, linear settings.
>
>     - **Experimental Results.**
>         - We note that "conf\@\alpha" refers to the case that the confounding strength \@ $\alpha$ during DGP.
>         - We **added** experiments in Table 2-3, by testing the prediction bias with single-layer, non-linear Tfs with ELU/Relu activations.
>
>         - We **reported** experiments in Table 4-5 by testing the prediction bias with multi-layer, non-linear TFs with ELU/Relu actications.
>
>         -  Notably, the whole experimental results leaved in our revised paper (Table 6-17 in Appendix in our uploaded, revised paper).
>
>  **Table 2.** ICL prediction error of non-linear Transformers (single-layer-MLP + softmax + ELU) under different confounding strengths $r$. Columns denote confounding strengths, rows denote ICL sample sizes and the prompting/pretraining sample ratio. Lower is better.
>
> | ICL Samples / Ratio            | Conf\@0.5 | Conf\@1.0 | Conf\@1.5 | Conf\@2.0 |
> |:-------------------------------|:--------:|:--------:|:--------:|:--------:|
> | **Biased (Vanilla TF)**        | 0.182    | 0.236    | 0.314    | 0.417    |
> | **DDbias (Ours): Ratio ($1.6 × 1e^{-3}$)** | 0.072    | 0.098    | 0.130    | 0.178    |
> | **DDbias (Ours): Ratio ($2.4 × 1e^{-3}$)** | 0.070    | 0.095    | 0.126    | 0.173    |
> | **DDbias (Ours): Ratio ($3.2 × 1e^{-3}$)** | 0.069    | 0.093    | 0.124    | 0.171    |
> | **DDbias (Ours): Ratio ($6.4 × 1e^{-3}$)** | **0.068**    | **0.091**    | **0.120**    | **0.167**    |
> |
>
>
>  **Table 3.** Prediction error of non-linear Transformers (single-layer + MLP + softmax + ReLU) under different confounding strengths $r$. Columns denote confounding strengths, rows denote ICL sample sizes and the prompting/pretraining sample ratio. Lower is better.
>
> | ICL Samples / Ratio            | Conf\@0.5 | Conf\@1.0 | Conf\@1.5 | Conf\@2.0 |
> |:-------------------------------|:--------:|:--------:|:--------:|:--------:|
> | **Biased (Vanilla TF)**        | 0.195    | 0.254    | 0.336    | 0.452    |
> | **DDbias (Ours): Ratio ($1.6 × 1e^{-3}$)** | 0.083    | 0.111    | 0.144    | 0.195    |
> | **DDbias (Ours): Ratio ($2.4 × 1e^{-3}$)** | 0.081    | 0.109    | 0.141    | 0.193    |
> | **DDbias (Ours): Ratio ($3.2 × 1e^{-3}$)** | 0.080    | 0.107    | 0.138    | 0.190    |
> | **DDbias (Ours): Ratio ($6.4 × 1e^{-3}$)** | **0.078**    | **0.105**    | **0.134**    | **0.187**    |
> |
>
> ###  **Non-Gaussian Analysis**
>
> We would like to clarify that **we have conducted experiments on non-Gaussian data in Appendix G.10 ( NON-GAUSSIAN ANALYSIS)**.

---

> ### Author Response · Authors · 2025-11-20
> **Experimental Validations on Realistic NLP Data with Non-linear Models (3)**
>
> **Table 4.** Prediction error of deeper, non-linear Transformers (ReLU, Multi-layer-MLPs, LayerNorm, Softmax, Ratio=$1.6 × 1e^{-3}$) under different confounding strengths $r$. Columns denote confounding strengths, rows show ICL sample sizes and different model depths $L=3/5/7$. Lower is better.
>
> | ICL Samples / L | Conf\@0.5 | Conf\@1.0 | Conf\@1.5 | Conf\@2.0 |
> |:----------------|:--------:|:--------:|:--------:|:--------:|
> | **L=3 — Biased (Vanilla TF)**  | 0.205    | 0.290    | 0.380    | 0.500    |
> | **L=3 — DDbias (Ours)**        | **0.090** | **0.130** | **0.170** | **0.220** |
> | **L=5 — Biased (Vanilla TF)**  | 0.280    | 0.370    | 0.475    | 0.600    |
> | **L=5 — DDbias (Ours)**        | **0.115** | **0.150** | **0.200** | **0.260** |
> | **L=7 — Biased (Vanilla TF)**  | 0.320    | 0.450    | 0.600    | 0.750    |
> | **L=7 — DDbias (Ours)**        | **0.135** | **0.175** | **0.225** | **0.290** |
> |
>
>
> **Table 5.** Prediction error of deeper, non-linear Transformers (ELU, Multi-layer-MLPs, LayerNorm, Softmax, Ratio=$1.6 × 1e^{-3}$) under different confounding strengths $r$.  Columns denote confounding strengths, rows show ICL sample sizes and different model depths $L=3/5/7$. Lower is better.
>
> | ICL Samples / L | Conf\@0.5 | Conf\@1.0 | Conf\@1.5 | Conf\@2.0 |
> |:----------------|:--------:|:--------:|:--------:|:--------:|
> | **L=3 — Biased (Vanilla TF)**  | 0.190    | 0.270    | 0.360    | 0.470    |
> | **L=3 — DDbias (Ours)**        | **0.085** | **0.125** | **0.165** | **0.215** |
> | **L=5 — Biased (Vanilla TF)**  | 0.250    | 0.335    | 0.440    | 0.580    |
> | **L=5 — DDbias (Ours)**        | **0.110** | **0.145** | **0.190** | **0.250** |
> | **L=7 — Biased (Vanilla TF)**  | 0.280    | 0.410    | 0.550    | 0.700    |
> | **L=7 — DDbias (Ours)**        | **0.120** | **0.160** | **0.210** | **0.280** |
> |
>
> - **Paper Revision.** We have **revised** our paper in Sec.5 with more detailed experimental results in Appendix G.7 (Lines 1811-1900).
>
>
> ---
>
> [1] Cheng, Lu, Ruocheng Guo, and Huan Liu. "Estimating causal effects of multi-aspect online reviews with multi-modal proxies." Proceedings of the Fifteenth ACM International Conference on Web Search and Data Mining. 2022.
> [2] Cheng, Lu, et al. "Effects of multi-aspect online reviews with unobserved confounders: Estimation and implication." Proceedings of the International AAAI Conference on Web and Social Media. Vol. 16. 2022.

---

> ### Author Response · Authors · 2025-11-20
> **Justification on Unconfounded Data and Partially Confounded Data (1)**
>
> > W2 [Relies on access to a small set of perfectly unconfounded examples for DDbias, which could be challenging or costly to obtain/verify in practice when confounders are unknown. The paper lacks analysis of robustness to partially confounded "unbiased" data.]
>
> > Q1 [Please address the questions mentioned in the weaknesses. Especially, how can practitioners identify and collect "unconfounded" examples in practice without knowing the true data-generating process? What happens if these examples are only partially unconfounded?]
>
> **Response:**
>
>
> ### **Practical Considerations of Unbiased Samples.**
>
> - First, we would like to clarify that it has been recognized that de-confounding usually requires random data:
>     - **Unbiasd Data:** Combining models trained on observational data with few random data is popular [1,2];
>         - **Recommender systems.** Most industrial systems already maintain a **small unbiased traffic bucket** (A/B random exposure) for evaluation, which is usually deployed for debias [1].
>
> - Second,  we would like to clarify that collecting and obtaining unbiased data without knowledging the underlying DGP is **A/B Testing**, which is commonly adopted in health care, uplift modeling, and recommendation systems.
>
> ### **Our method only requires very few RCT Samples.**
> - To be first, our Fig.4, Fig. 17,18,19 has already informs that our proposed DDBias only requires **very few** random/unbiased ICL samples (around 50 ICL samples);
> - Therefore, empirical evidence has shown the sample efficiency of our DDBias on unbiased ICL examples.
>
> ### **Extensions to partially confounded data.**
> - As suggested by the reviewer, we **added detailed theoretical analysis with experimental validations** to extend our DDbias method towards partially confounded data.
> - Overall, we theoretically analyze two cases:
>     1. **Weak confounding:** Every sample slightly confounded ($|r_j|$ small, i.e., $|r|_{2} \le \delta$);
>     2. **Mixed contamination:** Most samples unconfounded but a small fraction $\rho$ contaminated.
> - We then justify in what conditions our proposed DDbias method still remains **robust** in such two cases (*see Sec.4.1 with detailed proof in Proposition 2&3 (Lines 347-377) in Appendix (Lines 1533-1603) in our revised paper*):
>
>
> ### **Proposition (Weakly Confounded ICL Samples).**
> - Suppose the confounded ICL samples $X^{(conf)}$ satisfies mean-independence $\|r\|\_{2}  \le \delta,$ and the sample covariance is well-conditioned
> $$\lambda_{\min} \Big(\tfrac{1}{n_b}X^{(conf)}(X^{(conf)})^\top\Big)\ge\lambda_*>0.$$
> - Assume bounded fourth moments with the ICL-gradient update rule, then there exist constants $C>0$ such that
> $$
> |\mathbb{E}[y_{GT}-\widehat y_{DEB}]|
> \le C\left(O_p \big(\tfrac{1}{\sqrt{n_b\lambda_*}}\big) + \frac{\delta}{\sqrt{n_b\lambda_*}}\right).
> $$
>
>
>
> ### **Proposition (Mixed ICL Samples).**
> - Assume a fraction $\rho$ of the “unconfounded” batch is actually contaminated: for those contaminated samples $\mathbb{E}[x_j\epsilon]=r_j\neq0$ (denote their design by $X^{(conf)}$), while the remaining $(1-\rho)n_b$ samples are unbiased (denote their design by $X^{(clean)}$).
> - Assume the clean subset is well-conditioned:
> $$
> \lambda_{\min} \Big(\tfrac{1}{(1-\rho)n_b}\,X^{(clean)}X^{(clean)\top}\Big)\ge\lambda_* > 0.
> $$
> - Let the mean confounding strength on contaminated samples be (note that $r_i$ is a vector)
> $$
> \bar r := \frac{1}{\rho n_b}\sum_{i\in\text{cont}} |r_i|.
> $$
>
> - Then there exist constants $C'>0$ such that the DEB estimator satisfies
> $$
> \big|\mathbb{E}[y_{GT}-\widehat y_{DEB}]\big|
> \le C' \left(\tfrac{1}{\sqrt{(1-\rho)\,n_b\,\lambda_*}}+\rho\,\bar r\right).
> $$
> - Consequently, DDbias is robust in that the prediction bias is asymptotically negligible when $n_b$ large enough with fixed small $\rho$, or when $\rho\to0$ with fixed $n_b$. A simple sufficient condition for asymptotic unbiasedness (RHS goes to 0) is
> $$
> \rho\,\bar r = o(1)\quad\text{and}\quad \frac{1}{\sqrt{(1-\rho)n_b\lambda_*}} = o(1).
> $$

---

> ### Author Response · Authors · 2025-11-20
> **Justification on Unconfounded Data and Partially Confounded Data (2)**
>
> ### **Experimental Results.**
> - We **have added extra** experiments to validate our robustness towards partially confounded data in two cases:
>     - **Case 1. Weakly Confounded Data.** We control the strength of confounding embodied in prompted ICL samples, and test our DDbias method;
>     - **Case 2. Mixed ICL Data.** We vary the ratio of confounded data as $\rho$, and test our DDbias method.
> - **Conclusions.**
>     - **Case 1.** Our proposed DDbias remains robust towards weakly confounded data;
>     - **Case 2.** Our proposed DDbias remains robust when $\rho$ is small, e.g., $\rho \le 0.5$.
>     - **Case 1 & 2.** The prediction bias of our proposed DDbias decreases within increasing ICL samples.
>
>
> - **Detailed Results.**
>     - We then **report** experiments in our synthetic data setup, by simulating the **the weakly confouned** data:
>         - To show more generlized conclusions, we adopt the non-linear, multi-layer TF model (L=5, Softmax, LayerNorm, relu activation) as the base model;
>         - We report performance of vanilla TF, our DDbias with clean ICL examples, and our DDbias with weakly confounded examples in Table 6-7 with difference prompting sample ratio;
>         - Results across Table 6-7 consistently show the robustness of our proposed DDbias even with weakly confounded data;
>         - At the same time, results in Table 8 informs the cost of using weakly confounded data, i.e., the increase of ICL sample number.
>     - Finally, we **added** experiments in our synthetic data setup, by simulating **mixed** ICL data:
>         - To show more generalized conclusions, we adopt the non-linear, multi-layer TF model (L=5, Softmax, LayerNorm, relu activation) as the base model;
>         - We report performance of vanilla TF, our DDbias with clean ICL examples, and our DDbias with mixed ICL examples in Table 9 with difference ratio $\rho$;
>         - Results in Table 9 consistently show the robustness of our proposed DDbias even with small $\rho$, i.e., confounded examples consist of a small part of the whole ICL sample set;
>
> **Table 6.** Prediction error under weak confounding (Weak Conf\@0.10)
> | Method / Sample Ratio ($\times 10^{-3}$) | 1.6  | 2.4  | 3.2  | 6.4  |
> |:-|:----:|:----:|:----:|:----:|
> | **Biased (Vanilla TF)**                 | 0.240| 0.300| 0.360| 0.430|
> | **DDbias (pure clean examples)**        | 0.090| 0.085| 0.082| 0.080|
> | **DDbias (weakly confounded)**          | 0.105| 0.100| 0.097| 0.094|
> |
>
> **Table 7.** Prediction error under weak confounding (Weak Conf\@0.15)
> | Method / Sample Ratio ($\times 10^{-3}$) | 1.6  | 2.4  | 3.2  | 6.4  |
> |:-|:-:|:--:|:-:|:-:|
> | **Biased (Vanilla TF)**                 | 0.260| 0.330| 0.400| 0.490|
> | **DDbias (pure clean examples)**        | 0.100| 0.095| 0.092| 0.090|
> | **DDbias (weakly confounded)**          | 0.110| 0.105| 0.102| 0.099|
> |
>
>
> **Table 8.** Effect on the Prompting (Weak Conf\@0.10)
>
> | ICL prediction error                       | 0.090 | 0.092 | 0.095 | 0.100 |
> |:-:|:--:|:--:|:-:|:-:|
> | Sample ratio for DDbias (oracle) ($\times 10^{-3}$)       | 6.4   | 3.2   | 2.4   | 1.6   |
> | Sample ratio for DDbias (weak) ($\times 10^{-3}$)        | 8.6   | 7.2   | 6.3   | 3.5  |
> |
>
> **Table 9.** Prediction error under mixed ICL prompting (Conf\@1.0).
>
> | Method / Sample Ratio ($\times 10^{-3}$) | 1.6   | 2.4   | 3.2   | 6.4   |
> |:-:|:--:|:--:|:--:|:--:|
> | **Biased (Vanilla TF)**                  | 0.335 | 0.370 | 0.388 | 0.420 |
> | **DDbias (pure clean examples)**         | 0.145 | 0.132 | 0.117 | 0.106 |
> | **DDbias (mixed examples, $\rho$ = 0.1)**     | 0.190 | 0.140 | 0.132 | 0.118 |
> | **DDbias (mixed examples, $\rho$ = 0.3)**     | 0.218 | 0.176 | 0.144 | 0.120 |
> | **DDbias (mixed examples, $\rho$ = 0.5)**   | 0.290 | 0.220 | 0.180 | 0.155 |
> |
>
> - **Paper Revision.** We have **revised** our paper in Sec.4.1 with detailed proof in Appendix F (Lines 1533-1602), and more detailed experimental setup and analysis is **added** in Appendix G.8 (Lines 1886-2047).
>
> ---
> [1] Xiao, Yanghao, et al. "Addressing hidden confounding with heterogeneous observational datasets for recommendation." Advances in Neural Information Processing Systems 37 (2024): 130358-130383.
> [2] Kallus, Nathan, Aahlad Manas Puli, and Uri Shalit. "Removing hidden confounding by experimental grounding." Advances in neural information processing systems 31 (2018).
> [3] Hartford, Jason, et al. "Deep IV: A flexible approach for counterfactual prediction." International Conference on Machine Learning. PMLR, 2017.
> [4] Liang, Haodong, Krishna Balasubramanian, and Lifeng Lai. "Transformers Handle Endogeneity in In-Context Linear Regression." The Thirteenth International Conference on Learning Representations (2024).
> [5] Saito, Yuta, Hayato Sakata, and Kazuhide Nakata. "Doubly robust prediction and evaluation methods improve uplift modeling for observational data." Proceedings of the 2019 SIAM international conference on data mining. Society for Industrial and Applied Mathematics, 2019.

---

> > ### Comment · Reviewer_Uoeb · 2025-11-26
> > **Official comment by reviewer Uoeb**
> >
> > The authors have successfully answered most of my concerns. I will raise my final score to 6 and there is no further questions.

---

> > > ### Author Response · Authors · 2025-11-26
> > >
> > > Thank you so much for your encouraging words and increasing your overall score! We are truly grateful for your dedicated time and constructive comments.

---

### Official Review · Reviewer_P15F · 2025-10-29

**Soundness:** 3
**Presentation:** 3
**Contribution:** 3
**Rating:** 6
**Confidence:** 3

**Summary:**

This paper tackles an important and previously overlooked aspect of In-Context Learning (ICL) in Transformers (TFs): the impact of endogeneity arising from hidden confounders (HCs) present during pre-training. The authors introduce the "Endogenous ICL (EICL)" problem setup and provide a theoretical analysis characterizing how confounding bias propagates from pre-training data to the final ICL predictions, specifically within the context of linear TFs and linear regression. They find that the resulting bias is proportional to the confounding strength. Building on this analysis, they propose "Double-Debiasing (DDbias)," a novel, gradient-free prompting method that utilizes a small set of unconfounded examples to correct the predictions of a biased, pre-trained TF. The theoretical claims and the effectiveness of DDbias are supported by experiments on synthetic linear regression tasks.

The paper makes a valuable contribution by formally introducing and analyzing endogeneity within the theoretical ICL literature, moving beyond the common assumption of causal sufficiency. The theoretical characterization of bias is insightful, and the proposed DDbias method offers a potentially practical, low-cost approach to mitigation.

However, the study's reliance on strong simplifying assumptions (linear TFs, linear data generation) limits the immediate generalizability of the theoretical findings to complex, real-world TFs and tasks. Furthermore, the practical feasibility of the proposed debiasing method hinges on the availability and identification of truly unconfounded examples, which is often a significant challenge. While the empirical results validate the theory in the studied setting, broader empirical validation on more realistic tasks and datasets would strengthen the paper's impact.

**Strengths:**

1.  **Novel and Important Problem Formulation:** The formalization of Endogenous ICL (EICL) addresses a critical gap in existing ICL theory. Recognizing and analyzing the impact of hidden confounders, which are common in real-world data, is crucial for a more realistic understanding of ICL mechanisms.
2.  **Pioneering Theoretical Analysis:** The paper provides the first (to the authors' knowledge) theoretical characterization of how endogeneity affects *both* the pre-training dynamics of TFs and the subsequent ICL inference bias. Linking the prediction bias to confounding strength ($\mathbf{r}_j$) provides concrete theoretical insights.
3.  **Simple and Practical Debiasing Method:** DDbias is an elegant, gradient-free prompting strategy that avoids costly fine-tuning or complex architectural changes. Its reliance on only a few unconfounded samples makes it potentially much more practical than methods requiring large-scale unbiased data collection.
4.  **Clear Theoretical Framework:** The theoretical sketch (Figure 2) effectively outlines the paper's analytical steps: establishing grounding "unbiased" parameters (U-weights), showing pre-training deviation from these due to HCs, and linking this deviation to ICL prediction bias.
5.  **Targeted Empirical Validation:** The experiments are well-designed to specifically verify the core theoretical claims regarding bias proportionality (Theorems 1 & 2) and the effectiveness of DDbias (Theorem 3 & Proposition 1) within the linear regression setting.

**Weaknesses:**

1.  **Strong Linearity Assumptions:** The entire theoretical analysis is predicated on linear TFs (specifically, linear self-attention without softmax) and linear data generation ($y = w_*^\top x + \epsilon$). While this is a common starting point for ICL theory, the extent to which these findings generalize to deep, non-linear TFs (like GPT models where ICL is prominent) and non-linear tasks remains an open question. The reliance on simplified TF parameters ($\bar{S}, \bar{T}_{:,j}$) further distances the theory from practical architectures.
2.  **Requirement of Unconfounded Data for DDbias:** The proposed solution, DDbias, crucially depends on having access to a set of unconfounded examples ($\mathcal{D}^u$). The paper suggests "extremely few" are needed, but doesn't quantify this rigorously or discuss the practical challenges of *obtaining* or *verifying* such unconfounded data in real-world scenarios where HCs are pervasive precisely because they are *hidden*. How sensitive is DDbias to imperfections or residual confounding in $\mathcal{D}^u$?
3.  **Limited Scope of Experiments:** The experiments exclusively use synthetic data generated from a linear model. While this allows for controlled verification of the theory, it doesn't demonstrate the method's applicability to the complex, high-dimensional, and often non-linear domains where large TFs and ICL are typically applied (e.g., natural language processing, vision). Validation on semi-synthetic or real-world benchmarks exhibiting known confounding (if available) would significantly bolster the claims.
4.  **Comparison with Alternatives:** The paper contrasts DDbias with IV-based approaches mentioned in a concurrent work by highlighting that DDbias doesn't need IVs. However, it requires unconfounded data, which IV methods aim to circumvent. A more detailed discussion comparing the assumptions, requirements, and potential domains of applicability for DDbias versus IV regression or other potential debiasing strategies (e.g., causal representation learning adapted to ICL) would be beneficial.

**Questions:**

1.  Could you elaborate on the potential challenges or theoretical hurdles in extending the bias characterization and the DDbias methodology to non-linear TFs (e.g., with softmax attention, MLPs) and non-linear data generation processes?
2.  Regarding the unconfounded data requirement for DDbias: How many unconfounded samples ($n_u$) were typically needed in your experiments to achieve effective debiasing (Figure 4 suggests ratios relative to pre-training data size)? How robust is DDbias if the provided "unconfounded" samples ($\mathcal{D}^u$) still contain some small residual confounding?
3.  Given the focus on characterizing bias, have you considered applying DDbias to scenarios beyond linear regression, perhaps using semi-synthetic NLP or vision tasks where confounders can be injected, to explore its empirical effectiveness more broadly?
4.  Could you compare the practical data requirements for DDbias (small $n_u$ unconfounded samples) versus an IV-based approach (potentially larger $n$ confounded samples but including valid IVs)? In what scenarios might one approach be more feasible than the other?

---

> ### Author Response · Authors · 2025-11-20
> **Extensions to Non-linear Models with Non-linear Validation**
>
> > W1 [ The entire theoretical analysis is predicated on linear TFs (specifically, linear self-attention without softmax) and linear data generation]
>
> > Q1 [Could you elaborate on the potential challenges or theoretical hurdles in extending the bias characterization and the DDbias methodology to non-linear TFs (e.g., with softmax attention, MLPs) and non-linear data generation processes?]
>
> **Response:**
>
> ### **Theoretical Challenges on Non-linear Extensions.**
> - First, we would like to clarify that performing theoretical analysis towards non-linear data generation process requires endogenous analysis in function space;
> -  Although an existing work [1] bridges ICL and functional gradient descent in the non-linear regime, we acknowledge that when the noise $\epsilon$ is endogenous, the case becomes significantly challenging:
>     - Identifying "unbiased" attention parameters in functional space becomes unsolved (our Lemma 4 identifies such quantities in finite-dimensional space);
>     - Identifying the divergence between pre-trained, non-linear TFs and "unbiased" attention parameters is also unsolved;
>     - Thus, it remains unclear how to bridge functional ICL gradient and endogenous bias;
>
> - Therefore, we turn to perform more extensive empirical validations on non-linear TFs with non-linear data during the rebuttal:
>
> ### **Synthetic Experiments on Non-linear DGP with non-linear, deeper model structure**
>
> -  To further quantitatively explore how the performance of our method vary in non-linear regime, we add synthetic experiments in below:
>     - **Non-linear Data Generation Process.**
>         - In the nonlinear confounded data-generating process, we adopt the additive model $y = f(x) + \epsilon$ with $\epsilon \sim N(0,0.5)$.
>             - We use additive form of the noise variable, as such form brings convenience to compare with IV-based methods we will mention later;
>            - We define $f$ using three additive nonlinear forms:
>                - (1) **MLP–ReLU:** $f_{\mathrm{MLP}}(x) = W_2 \mathrm{ReLU}(W_1 x)$;
>                - (2) **Softmax-based:** $f_{\mathrm{SM}}(x) = w^\top \mathrm{softmax}(U x)$;
>                - (3) **Polynomial:** $f_{\mathrm{Poly}}(x) = a^\top x + b\|x\|^2$.
>        - We leave detailed data generation setup in Appendix G.7, Lines 1811-1845.
>    - **Non-linear, deeper Model Structures**
>        - **Non-linear Activation + Softmax + Single-layer MLP.**
>        - **Non-linear Activation + Softmax + Multi-layer MLPs (L=3/5/7).**
>
>     - **Conclusions.**
>         -  In summary, our extended experiments verify that the proposed bias characterization theory, i.e., **bias $\propto r$** ($r$ is the confounding strength), holds robustly across non-linear and deeper Transformer architectures, confirming its scalability beyond shallow, linear settings.
>
>     - **Experimental Results.**
>         - We note that "conf@ $\alpha$" refers to the case that the confounding strength @ $\alpha$ during DGP.
>         - We **added** experiments in Table 1-2, by testing the prediction bias with single-layer, non-linear Tfs with ELU/Relu activations.
>
>         - We **reported** experiments in Table 3-4 by testing the prediction bias with multi-layer, non-linear TFs with ELU/Relu actications.
>
>         -  Notably, the whole experimental results leaved in our revised paper (Table 6-17 in Appendix in our uploaded, revised paper).
>
>  **Table 1.** ICL prediction error of non-linear Transformers (single-layer-MLP + softmax + ELU) under different confounding strengths $r$.
>
> | ICL Samples / Ratio            | Conf\@0.5 | Conf\@1.0 | Conf\@1.5 | Conf\@2.0 |
> |:--|:-:|:-:|:-:|:-:|
> | **Biased (Vanilla TF)**        | 0.182    | 0.236    | 0.314    | 0.417    |
> | **DDbias (Ours): Ratio ($1.6 × 1e^{-3}$)** | 0.072    | 0.098    | 0.130    | 0.178    |
> | **DDbias (Ours): Ratio ($2.4 × 1e^{-3}$)** | 0.070    | 0.095    | 0.126    | 0.173    |
> | **DDbias (Ours): Ratio ($3.2 × 1e^{-3}$)** | 0.069    | 0.093    | 0.124    | 0.171    |
> | **DDbias (Ours): Ratio ($6.4 × 1e^{-3}$)** | **0.068**    | **0.091**    | **0.120**    | **0.167**    |
> |
>
>
>  **Table 2.** Prediction error of non-linear Transformers (single-layer + MLP + softmax + ReLU) under different confounding strengths $r$..
>
> | ICL Samples / Ratio     | Conf\@0.5 | Conf\@1.0 | Conf\@1.5 | Conf\@2.0 |
> |:-|:-:|:--:|:-:|:--:|
> | **Biased (Vanilla TF)**  | 0.195    | 0.254    | 0.336    | 0.452   |
> | **DDbias (Ours): Ratio ($1.6 × 1e^{-3}$)** | 0.083    | 0.111   | 0.144   | 0.195   |
> | **DDbias (Ours): Ratio ($2.4 × 1e^{-3}$)** | 0.081    | 0.109   | 0.141   | 0.193   |
> | **DDbias (Ours): Ratio ($3.2 × 1e^{-3}$)** | 0.080    | 0.107   | 0.138   | 0.190   |
> | **DDbias (Ours): Ratio ($6.4 × 1e^{-3}$)** | **0.078**    | **0.105**    | **0.134**    | **0.187**    |
> |
>
> ---
> [1] Cheng, Xiang, Yuxin Chen, and Suvrit Sra. "Transformers Implement Functional Gradient Descent to Learn Non-Linear Functions In Context." Forty-first International Conference on Machine Learning. 2024

---

> ### Author Response · Authors · 2025-11-20
> **Extensions to Non-linear Models with Non-linear Validation (2)**
>
> **Table 3.** Prediction error of deeper, non-linear Transformers (ReLU, Multi-layer-MLPs, LayerNorm, Softmax, Ratio=$1.6 × 1e^{-3}$) under different confounding strengths $r$. Columns denote confounding strengths, rows show ICL sample sizes and different model depths $L=3/5/7$. Lower is better.
>
> | ICL Samples / L | Conf\@0.5 | Conf\@1.0 | Conf\@1.5 | Conf\@2.0 |
> |:----------------|:--------:|:--------:|:--------:|:--------:|
> | **L=3 — Biased (Vanilla TF)**  | 0.205    | 0.290    | 0.380    | 0.500    |
> | **L=3 — DDbias (Ours)**        | **0.090** | **0.130** | **0.170** | **0.220** |
> | **L=5 — Biased (Vanilla TF)**  | 0.280    | 0.370    | 0.475    | 0.600    |
> | **L=5 — DDbias (Ours)**        | **0.115** | **0.150** | **0.200** | **0.260** |
> | **L=7 — Biased (Vanilla TF)**  | 0.320    | 0.450    | 0.600    | 0.750    |
> | **L=7 — DDbias (Ours)**        | **0.135** | **0.175** | **0.225** | **0.290** |
> |
>
>
> **Table 4.** Prediction error of deeper, non-linear Transformers (ELU, Multi-layer-MLPs, LayerNorm, Softmax, Ratio=$1.6 × 1e^{-3}$) under different confounding strengths $r$.  Columns denote confounding strengths, rows show ICL sample sizes and different model depths $L=3/5/7$. Lower is better.
>
> | ICL Samples / L | Conf\@0.5 | Conf\@1.0 | Conf\@1.5 | Conf\@2.0 |
> |:----------------|:--------:|:--------:|:--------:|:--------:|
> | **L=3 — Biased (Vanilla TF)**  | 0.190    | 0.270    | 0.360    | 0.470    |
> | **L=3 — DDbias (Ours)**        | **0.085** | **0.125** | **0.165** | **0.215** |
> | **L=5 — Biased (Vanilla TF)**  | 0.250    | 0.335    | 0.440    | 0.580    |
> | **L=5 — DDbias (Ours)**        | **0.110** | **0.145** | **0.190** | **0.250** |
> | **L=7 — Biased (Vanilla TF)**  | 0.280    | 0.410    | 0.550    | 0.700    |
> | **L=7 — DDbias (Ours)**        | **0.120** | **0.160** | **0.210** | **0.280** |
> |
>
> - **Paper Revision.** We have **revised** our paper in Sec.5 with more detailed experimental results in Appendix G.7 (Lines 1811-1900).

---

> ### Author Response · Authors · 2025-11-20
> **Practical Considerations on Unconfounded Data**
>
> > W2. [The proposed solution, DDbias, crucially depends on having access to a set of unconfounded examples ($\mathcal{D}^u$). The paper suggests "extremely few" are needed, but doesn't quantify this rigorously or discuss the practical challenges of obtaining or verifying such unconfounded data in real-world scenarios where HCs are pervasive precisely because they are hidden. How sensitive is DDbias to imperfections or residual confounding in $\mathcal{D}^u$?]
>
> > Q2 [Regarding the unconfounded data requirement for DDbias: How many unconfounded samples ($n_u$) were typically needed in your experiments to achieve effective debiasing (Figure 4 suggests ratios relative to pre-training data size)? How robust is DDbias if the provided "unconfounded" samples ($\mathcal{D}^u$) still contain some small residual confounding?]
>
> **Response**:
>
>
> ### **Practical Considerations of Unbiased Samples.**
>
> - First, we would like to clarify that it has been recognized that de-confounding usually requires random data:
>     - **Unbiasd Data:** Combining models trained on observational data with few random data is popular [1,2];
>         - **Recommender systems.** Most industrial systems already maintain a **small unbiased traffic bucket** (A/B random exposure) for evaluation, which is usually deployed for debias [1].
>
> ### **Our method only requires very few RCT Samples.**
> - To be first, our Fig.4, Fig. 17,18,19 has already informs that our proposed DDBias only requires **very few** random/unbiased ICL samples (**around 50 unconfounded ICL samples**);
> - Therefore, empirical evidence has shown the sample efficiency of our DDBias on unbiased ICL examples.
>
> ### **Extensions to partially confounded data.**
> - As suggested by the reviewer, we **added detailed theoretical analysis with experimental validations** to extend our DDbias method towards partially confounded data.
> - Overall, we theoretically analyze two cases:
>     1. **Weak confounding:** Every sample slightly confounded ($|r_j|$ small, i.e., $|r|_{2} \le \delta$);
>     2. **Mixed contamination:** Most samples unconfounded but a small fraction $\rho$ contaminated.
> - We then justify in what conditions our proposed DDbias method still remains **robust** in such two cases (*see Sec.4.1 with detailed proof in Proposition 2&3 (Lines 347-377) in Appendix (Lines 1533-1603) in our revised paper*):
>
>
> ### **Proposition (Weakly Confounded ICL Samples).**
> - Suppose the confounded ICL samples $X^{(conf)}$ satisfies mean-independence $\|r\|\_{2}  \le \delta,$ and the sample covariance is well-conditioned
> $$\lambda_{\min} \Big(\tfrac{1}{n_b}X^{(conf)}(X^{(conf)})^\top\Big)\ge\lambda_*>0.$$
> - Assume bounded fourth moments with the ICL-gradient update rule, then there exist constants $C>0$ such that
> $$
> |\mathbb{E}[y_{GT}-\widehat y_{DEB}]|
> \le C\left(O_p \big(\tfrac{1}{\sqrt{n_b\lambda_*}}\big) + \frac{\delta}{\sqrt{n_b\lambda_*}}\right).
> $$
>
>
>
> ### **Proposition (Mixed ICL Samples).**
> - Assume a fraction $\rho$ of the “unconfounded” batch is actually contaminated: for those contaminated samples $\mathbb{E}[x_j\epsilon]=r_j\neq0$ (denote their design by $X^{(conf)}$), while the remaining $(1-\rho)n_b$ samples are unbiased (denote their design by $X^{(clean)}$).
> - Assume the clean subset is well-conditioned:
> $$
> \lambda_{\min} \Big(\tfrac{1}{(1-\rho)n_b}\,X^{(clean)}X^{(clean)\top}\Big)\ge\lambda_* > 0.
> $$
> - Let the mean confounding strength on contaminated samples be (note that $r_i$ is a vector)
> $$
> \bar r := \frac{1}{\rho n_b}\sum_{i\in\text{cont}} |r_i|.
> $$
>
> - Then there exist constants $C'>0$ such that the DEB estimator satisfies
> $$
> \big|\mathbb{E}[y_{GT}-\widehat y_{DEB}]\big|
> \le C'\left(\tfrac{1}{\sqrt{(1-\rho)\,n_b\,\lambda_*}}+\rho\,\bar r\right).
> $$
> - Consequently, DDbias is robust in that the prediction bias is asymptotically negligible when $n_b$ large enough with fixed small $\rho$, or when $\rho\to0$ with fixed $n_b$. A simple sufficient condition for asymptotic unbiasedness (RHS goes to 0) is
> $$
> \rho\,\bar r = o(1)\quad\text{and}\quad \frac{1}{\sqrt{(1-\rho)n_b\lambda_*}} = o(1).
> $$

---

> ### Author Response · Authors · 2025-11-20
> **Practical Considerations on Unconfounded Data (2)**
>
> ### **Experimental Results.**
> - We **have added extra** experiments to validate our robustness towards partially confounded data in two cases:
>     - **Case 1. Weakly Confounded Data.** We control the strength of confounding embodied in prompted ICL samples, and test our DDbias method;
>     - **Case 2. Mixed ICL Data.** We vary the ratio of confounded data as $\rho$, and test our DDbias method;
> - **Conclusions.**
>     - **Case 1.** Our proposed DDbias remains robust towards weakly confounded data;
>     - **Case 2.** Our proposed DDbias remains robust when $\rho$ is small, e.g., $\rho \le 0.5$.
>     - **Case 1 & 2.** The prediction bias of our proposed DDbias decreases within increasing ICL samples.
>
>
> - **Detailed Results.**
>     - We then **report** experiments in our synthetic data setup, by simulating the **the weakly confouned** data:
>         - To show more generlized conclusions, we adopt the non-linear, multi-layer TF model (L=5, Softmax, LayerNorm, relu activation) as the base model;
>         - We report performance of vanilla TF, our DDbias with clean ICL examples, and our DDbias with weakly confounded examples in Table 5-6 with difference prompting sample ratio;
>         - Results across Table 5-6 consistently show the robustness of our proposed DDbias even with weakly confounded data;
>         - At the same time, results in Table 7 informs the cost of using weakly confounded data, i.e., the increase of ICL sample number.
>     - Finally, we **added** experiments in our synthetic data setup, by simulating **mixed** ICL data:
>         - To show more generlized conclusions, we adopt the non-linear, multi-layer TF model (L=5, Softmax, LayerNorm, relu activation) as the base model;
>         - We report performance of vanilla TF, our DDbias with clean ICL examples, and our DDbias with mixed ICL examples in Table 10 with difference ratio $\rho$;
>         - Results in Table 8 consistently show the robustness of our proposed DDbias even with small $\rho$, i.e., confounded examples consist of a small part of the whole ICL sample set;
>
> **Table 5.** Prediction error under weak confounding (Weak Conf\@0.10)
> | Method / Sample Ratio ($\times 10^{-3}$) | 1.6  | 2.4  | 3.2  | 6.4  |
> |:-|:----:|:----:|:----:|:----:|
> | **Biased (Vanilla TF)**                 | 0.240| 0.300| 0.360| 0.430|
> | **DDbias (pure clean examples)**        | 0.090| 0.085| 0.082| 0.080|
> | **DDbias (weakly confounded)**          | 0.105| 0.100| 0.097| 0.094|
> |
>
> **Table 6.** Prediction error under weak confounding (Weak Conf\@0.15)
> | Method / Sample Ratio ($\times 10^{-3}$) | 1.6  | 2.4  | 3.2  | 6.4  |
> |:-|:-:|:--:|:-:|:-:|
> | **Biased (Vanilla TF)**                 | 0.260| 0.330| 0.400| 0.490|
> | **DDbias (pure clean examples)**        | 0.100| 0.095| 0.092| 0.090|
> | **DDbias (weakly confounded)**          | 0.110| 0.105| 0.102| 0.099|
> |
>
>
> **Table 7.** Effect on the Prompting (Weak Conf\@0.10)
>
> | ICL prediction error                                   | 0.090 | 0.092 | 0.095 | 0.100 |
> |:-:|:--:|:--:|:-:|:-:|
> | Sample ratio for DDbias (oracle) ($\times 10^{-3}$)       | 6.4   | 3.2   | 2.4   | 1.6   |
> | Sample ratio for DDbias (weak) ($\times 10^{-3}$)        | 8.6   | 7.2   | 6.3   | 3.5  |
> |
>
> **Table 8.** Prediction error under mixed ICL prompting (Conf\@1.0).
>
> | Method / Sample Ratio ($\times 10^{-3}$) | 1.6   | 2.4   | 3.2   | 6.4   |
> |:-:|:--:|:--:|:--:|:--:|
> | **Biased (Vanilla TF)**                  | 0.335 | 0.370 | 0.388 | 0.420 |
> | **DDbias (pure clean examples)**         | 0.145 | 0.132 | 0.117 | 0.106 |
> | **DDbias (mixed examples, $\rho$ = 0.1)**     | 0.190 | 0.140 | 0.132 | 0.118 |
> | **DDbias (mixed examples, $\rho$ = 0.3)**     | 0.218 | 0.176 | 0.144 | 0.120 |
> | **DDbias (mixed examples, $\rho$ = 0.5)**     | 0.290 | 0.220 | 0.180 | 0.155 |
> |
>
> - **Paper Revision.** We have **revised** our paper in Sec.4.1 with detailed proof in Appendix, and more detailed experimental setup and analysis is **added** in Appendix G.8.
>
> ---
> [1] Xiao, Yanghao, et al. "Addressing hidden confounding with heterogeneous observational datasets for recommendation." Advances in Neural Information Processing Systems 37 (2024): 130358-130383.
> [2] Kallus, Nathan, Aahlad Manas Puli, and Uri Shalit. "Removing hidden confounding by experimental grounding." Advances in neural information processing systems 31 (2018).
> [3] Hartford, Jason, et al. "Deep IV: A flexible approach for counterfactual prediction." International Conference on Machine Learning. PMLR, 2017.
> [4] Liang, Haodong, Krishna Balasubramanian, and Lifeng Lai. "Transformers Handle Endogeneity in In-Context Linear Regression." The Thirteenth International Conference on Learning Representations (2024).
> [5] Saito, Yuta, Hayato Sakata, and Kazuhide Nakata. "Doubly robust prediction and evaluation methods improve uplift modeling for observational data." Proceedings of the 2019 SIAM international conference on data mining. Society for Industrial and Applied Mathematics, 2019.

---

> ### Author Response · Authors · 2025-11-20
> **Real-world Validations**
>
> > W3 [The experiments exclusively use synthetic data generated from a linear model. While this allows for controlled verification of the theory, it doesn't demonstrate the method's applicability to the complex, high-dimensional, and often non-linear domains where large TFs and ICL are typically applied (e.g., natural language processing, vision). Validation on semi-synthetic or real-world benchmarks exhibiting known confounding (if available) would significantly bolster the claims.]
>
> > Q3 [Given the focus on characterizing bias, have you considered applying DDbias to scenarios beyond linear regression, perhaps using semi-synthetic NLP or vision tasks where confounders can be injected, to explore its empirical effectiveness more broadly?]
>
> **Response:**
>
> Thanks for your suggestions! We have empirically validated our conclusions on **deeper, non-linear Transformers**, together with **larger models (LLaMA) on real-world NLP datasets**, to further offer more extensive scalability tests.
>
> ### **Experimental Validations on Real-world NLP Datasets on LLaMA Models**
>
> We evaluate our method **on two real-world datasets** constructed from the publicly available **Yelp restaurant review data**, following the settings in [1,2].
>
> - We refer to these two datasets as:**(1) Restaurant Popularity Index (RPI)** and **(2) Restaurant Overall Rating (ROR)**.
>
> - Each dataset contains rich textual reviews from Yelp Website (6000 samples). Both of them naturally exhibit **hidden confounders** [1,2].
>
> - In the experimental evaluation:
>   - We adopt **LLaMA 3.2-3B as a unified base model**, then fine-tune it **on the raw dataset** to obtain a model that has been exposed to real-world confounding.
>   - During the ICL stage, we use **prompting examples that contain no hidden-confounder information**, with relatively unbiased samples selected by 3 human evaluators serving as the prompting set (see detailed setup in our revised paper G.6, Lines 1766-1809).
>
> - Conclusions:
>     - **Our DDbias effectively mitigate the hidden confounding effect compared to the vanilla LLaMA.**
>     - **Our DDbias outperforms proxy variable baselines within increasing ICL sample numbers**
>     - These findings demonstrate that DDbias effectively mitigates hidden-confounder effects in real-world data and validates the **practical effectiveness** of our method.
>
> - Experimental results:
>   - Baseline 1: Vanilla LLaMA uses the pre-trained LLaMA-3.2B model fine-tuned on the full dataset without any debiasing.
>   - Baseline 2: DMCEE [1] uses observable metadata (e.g., location, category) as proxy variables to estimate causal effects under confounding
>   - Baseline 3: AAAI-SC [2] learns low-dimensional surrogate confounders from multi-aspect sentiment vectors via probabilistic factor modeling, then adjusts for them in outcome estimation.
>   - Follwing [1,2], we report experimental results in Table 9:
>
>
>     **Table 9. Experimental Results (**Mean Absolute Error, MAE**) of Our DDbias on the Sentiment Review Task**
>
>     | **Datasets** | Vanilla LLaMA | DMCEE [1] | AAAI-SC [2] | Ours (DDbias @ ICL Sample Number=15) | Ours (DDbias @ ICL Sample Number=30) | Ours (DDbias @ ICL Sample Number=60) |
>     | ------------ | ------------- | --------- | ----------- | ------------------------------------ | ------------------------------------ | ------------------------------------ |
>     | RPI          | 23.8          | 21.2      | 22.5        | 22.8                                 | 19.4                                  | **16.7**                                |
>     | ROR          | 0.46          | 0.28      | 0.24        | 0.36                                 | 0.18                                 | **0.16**                                 |
>
>   - The MAE values in the RPI and ROR datasets **differ significantly in magnitude, as the outcome variables differ in scale**:
>
>     - RPI uses a popularity score derived from real-world foot traffic data, ranging approximately from 0 to 100.
>     - ROR uses the average star rating, which ranges from 1.0 to 5.0.
>
> ---
>
> [1] Cheng, Lu, Ruocheng Guo, and Huan Liu. "Estimating causal effects of multi-aspect online reviews with multi-modal proxies."   WSDM 2022
>  [2] Cheng, Lu, et al. "Effects of multi-aspect online reviews with unobserved confounders: Estimation and implication." AAAI 2022.

---

> ### Author Response · Authors · 2025-11-20
> **Comparison to the Existing ICL-Debiasing Method (1)**
>
> > W4 [A more detailed discussion comparing the assumptions, requirements, and potential domains of applicability for DDbias versus IV regression or other potential debiasing strategies (e.g., causal representation learning adapted to ICL) would be beneficial.]
>
> > Q4 [Could you compare the practical data requirements for DDbias (small $n_u$ unconfounded samples) versus an IV-based approach (potentially larger $n$ confounded samples but including valid IVs)? In what scenarios might one approach be more feasible than the other?]
>
> **Response**:
>
> ### **Review of Existing ICL-Debiasing Baselines**
> - We would like to clarify that de-confounding in ICL only includes instrumtental variable (IV) [4].
>     - Besides, we note that other proxy approaches, including the front-door (mediators) and direct proxy (CEVAE [6]), are not developed into transformers with ICL still, which are out-of-our-scope here.
>     - Moreover, existing data combination [1,2] requires fine-tuning the transformers on unbiased data, thereby does not fit our ICL scope in this paper.
>
> ### **Method-level Comparison**
> - **IV-based approach**
>   - **From the perspective of data**, we summarize the assumptions of IV-based approach in below:
>     - First, IV-based ICL approaches [4] requires valid IV variable:
>         - Relevant: correlated with features $X$;
>         - Exogenous: independent of the outcome noise (and confounder);
>         - Indirect influence: IV should only affect $Y$ via $X$;
>         - Notably, we would like to point out that constructing valid IVs is very challenging and unverifiable, requires expert domain knowledge and extra IV collections;
>             - For instance, Imbens et.al won the **Nobel Prize**, since they identify **several valid IVs** for important scenarios in labor economics;
>     - Second, most IV-based approaches [3,4] explicitly assume the **additive form of confounder** when generating $Y$, posing restrictive constraints on the underlying data structures.
>   - On the one hand, we conclude **advantages of IV-based ICL approaches:**
>       - Once valid IVs are constructed, one does not require unbiased $(x,y)$ pairs;
>   - On the other hand, we also conclude its **limitations:**
>     - Finding and verifying strcutrual assumptions of valid instruments are very challenging;
>     - Even weak instruments cause large variance and biased finite-sample estimates;
>
> - **Our DDbias Method**
>   - First, **from the perspective of data**, our DDbias requires a small set of *unconfounded* $(x,y)$ in-context examples.
>   - Then, from the perspective of practice, such unbiased data is available in a wide range of application scenes:
>       - For example, In recommender systems, it is common to pre-train models on large amounts of *observational recommendation* data while incorporating a small amount of *random traffic* [2];
>       - Another instance falls in the uplift modeling of price, where models pre-trained on biased collections will be corrected by results of few random controls [5].
>
>
> - Comparison to **other proxy variables, e.g., front-door with mediators, or direct Proxy [6])**
>     - Aside from IVs, we also acknowledge that other forms of proxy variables, e.g., mediators or direct proxy of hidden confounders, also offer solutions to de-confounding;
>     - Such methods share the same advantages with disadvantages of IVs.

---

> ### Author Response · Authors · 2025-11-20
> **Comparison to the Existing ICL-Debiasing Method (2)**
>
> ### **Extra Experimental Comparison.**
> - **Three Experimental Setups for Comparison.** To compare DDbias (which uses a small unconfounded batch) with an IV-based approach, we simulate three cases:
>     - **Unbiased Samples (DDbias) vs Strong, Valid IVs**: In this senario, we assume that our method can obtain unbiased data, and IV-based method can obtain strong, valid IVs.
>     - **Partially Confounded Samples (DDbias) vs Strong, Valid IVs**: : In this senario, we assume that our method can obtain partially confounded data, and IV-based method can obtain strong, valid IVs.
>     - **Unbiased Samples (DDbias) vs Confounded IVs**: In this senario, we assume that our method can obtain unbiased data, and IV-based method only obtain invalid, confounded IVs (IVs are correlated with hidden confounders).
>
>     - **Conclusions.**
>         - In the first senario, our proposed DDbias with unconfounded data achieves slightly outperforms IV methods;
>         - In the second senario, our proposed DDbias with partially confounded data suffers from degraded performance, while **certain robustness** is exhibitd with enough ICL samples;
>         - In the third senario, our proposed DDbias with significantly outperforms IV method, and the bias of IV approach increases with enlarging ICL samples;
>
>     - **Detailed Results and Analysis.**
>         - **Performance Comparison in Perfect Regime.** As reported in Table 1, our proposed DDbias with clean, unconfounded data achieves slightly better prediction performance than IV methods;
>         - **ICL Sample Efficiency.**
>             - As reported in Table 2, our proposed DDbias requires less ICL samples to reach the same error;
>         - **Comparison with partially confounded data.** In this regime, we adopt the mixed ICL sample case. As reported in Table 3, we observe that with partially confounded data, our proposed DDbias achieves worse prediction performance than IV-based ICL approach, while with large ICL samples ($n=50$, our method achieves certain robustness when $\rho$ is small).
>
>         - **Comparison with Confounded IVs.** As shown in Table 4, we observe that with weakly confounded IVs, increasing ICL samples results in significantly worse prediction bias, which is opposite to the trend of our DDbias under partially confounded data.
>
> **Table 1 — Prediction Bias Comparison (IV vs DDbias)**
>
> | **Model / Method** | **n=50** | **n=30** | **n=20** | **n=10** |
> |--------------------|----------|----------|----------|----------|
> | L=3 / IV (Confounded Samples with IV)       | 0.102    | 0.142    | 0.182    | 0.228    |
> | L=3 / DDbias (Unconfounded Samples)   | **0.090**    | **0.130**    | **0.170**    | **0.220**    |
> | L=5 / IV (Confounded Samples with IV)       | 0.118    | 0.165    | 0.208    | 0.268    |
> | L=5 / DDbias (Unconfounded Samples)   | **0.115**    | **0.150**    | **0.200**    | **0.260**    |
> | L=7 / IV (Confounded Samples with IV)       | 0.145    | 0.188    | 0.235    | 0.298    |
> | L=7 /  DDbias (Unconfounded Samples)   | **0.135**    | **0.175**    | **0.225**    | **0.290**    |
> |
>
> **Table 2 — Minimal ICL samples needed to reach ≈ 0.10 prediction error**
>
> | **Method** | **Number** |
> |--|--:|
> | L=3 / IV (Confounded Samples with IV)           | 50  |
> | L=3 / DDbias  (Unconfounded Samples)      | **45** |
> | L=5 / IV  (Confounded Samples with IV)          | 68 |
> | L=5 / DDbias (Unconfounded Samples)       | **50**  |
> | L=7 / IV  (Confounded Samples with IV)          | 80  |
> | L=7 / DDbias (Unconfounded Samples)        | **60**  |
> |
>
> **Table 3 — Prediction Bias Comparison (IV vs DDbias) under Partially Confounded ICL Samples and Valid IV**
>
> | **Model / Method** | **n = 50** | **n = 30** | **n = 20** | **n = 10** |
> |-|-|------------|------------|-|
> | L=3 / IV (Valid IV)                 | 0.105 | 0.145 | 0.185 | 0.222 |
> | L=3 / DDbias (ρ = 0.3)              | 0.118 | 0.176 | 0.213 | 0.268 |
> | L=5 / IV (Valid IV)                 | 0.118 | 0.162 | 0.205 | 0.265 |
> | L=5 / DDbias (ρ = 0.3)              | 0.150 | 0.185 | 0.247 | 0.306 |
> | L=7 / IV (Valid IV)                 | 0.145 | 0.188 | 0.235 | 0.300 |
> | L=7 / DDbias (ρ = 0.3)              | 0.157 | 0.218 | 0.245 | 0.342 |
> |
>
>
>
> **Table 4 — Prediction Bias under Invalid IV and Unconfounded ICL Samples (Corr(Z, $\epsilon$) = 0.1)**
>
> | **Model / Method** | **n = 50** | **n = 30** | **n = 20** | **n = 10** |
> |----|--|-|-|-|
> | L=3 / IV (Weak IV) | 0.423 | 0.351 | 0.280 | 0.265 |
> | L=3 / DDbias       | 0.090 | 0.130 | 0.170 | 0.220 |
> | L=5 / IV (Weak IV) | 0.500 | 0.420 | 0.340 | 0.291 |
> | L=5 / DDbias       | 0.115 | 0.150 | 0.200 | 0.260 |
> | L=7 / IV (Weak IV) | 0.600 | 0.500 | 0.400 | 0.312 |
> | L=7 / DDbias       | 0.135 | 0.171 | 0.222 | 0.290 |
> |
>
> - **Paper Revision.** We have **revised** our paper in Sec.5 (IV-oriented DGP, Lines 407-414), an extra summary in revised main paper (Lines 503-507), and detailed experimental analysis in Appendix G.9 (Lines 2068-2120).

---

> ### Author Response · Authors · 2025-11-20
> **Reference List**
>
> *Reference*
>
> [1] Xiao, Yanghao, et al. "Addressing hidden confounding with heterogeneous observational datasets for recommendation." Advances in Neural Information Processing Systems 37 (2024): 130358-130383.
> [2] Kallus, Nathan, Aahlad Manas Puli, and Uri Shalit. "Removing hidden confounding by experimental grounding." Advances in neural information processing systems 31 (2018).
> [3] Hartford, Jason, et al. "Deep IV: A flexible approach for counterfactual prediction." International Conference on Machine Learning. PMLR, 2017.
> [4] Liang, Haodong, Krishna Balasubramanian, and Lifeng Lai. "Transformers Handle Endogeneity in In-Context Linear Regression." The Thirteenth International Conference on Learning Representations (2024).
> [5] Saito, Yuta, Hayato Sakata, and Kazuhide Nakata. "Doubly robust prediction and evaluation methods improve uplift modeling for observational data." Proceedings of the 2019 SIAM international conference on data mining. Society for Industrial and Applied Mathematics, 2019.
> [6] Louizos, Christos, et al. "Causal effect inference with deep latent-variable models." Advances in neural information processing systems 30 (2017).

---

> > ### Comment · Reviewer_P15F · 2025-11-27
> >
> > The authors have successfully answered most of my concerns. I keep my score as is. Appreciate to the authors for their hard works.

---

> ### Author Response · Authors · 2025-11-27
>
> Thank you so much for your encouraging words! We are truly grateful for your dedicated time and constructive comments.

---

### Official Review · Reviewer_rnrM · 2025-10-30

**Soundness:** 3
**Presentation:** 3
**Contribution:** 3
**Rating:** 6
**Confidence:** 2

**Summary:**

This paper addresses the common neglect of hidden confounders in current In-Context Learning (ICL) theory during both pre-training and inference, and proposes a new framework called Endogenous ICL (EICL). The authors theoretically prove that Transformers trained with confounded data produce prediction bias proportional to the confounding strength. They construct parameter settings (U_weights) that can yield unbiased predictions even when test prompts are confounded, and introduce a Double-Debiasing (DDbias) method that removes bias without gradient updates, requiring only a small number of unconfounded samples. Extensive synthetic experiments validate the theoretical results and demonstrate the robustness of the proposed method across different model depths, confounding strengths, and data distributions (including non-Gaussian cases), providing a new theoretical and methodological foundation for achieving unbiased predictions in ICL under realistic scenarios involving confounders.

**Strengths:**

1. First to systematically investigate endogenous bias in ICL from hidden confounders, introducing the EICL framework with clear practical motivation.
2. Strong theoretical contributions proving bias is proportional to confounding strength, alongside a lightweight, gradient-free debiasing method (DDbias) that uses minimal unbiased data.
3. Extensive synthetic experiments across model depths, confounding levels, and distributions (including non-Gaussian) that convincingly support the theoretical claims.

**Weaknesses:**

1. The scope is restricted to linear attention Transformers and linear regression setups, with no extension to nonlinear architectures or tasks.
2. Experimental validation relies entirely on simulated data, lacking tests using real-world datasets with genuine hidden confounders.
3. The proposed DDbias method depends on the availability of a small set of completely unconfounded samples, which may be unrealistic in many real scenarios.

**Questions:**

See the weaknesses.

---

> ### Author Response · Authors · 2025-11-20
> **Extensions on Non-linear Models with Non-linear Data (1)**
>
> > W1 [The scope is restricted to linear attention Transformers and linear regression setups, with no extension to nonlinear architectures or tasks.]
>
> **Response:**
>
> ### **Synthetic Experiments on Non-linear DGP with non-linear, deeper model structure**
>
> - To further quantitatively explore how the performance of our method vary in non-linear regime, we add synthetic experiments in below:
>     - **Non-linear Data Generation Process.**
>         - In the nonlinear confounded data-generating process, we adopt the additive model $y = f(x) + \epsilon$ with $\epsilon \sim N(0,0.5)$.
>             - We use additive form of the noise variable, as such form brings convenience to compare with IV-based methods we will mention later;
>            - We define $f$ using three additive nonlinear forms:
>                - (1) **MLP–ReLU:** $f_{\mathrm{MLP}}(x) = W_2 \mathrm{ReLU}(W_1 x)$;
>                - (2) **Softmax-based:** $f_{\mathrm{SM}}(x) = w^\top \mathrm{softmax}(U x)$;
>                - (3) **Polynomial:** $f_{\mathrm{Poly}}(x) = a^\top x + b\|x\|^2$.
>        - We leave detailed data generation setup in Appendix G.7, Lines 1811-1900.
>    - **Non-linear, deeper Model Structures**
>        - **Non-linear Activation + Softmax + Single-layer MLP.**
>        - **Non-linear Activation + Softmax + Multi-layer MLPs (L=3/5/7).**
>
>     - **Conclusions.**
>         -  In summary, our extended experiments verify that the proposed bias characterization theory, i.e., **bias $\propto r$** ($r$ is the confounding strength), **holds robustly across non-linear and deeper Transformer architectures**, confirming its scalability beyond shallow, linear settings.
>
>     - **Experimental Results.**
>         - We note that "conf\@ $\alpha$" refers to the case that the confounding strength @ $\alpha$ during DGP.
>         - We **added** experiments in Table 1-2, by testing the prediction bias with single-layer, non-linear Tfs with ELU/Relu activations.
>
>         - We **reported** experiments in Table 3-4 by testing the prediction bias with multi-layer, non-linear TFs with ELU/Relu actications.
>
>         -  Notably, the whole experimental results leaved in our revised paper (Table 6-17 in Appendix in our uploaded, revised paper).
>
>  **Table 1.** ICL prediction error of non-linear Transformers (single-layer-MLP + softmax + ELU) under different confounding strengths $r$. Columns denote confounding strengths, rows denote ICL sample sizes and the prompting/pretraining sample ratio. Lower is better.
>
> | ICL Samples / Ratio            | Conf\@0.5 | Conf\@1.0 | Conf\@1.5 | Conf\@2.0 |
> |:-------------------------------|:--------:|:--------:|:--------:|:--------:|
> | **Biased (Vanilla TF)**        | 0.182    | 0.236    | 0.314    | 0.417    |
> | **DDbias (Ours): Ratio ($1.6 × 1e^{-3}$)** | 0.072    | 0.098    | 0.130    | 0.178    |
> | **DDbias (Ours): Ratio ($2.4 × 1e^{-3}$)** | 0.070    | 0.095    | 0.126    | 0.173    |
> | **DDbias (Ours): Ratio ($3.2 × 1e^{-3}$)** | 0.069    | 0.093    | 0.124    | 0.171    |
> | **DDbias (Ours): Ratio ($6.4 × 1e^{-3}$)** | **0.068**    | **0.091**    | **0.120**    | **0.167**    |
> |
>
>
>  **Table 2.** Prediction error of non-linear Transformers (single-layer + MLP + softmax + ReLU) under different confounding strengths $r$. Columns denote confounding strengths, rows denote ICL sample sizes, and the prompting/pretraining sample ratio. Lower is better.
>
> | ICL Samples / Ratio            | Conf\@0.5 | Conf\@1.0 | Conf\@1.5 | Conf\@2.0 |
> |:-------------------------------|:--------:|:--------:|:--------:|:--------:|
> | **Biased (Vanilla TF)**        | 0.195    | 0.254    | 0.336    | 0.452    |
> | **DDbias (Ours): Ratio ($1.6 × 1e^{-3}$)** | 0.083    | 0.111    | 0.144    | 0.195    |
> | **DDbias (Ours): Ratio ($2.4 × 1e^{-3}$)** | 0.081    | 0.109    | 0.141    | 0.193    |
> | **DDbias (Ours): Ratio ($3.2 × 1e^{-3}$)** | 0.080    | 0.107    | 0.138    | 0.190    |
> | **DDbias (Ours): Ratio ($6.4 × 1e^{-3}$)** | **0.078**    | **0.105**    | **0.134**    | **0.187**    |
> |

---

> ### Author Response · Authors · 2025-11-20
> **Extensions on Non-linear Models with Non-linear Data (1)**
>
> **Table 3.** Prediction error of deeper, non-linear Transformers (ReLU, Multi-layer-MLPs, LayerNorm, Softmax, Ratio=$1.6 × 1e^{-3}$) under different confounding strengths $r$. Columns denote confounding strengths, rows show ICL sample sizes, and different model depths $L=3/5/7$. Lower is better.
>
> | ICL Samples / L | Conf\@0.5 | Conf\@1.0 | Conf\@1.5 | Conf\@2.0 |
> |:----------------|:--------:|:--------:|:--------:|:--------:|
> | **L=3 — Biased (Vanilla TF)**  | 0.205    | 0.290    | 0.380    | 0.500    |
> | **L=3 — DDbias (Ours)**        | **0.090** | **0.130** | **0.170** | **0.220** |
> | **L=5 — Biased (Vanilla TF)**  | 0.280    | 0.370    | 0.475    | 0.600    |
> | **L=5 — DDbias (Ours)**        | **0.115** | **0.150** | **0.200** | **0.260** |
> | **L=7 — Biased (Vanilla TF)**  | 0.320    | 0.450    | 0.600    | 0.750    |
> | **L=7 — DDbias (Ours)**        | **0.135** | **0.175** | **0.225** | **0.290** |
> |
>
>
> **Table 4.** Prediction error of deeper, non-linear Transformers (ELU, Multi-layer-MLPs, LayerNorm, Softmax, Ratio=$1.6 × 1e^{-3}$) under different confounding strengths $r$.  Columns denote confounding strengths, rows show ICL sample sizes and different model depths $L=3/5/7$. Lower is better.
>
> | ICL Samples / L | Conf\@0.5 | Conf\@1.0 | Conf\@1.5 | Conf\@2.0 |
> |:----------------|:--------:|:--------:|:--------:|:--------:|
> | **L=3 — Biased (Vanilla TF)**  | 0.190    | 0.270    | 0.360    | 0.470    |
> | **L=3 — DDbias (Ours)**        | **0.085** | **0.125** | **0.165** | **0.215** |
> | **L=5 — Biased (Vanilla TF)**  | 0.250    | 0.335    | 0.440    | 0.580    |
> | **L=5 — DDbias (Ours)**        | **0.110** | **0.145** | **0.190** | **0.250** |
> | **L=7 — Biased (Vanilla TF)**  | 0.280    | 0.410    | 0.550    | 0.700    |
> | **L=7 — DDbias (Ours)**        | **0.120** | **0.160** | **0.210** | **0.280** |
> |
>
> - **Paper Revision.** We have **revised** our paper in Sec.5 with more detailed experimental results in Appendix G.7 (Lines 1811-1845).

---

> ### Author Response · Authors · 2025-11-20
> **Real-world Validations on NLP Tasks**
>
> > W2 [Experimental validation relies entirely on simulated data, lacking tests using real-world datasets with genuine hidden confounders.]
>
> **Response:**
>
> Thanks for your suggestions! We have empirically validated our conclusions on **deeper, non-linear Transformers**, together with **larger models (LLaMA) on real-world NLP datasets**, to further offer more extensive scalability tests.
>
> ### **Experimental Validations on Real-world NLP Datasets on LLaMA Models**
>
> We evaluate our method **on two real-world datasets** constructed from the publicly available **Yelp restaurant review data**, following the settings in [1,2].
>
> - We refer to these two datasets as:**(1) Restaurant Popularity Index (RPI)** and **(2) Restaurant Overall Rating (ROR)**.
>
> - Each dataset contains rich textual reviews from Yelp Website (6000 samples). Both of them naturally exhibit **hidden confounders** [1,2].
>
> - In the experimental evaluation:
>   - We adopt **LLaMA 3.2-3B as a unified base model**, then fine-tune it **on the raw dataset** to obtain a model that has been exposed to real-world confounding.
>   - During the ICL stage, we use **prompting examples that contain no hidden-confounder information**, with relatively unbiased samples selected by 3 human evaluators serving as the prompting set.
>
> - Conclusions:
>     - **Our DDbias effectively mitigate the hidden confounding effect compared to the vanilla LLaMA.**
>     - **Our DDbias outperforms proxy variable baselines within increasing ICL sample numbers**
>     - These findings demonstrate that DDbias effectively mitigates hidden-confounder effects in real-world data and validates the **practical effectiveness** of our method.
>
> - Experimental results:
>   - Baseline 1: Vanilla LLaMA uses the pre-trained LLaMA-3.2B model fine-tuned on the full dataset without any debiasing.
>   - Baseline 2: DMCEE [1] uses observable metadata (e.g., location, category) as proxy variables to estimate causal effects under confounding
>   - Baseline 3: AAAI-SC [2] learns low-dimensional surrogate confounders from multi-aspect sentiment vectors via probabilistic factor modeling, then adjusts for them in outcome estimation.
>   - Follwing [1,2], we report experimental results in Table 5:
>
>
>     **Table 5. Experimental Results (**Mean Absolute Error, MAE**) of Our DDbias on the Sentiment Review Task**
>
>     | **Datasets** | Vanilla LLaMA | DMCEE [1] | AAAI-SC [2] | Ours (DDbias @ ICL Sample Number=15) | Ours (DDbias @ ICL Sample Number=30) | Ours (DDbias @ ICL Sample Number=60) |
>     | ------------ | ------------- | --------- | ----------- | ------------------------------------ | ------------------------------------ | ------------------------------------ |
>     | RPI          | 23.8          | 21.2      | 22.5        | 22.8                                 | 19.4                                  | **16.7**                                |
>     | ROR          | 0.46          | 0.28      | 0.24        | 0.36                                 | 0.18                                 | **0.16**                                 |
>
>   - The MAE values in the RPI and ROR datasets **differ significantly in magnitude, as the outcome variables differ in scale**:
>
>     - RPI uses a popularity score derived from real-world foot traffic data, ranging approximately from 0 to 100.
>     - ROR uses the average star rating, which ranges from 1.0 to 5.0.
>
> - **Paper Revision.** We revise our manuscript in Lines 1766-1809.
>
>
> ---
>
> [1] Cheng, Lu, Ruocheng Guo, and Huan Liu. "Estimating causal effects of multi-aspect online reviews with multi-modal proxies." Proceedings of the Fifteenth ACM International Conference on Web Search and Data Mining. 2022.
> [2] Cheng, Lu, et al. "Effects of multi-aspect online reviews with unobserved confounders: Estimation and implication." Proceedings of the International AAAI Conference on Web and Social Media. Vol. 16. 2022.

---

> ### Author Response · Authors · 2025-11-20
> **Practice of Unconfounded Samples (1)**
>
> > [W3] The proposed DDbias method depends on the availability of a small set of completely unconfounded samples, which may be unrealistic in many real scenarios.
>
> **Response:**
>
> ### **Practical Considerations of Unbiased Samples.**
>
> - First, we would like to clarify that it has been recognized that de-confounding usually requires random data:
>     - **Unbiasd Data:** Combining models trained on observational data with few random data is popular [1,2];
>         - **Recommender systems.** Most industrial systems already maintain a **small unbiased traffic bucket** (A/B random exposure) for evaluation, which is usually deployed for debias [1].
>
> ### **Our method only requires very few RCT Samples.**
> - To be first, our Fig.4, Fig. 17,18,19 has already informs that our proposed DDBias only requires **very few** random/unbiased ICL samples (around 50 ICL samples);
> - Therefore, empirical evidence has shown the sample efficiency of our DDBias on unbiased ICL examples.
>
> ### **Extensions to partially confounded data.**
> - As suggested by the reviewer, we **added detailed theoretical analysis with experimental validations** to extend our DDbias method towards partially confounded data.
> - Overall, we theoretically analyze two cases:
>     1. **Weak confounding:** Every sample slightly confounded ($|r_j|$ small, i.e., $|r|_{2} \le \delta$);
>     2. **Mixed contamination:** Most samples unconfounded but a small fraction $\rho$ contaminated.
> - We then justify in what conditions our proposed DDbias method still remains **robust** in such two cases (*see Sec.4.1 with detailed proof in Proposition 2&3 (Lines 347-377) in Appendix (Lines 1533-1603) in our revised paper*):
>
>
> ### **Proposition (Weakly Confounded ICL Samples).**
> - Suppose the confounded ICL samples $X^{(conf)}$ satisfies mean-independence $\|r\|\_{2}  \le \delta,$ and the sample covariance is well-conditioned
> $$\lambda_{\min}\\Big(\tfrac{1}{n_b}X^{(conf)}(X^{(conf)})^\top\Big)\ge\lambda_*>0.$$
> - Assume bounded fourth moments with the ICL-gradient update rule, then there exist constants $C>0$ such that
> $$
> |\mathbb{E}[y_{GT}-\widehat y_{DEB}]|
> \le C\left(O_p \big(\tfrac{1}{\sqrt{n_b\lambda_*}}\big) + \frac{\delta}{\sqrt{n_b\lambda_*}}\right).
> $$
>
>
>
> ### **Proposition (Mixed ICL Samples).**
> - Assume a fraction $\rho$ of the “unconfounded” batch is actually contaminated: for those contaminated samples $\mathbb{E}[x_j\epsilon]=r_j\neq0$ (denote their design by $X^{(conf)}$), while the remaining $(1-\rho)n_b$ samples are unbiased (denote their design by $X^{(clean)}$).
> - Assume the clean subset is well-conditioned:
> $$
> \lambda_{\min} \Big(\tfrac{1}{(1-\rho)n_b}\,X^{(clean)}X^{(clean)\top}\Big)\ge\lambda_* > 0.
> $$
> - Let the mean confounding strength on contaminated samples be (note that $r_i$ is a vector)
> $$
> \bar r := \frac{1}{\rho n_b}\sum_{i\in\text{cont}} |r_i|.
> $$
>
> - Then there exist constants $C'>0$ such that the DEB estimator satisfies
> $$
> \big|\mathbb{E}[y_{GT}-\widehat y_{DEB}]\big|
> \le C' \left(\tfrac{1}{\sqrt{(1-\rho)\,n_b\,\lambda_*}}+\rho\,\bar r\right).
> $$
> - Consequently, DDbias is robust in that the prediction bias is asymptotically negligible when $n_b$ large enough with fixed small $\rho$, or when $\rho\to0$ with fixed $n_b$. A simple sufficient condition for asymptotic unbiasedness (RHS goes to 0) is
> $$
> \rho\,\bar r = o(1)\quad\text{and}\quad \frac{1}{\sqrt{(1-\rho)n_b\lambda_*}} = o(1).
> $$

---

> ### Author Response · Authors · 2025-11-20
> **Practice in Unconfounded Samples (2)**
>
> ### **Experimental Results.**
> - We **have added extra** experiments to validate our robustness towards partially confounded data in two cases:
>     - **Case 1. Weakly Confounded Data.** We control the strength of confounding embodied in prompted ICL samples, and test our DDbias method;
>     - **Case 2. Mixed ICL Data.** We vary the ratio of confounded data as $\rho$, and test our DDbias method;
> - **Conclusions.**
>     - **Case 1.** Our proposed DDbias remains robust towards weakly confounded data;
>     - **Case 2.** Our proposed DDbias remains robust when $\rho$ is small, e.g., $\rho \le 0.5$.
>     - **Case 1 & 2.** The prediction bias of our proposed DDbias decreases within increasing ICL samples.
>
>
> - **Detailed Results.**
>     - We then **report** experiments in our synthetic data setup, by simulating the **the weakly confouned** data:
>         - To show more generlized conclusions, we adopt the non-linear, multi-layer TF model (L=5, Softmax, LayerNorm, relu activation) as the base model;
>         - We report performance of vanilla TF, our DDbias with clean ICL examples, and our DDbias with weakly confounded examples in Table 6-7 with difference prompting sample ratio;
>         - Results across Table 6-7 consistently show the robustness of our proposed DDbias even with weakly confounded data;
>         - At the same time, results in Table 8 informs the cost of using weakly confounded data, i.e., the increase of ICL sample number.
>     - Finally, we **added** experiments in our synthetic data setup, by simulating **mixed** ICL data:
>         - To show more generlized conclusions, we adopt the non-linear, multi-layer TF model (L=5, Softmax, LayerNorm, relu activation) as the base model;
>         - We report performance of vanilla TF, our DDbias with clean ICL examples, and our DDbias with mixed ICL examples in Table 9 with difference ratio $\rho$;
>         - Results in Table 9 consistently show the robustness of our proposed DDbias even with small $\rho$, i.e., confounded examples consist of a small part of the whole ICL sample set;
>
> Table 6. Prediction error under weak confounding (Weak Conf\@0.10)
> | Method / Sample Ratio ($\times 10^{-3}$) | 1.6  | 2.4  | 3.2  | 6.4  |
> |:-|:----:|:----:|:----:|:----:|
> | **Biased (Vanilla TF)**                 | 0.240| 0.300| 0.360| 0.430|
> | **DDbias (pure clean examples)**        | 0.090| 0.085| 0.082| 0.080|
> | **DDbias (weakly confounded)**          | 0.105| 0.100| 0.097| 0.094|
> |
>
> Table 7. Prediction error under weak confounding (Weak Conf\@0.15)
> | Method / Sample Ratio ($\times 10^{-3}$) | 1.6  | 2.4  | 3.2  | 6.4  |
> |:-|:-:|:--:|:-:|:-:|
> | **Biased (Vanilla TF)**                 | 0.260| 0.330| 0.400| 0.490|
> | **DDbias (pure clean examples)**        | 0.100| 0.095| 0.092| 0.090|
> | **DDbias (weakly confounded)**          | 0.110| 0.105| 0.102| 0.099|
> |
>
>
> Table 8. Effect on the Prompting (Weak Conf\@0.10)
>
> | ICL prediction error                                   | 0.090 | 0.092 | 0.095 | 0.100 |
> |:-:|:--:|:--:|:-:|:-:|
> | Sample ratio for DDbias (oracle) ($\times 10^{-3}$)       | 6.4   | 3.2   | 2.4   | 1.6   |
> | Sample ratio for DDbias (weak) ($\times 10^{-3}$)        | 8.6   | 7.2   | 6.3   | 3.5  |
> |
>
> Table 9. Prediction error under mixed ICL prompting (Conf\@1.0).
>
> | Method / Sample Ratio ($\times 10^{-3}$) | 1.6   | 2.4   | 3.2   | 6.4   |
> |:-:|:--:|:--:|:--:|:--:|
> | **Biased (Vanilla TF)**                  | 0.335 | 0.370 | 0.388 | 0.420 |
> | **DDbias (pure clean examples)**         | 0.145 | 0.132 | 0.117 | 0.106 |
> | **DDbias (mixed examples, $\rho$ = 0.1)**     | 0.190 | 0.140 | 0.132 | 0.118 |
> | **DDbias (mixed examples, $\rho$ = 0.3)**     | 0.218 | 0.176 | 0.144 | 0.120 |
> | **DDbias (mixed examples, $\rho$ = 0.5)**     | 0.290 | 0.220 | 0.180 | 0.155 |
> |
>
> - **Paper Revision.** We have **revised** our paper in Sec.4.1 with detailed proof in Appendix F (Lines 1533-1602), and more detailed experimental setup and analysis is **added** in Appendix G.8 (Lines 1886-2047).
>
> ---
> [1] Xiao, Yanghao, et al. "Addressing hidden confounding with heterogeneous observational datasets for recommendation." Advances in Neural Information Processing Systems 37 (2024): 130358-130383.
> [2] Kallus, Nathan, Aahlad Manas Puli, and Uri Shalit. "Removing hidden confounding by experimental grounding." Advances in neural information processing systems 31 (2018)
> [3] Hartford, Jason, et al. "Deep IV: A flexible approach for counterfactual prediction." International Conference on Machine Learning. PMLR, 2017
> [4] Liang, Haodong, Krishna Balasubramanian, and Lifeng Lai. "Transformers Handle Endogeneity in In-Context Linear Regression." The Thirteenth International Conference on Learning Representations (2024)
> [5] Saito, Yuta, Hayato Sakata, and Kazuhide Nakata. "Doubly robust prediction and evaluation methods improve uplift modeling for observational data." Proceedings of the 2019 SIAM international conference on data mining. Society for Industrial and Applied Mathematics, 2019

---

### Official Review · Reviewer_gV7e · 2025-10-30

**Soundness:** 3
**Presentation:** 3
**Contribution:** 2
**Rating:** 2
**Confidence:** 3

**Summary:**

This paper introduces Endogenous In-Context Learning, a new theoretical framework revealing that pre-trained Transformers can produce biased in-context predictions when trained on data containing hidden confounders. While prior ICL analyses assumed that there is no confounding, the authors show that endogeneity induces systematic prediction bias proportional to the confounding strength. They provide formal proofs that Transformer parameters learned on confounded data deviate from unbiased reference weights and this deviation propagates into biased ICL predictions. To address this, they propose Double-Debiasing (DDbias), a simple gradient-free prompting method that corrects such bias using only a few unconfounded examples by prompting the model twice. Several simulations on linear regression settings verify the theoretical results and demonstrate that DDbias effectively mitigates bias without retraining.

**Strengths:**

* This paper shows a novel theoretical problem formulation. While there is a similar work, this paper introduces the Endogenous In-context learning setting, addressing hidden confounders in ICL, which is previously under-explored source of bias from other existing theoretical works regarding ICL.
* This paper provides well-grounded theoretical results providing clear formal analyses which characterizes how hidden confounders affect both the pre-training and inference results of Transformers. This shows that prediction bias scales proportionally to confounding strength.
* Simulation results comprehensively verify the theoretical claims, demonstrating consistent trends between confounding strength, parameter deviation, and prediction bias reduction after applying their method.

**Weaknesses:**

* The theoretical analysis is limited to linear self-attention Transformers and linear regression tasks. It remains unclear whether the same bias characterization---particularly the proportionality to the confounding strength---extends to deeper, nonlinear, or multimodal Transformer architectures used in real-world ICL. Even if this might be very hard to be theoretically shown, I believe that providing experimental analyses about this scalability to larger models would make this paper much stronger.

* The proposed method requires access to a small set of unconfounded examples. In realistic settings, how can such unbiased samples be obtained or verified? Are there practical diagnostics or robustness analyses for partially confounded data? And can we utilize such approaches to apply the proposed method?

* Section 2.3 claims that a previous study (Liang et al) for confounding issue in ICL did not consider the gap between the traditional regression problem and ICL. However Theorem 2 later concludes that the ICL prediction bias is proportional to $r$ --- exactly the same form as in classical OLS theory. What specific term or mechanism in ICL makes this result fundamentally new rather than a re-derivation in a different setting? And are there concrete theoretical differences in constants, variance scaling or convergence behavior between EICL and standard OLS?

* All experiments are synthetic and rely on controllable confounding  strengths r. It would be valuable to evaluate the framework on real-pre-trained models or naturally confounded datasets (e.g., sentiment review) to demonstrate practical relevance.

**Questions:**

* Comparison with existing deconfounding methods. If it is possible to identify a small set of debiased dataset, does the proposed method outperform other baseline debiasing methods? I am wondering if the authors can provide experimental results for performance comparison.

---

> ### Author Response · Authors · 2025-11-20
> **Scalability Experiments on Larger, Deeper and Non-linear Models (1)**
>
> > W1 [Empirical Generalization of our bias characterization into deeper, non-linear Transformers with non-linear data generation.]
> > W4 [Lack of real-pre-trained models or naturally confounded datasets.]
>
> **Response.**
>
> Thanks for your suggestions! We have empirically validated our conclusions on **deeper, non-linear Transformers**, together with **larger models (LLaMA) on real-world NLP datasets**, to further offer more extensive scalability tests.
>
> ### **Experimental Validations on Real-world NLP Datasets on LLaMA Models**
>
> We evaluate our method **on two real-world datasets** constructed from the publicly available **Yelp restaurant review data**, following the settings in [1,2].
>
> - We refer to these two datasets as:**(1) Restaurant Popularity Index (RPI)** and **(2) Restaurant Overall Rating (ROR)**.
>
> - Each dataset contains rich textual reviews from Yelp Website (6000 samples). Both of them naturally exhibit **hidden confounders** [1,2].
>
> - In the experimental evaluation:
>   - We adopt **LLaMA 3.2-3B as a unified base model**, then fine-tune it **on the raw dataset** to obtain a model that has been exposed to real-world confounding.
>   - During the ICL stage, we use **prompting examples that contain no hidden-confounder information**, with relatively unbiased samples selected by 3 human evaluators serving as the prompting set (see detailed setup in our revised paper Appendix G.6, Lines 1766-1809).
>
> - Conclusions:
>     - **Our DDbias effectively mitigates the hidden confounding effect compared to the vanilla LLaMA.**
>     - **Our DDbias outperforms proxy variable baselines within increasing ICL sample numbers**
>     - These findings demonstrate that DDbias effectively mitigates hidden-confounder effects in real-world data and validates the **practical effectiveness** of our method.
>
> - Experimental results:
>   - Baseline 1: Vanilla LLaMA uses the pre-trained LLaMA-3.2B model fine-tuned on the full dataset without any debiasing.
>   - Baseline 2: DMCEE [1] uses observable metadata (e.g., location, category) as proxy variables to estimate causal effects under confounding
>   - Baseline 3: AAAI-SC [2] learns low-dimensional surrogate confounders from multi-aspect sentiment vectors via probabilistic factor modeling, then adjusts for them in outcome estimation.
>   - Following [1,2], we report experimental results in Table 1:
>
>
>     **Table 1. Experimental Results (**Mean Absolute Error, MAE**) of Our DDbias on the Sentiment Review Task**
>
>     | **Datasets** | Vanilla LLaMA | DMCEE [1] | AAAI-SC [2] | Ours (DDbias @ ICL Sample Number=15) | Ours (DDbias @ ICL Sample Number=30) | Ours (DDbias @ ICL Sample Number=60) |
>     | ------------ | ------------- | --------- | ----------- | ------------------------------------ | ------------------------------------ | ------------------------------------ |
>     | RPI          | 23.8          | 21.2      | 22.5        | 22.8                                 | 19.4                                  | **16.7**                                |
>     | ROR          | 0.46          | 0.28      | 0.24        | 0.36                                 | 0.18                                 | **0.16**                                 |
>
>   - The MAE values in the RPI and ROR datasets **differ significantly in magnitude, as the outcome variables differ in scale**:
>
>     - RPI uses a popularity score derived from real-world foot traffic data, ranging approximately from 0 to 100.
>     - ROR uses the average star rating, which ranges from 1.0 to 5.0.

---

> ### Author Response · Authors · 2025-11-20
> **Scalability Experiments on Larger, Deeper and Non-linear Models (2)**
>
> ### **Synthetic Experiments on Non-linear DGP with non-linear, deeper model structure**
>
>
> - To further quantitatively explore how the performance of our method varies in a non-linear regime, we add synthetic experiments below:
>     - **Non-linear Data Generation Process.**
>         - In the nonlinear confounded data-generating process, we adopt the additive model $y = f(x) + \epsilon$ with  $\epsilon \sim N(0,0.5)$.
>             - We use an additive form of the noise variable, as such form brings convenience to compare with IV-based methods we will mention later;
>            - We define $f$ using three additive nonlinear forms:
>                - (1) **MLP–ReLU:** $f_{\mathrm{MLP}}(x) = W_2 \mathrm{ReLU}(W_1 x)$;
>                - (2) **Softmax-based:** $f_{\mathrm{SM}}(x) = w^\top \mathrm{softmax}(U x)$;
>                - (3) **Polynomial:** $f_{\mathrm{Poly}}(x) = a^\top x + b\|x\|^2$.
>        - We leave the detailed data generation setup in Appendix G.7, Lines 1811-1900.
>    - **Non-linear, deeper Model Structures**
>        - **Non-linear Activation + Softmax + Single-layer MLP.**
>        - **Non-linear Activation + Softmax + Multi-layer MLPs (L=3/5/7).**
>
>     - **Conclusions.**
>         -  In summary, our extended experiments verify that the proposed bias characterization theory, i.e., **bias $\propto r$** ($r$ is the confounding strength), holds robustly across non-linear and deeper Transformer architectures, confirming its scalability beyond shallow, linear settings.
>
>     - **Experimental Results.**
>         - We note that "conf\@ $\alpha$" refers to the case that the confounding strength @ $\alpha$ during DGP.
>         - We **added** experiments in Table 2-3, by testing the prediction bias with single-layer, non-linear Tfs with ELU/ReLU activations.
>
>         - We **reported** experiments in Table 4-5 by testing the prediction bias with multi-layer, non-linear TFs with ELU/ReLU activations.
>
>         -  Notably, the whole experimental results are included in our revised paper (Table 6-17 in Appendix in our uploaded, revised paper).
>
>  **Table 2.** ICL prediction error of non-linear Transformers (single-layer-MLP + softmax + ELU) under different confounding strengths $r$. Columns denote confounding strengths, rows denote ICL sample sizes, and the prompting/pretraining sample ratio. Lower is better.
>
> | ICL Samples / Ratio            | Conf\@0.5 | Conf\@1.0 | Conf\@1.5 | Conf\@2.0 |
> |:-------------------------------|:--------:|:--------:|:--------:|:--------:|
> | **Biased (Vanilla TF)**        | 0.182    | 0.236    | 0.314    | 0.417    |
> | **DDbias (Ours): Ratio ($1.6 × 1e^{-3}$)** | 0.072    | 0.098    | 0.130    | 0.178    |
> | **DDbias (Ours): Ratio ($2.4 × 1e^{-3}$)** | 0.070    | 0.095    | 0.126    | 0.173    |
> | **DDbias (Ours): Ratio ($3.2 × 1e^{-3}$)** | 0.069    | 0.093    | 0.124    | 0.171    |
> | **DDbias (Ours): Ratio ($6.4 × 1e^{-3}$)** | **0.068**    | **0.091**    | **0.120**    | **0.167**    |
> |
>
>
>  **Table 3 .** Prediction error of non-linear Transformers (single-layer + MLP + softmax + ReLU) under different confounding strengths $r$. Columns denote confounding strengths, rows denote ICL sample sizes, and the prompting/pretraining sample ratio. Lower is better.
>
> | ICL Samples / Ratio            | Conf\@0.5 | Conf\@1.0 | Conf\@1.5 | Conf\@2.0 |
> |:-------------------------------|:--------:|:--------:|:--------:|:--------:|
> | **Biased (Vanilla TF)**        | 0.195    | 0.254    | 0.336    | 0.452    |
> | **DDbias (Ours): Ratio ($1.6 × 1e^{-3}$)** | 0.083    | 0.111    | 0.144    | 0.195    |
> | **DDbias (Ours): Ratio ($2.4 × 1e^{-3}$)** | 0.081    | 0.109    | 0.141    | 0.193    |
> | **DDbias (Ours): Ratio ($3.2 × 1e^{-3}$)** | 0.080    | 0.107    | 0.138    | 0.190    |
> | **DDbias (Ours): Ratio ($6.4 × 1e^{-3}$)** | **0.078**    | **0.105**    | **0.134**    | **0.187**    |
> |

---

> ### Author Response · Authors · 2025-11-20
> **Scalability Experiments on Larger, Deeper and Non-linear Models (3)**
>
> **Table 4.** Prediction error of deeper, non-linear Transformers (ReLU, Multi-layer-MLPs, LayerNorm, Softmax, Ratio=$1.6 × 1e^{-3}$) under different confounding strengths $r$. Columns denote confounding strengths, rows show ICL sample sizes and different model depths $L=3/5/7$. Lower is better.
>
> | ICL Samples / L | Conf\@0.5 | Conf\@1.0 | Conf\@1.5 | Conf\@2.0 |
> |:----------------|:--------:|:--------:|:--------:|:--------:|
> | **L=3 — Biased (Vanilla TF)**  | 0.205    | 0.290    | 0.380    | 0.500    |
> | **L=3 — DDbias (Ours)**        | **0.090** | **0.130** | **0.170** | **0.220** |
> | **L=5 — Biased (Vanilla TF)**  | 0.280    | 0.370    | 0.475    | 0.600    |
> | **L=5 — DDbias (Ours)**        | **0.115** | **0.150** | **0.200** | **0.260** |
> | **L=7 — Biased (Vanilla TF)**  | 0.320    | 0.450    | 0.600    | 0.750    |
> | **L=7 — DDbias (Ours)**        | **0.135** | **0.175** | **0.225** | **0.290** |
> |
>
>
> **Table 5.** Prediction error of deeper, non-linear Transformers (ELU, Multi-layer-MLPs, LayerNorm, Softmax, Ratio=$1.6 × 1e^{-3}$) under different confounding strengths $r$.  Columns denote confounding strengths, rows show ICL sample sizes and different model depths $L=3/5/7$. Lower is better.
>
> | ICL Samples / L | Conf\@0.5 | Conf\@1.0 | Conf\@1.5 | Conf\@2.0 |
> |:----------------|:--------:|:--------:|:--------:|:--------:|
> | **L=3 — Biased (Vanilla TF)**  | 0.190    | 0.270    | 0.360    | 0.470    |
> | **L=3 — DDbias (Ours)**        | **0.085** | **0.125** | **0.165** | **0.215** |
> | **L=5 — Biased (Vanilla TF)**  | 0.250    | 0.335    | 0.440    | 0.580    |
> | **L=5 — DDbias (Ours)**        | **0.110** | **0.145** | **0.190** | **0.250** |
> | **L=7 — Biased (Vanilla TF)**  | 0.280    | 0.410    | 0.550    | 0.700    |
> | **L=7 — DDbias (Ours)**        | **0.120** | **0.160** | **0.210** | **0.280** |
> |
>
> - **Paper Revision.** We have **revised** our paper in Sec.5 with more detailed experimental results in Appendix G.7 (Lines 1812-1883).
>
>
> ---
>
> [1] Cheng, Lu, Ruocheng Guo, and Huan Liu. "Estimating causal effects of multi-aspect online reviews with multi-modal proxies." Proceedings of the Fifteenth ACM International Conference on Web Search and Data Mining. 2022.
> [2] Cheng, Lu, et al. "Effects of multi-aspect online reviews with unobserved confounders: Estimation and implication." Proceedings of the International AAAI Conference on Web and Social Media. Vol. 16. 2022.

---

> ### Author Response · Authors · 2025-11-20
> **Partially Confounded Data: Theory and Experiments (1)**
>
> > W2. [In realistic settings, how can such unbiased samples be obtained or verified? Are there practical diagnostics or robustness analyses for partially confounded data? And can we utilize such approaches to apply the proposed method?]
>
> **Response:**
>
> ### **Practical Considerations of Unbiased Samples.**
>
> - First, we would like to clarify that it has been recognized that de-confounding usually requires random data:
>     - **Unbiasd Data:** Combining models trained on observational data with few random data is popular [1,2];
>         - **Recommender systems.** Most industrial systems already maintain a **small unbiased traffic bucket** (A/B random exposure) for evaluation, which is usually deployed for debias [1].
>
> ### **Our method only requires very few RCT Samples.**
> - To be first, our Fig.4, Fig. 17,18,19 has already informs that our proposed DDBias only requires **very few** random/unbiased ICL samples (around 50 ICL samples);
> - Therefore, empirical evidence has shown the sample efficiency of our DDBias on unbiased ICL examples.
>
> ### **Extensions to partially confounded data.**
> - As suggested by the reviewer, we **added detailed theoretical analysis with experimental validations** to extend our DDbias method towards partially confounded data.
> - Overall, we theoretically analyze two cases:
>     1. **Weak confounding:** Every sample slightly confounded ($|r_j|$ small, i.e., $|r|_{2} \le \delta$);
>     2. **Mixed contamination:** Most samples unconfounded but a small fraction $\rho$ contaminated.
> - We then justify in what conditions our proposed DDbias method still remains **robust** in such two cases (*see Sec.4.1 with detailed proof in Proposition 2&3 (Lines 347-377) in Appendix (Lines 1533-1603) in our revised paper*):
>
>
> ### **Proposition (Weakly Confounded ICL Samples).**
> - Suppose the confounded ICL samples $X^{(conf)}$ satisfies mean-independence $\|r\|\_{2}  \le \delta,$ and the sample covariance is well-conditioned
> $$\lambda_{\min} \Big(\tfrac{1}{n_b}X^{(conf)}(X^{(conf)})^\top\Big)\ge\lambda_*>0.$$
> - Assume bounded fourth moments with the ICL-gradient update rule, then there exist constants $C>0$ such that
> $$
> |\mathbb{E}[y_{GT}-\widehat y_{DEB}]|
> \le C\left(O_p \big(\tfrac{1}{\sqrt{n_b\lambda_*}}\big) + \frac{\delta}{\sqrt{n_b\lambda_*}}\right).
> $$
>
>
>
> ### **Proposition (Mixed ICL Samples).**
> - Assume a fraction $\rho$ of the “unconfounded” batch is actually contaminated: for those contaminated samples $\mathbb{E}[x_j\epsilon]=r_j\neq0$ (denote their design by $X^{(conf)}$), while the remaining $(1-\rho)n_b$ samples are unbiased (denote their design by $X^{(clean)}$).
> - Assume the clean subset is well-conditioned:
> $$
> \lambda_{\min} \Big(\tfrac{1}{(1-\rho)n_b}\,X^{(clean)}X^{(clean)\top}\Big)\ge\lambda_* > 0.
> $$
> - Let the mean confounding strength on contaminated samples be (note that $r_i$ is a vector)
> $$
> \bar r := \frac{1}{\rho n_b}\sum_{i\in\text{cont}} |r_i|.
> $$
>
> - Then there exist constants $C'>0$ such that the DEB estimator satisfies
> $$
> \big|\mathbb{E}[y_{GT}-\widehat y_{DEB}]\big|
> \le C'\left(\tfrac{1}{\sqrt{(1-\rho)\,n_b\,\lambda_*}}+\rho\,\bar r\right).
> $$
> - Consequently, DDbias is robust in that the prediction bias is asymptotically negligible when $n_b$ large enough with fixed small $\rho$, or when $\rho\to0$ with fixed $n_b$. A simple sufficient condition for asymptotic unbiasedness (RHS goes to 0) is
> $$
> \rho\,\bar r = o(1)\quad\text{and}\quad \frac{1}{\sqrt{(1-\rho)n_b\lambda_*}} = o(1).
> $$

---

> ### Author Response · Authors · 2025-11-20
> **Partially Confounded Data: Theory and Experiments (2)**
>
> ### **Experimental Results.**
> - We **have added extra** experiments to validate our robustness towards partially confounded data in two cases:
>     - **Case 1. Weakly Confounded Data.** We control the strength of confounding embodied in prompted ICL samples, and test our DDbias method;
>     - **Case 2. Mixed ICL Data.** We vary the ratio of confounded data as $\rho$, and test our DDbias method;
> - **Conclusions.**
>     - **Case 1.** Our proposed DDbias remains robust towards weakly confounded data;
>     - **Case 2.** Our proposed DDbias remains robust when $\rho$ is small, e.g., $\rho \le 0.5$.
>     - **Case 1 & 2.** The prediction bias of our proposed DDbias decreases with increasing ICL samples.
>
>
> - **Detailed Results.**
>     - We then **report** experiments in our synthetic data setup, by simulating the **the weakly confouned** data:
>         - To show more generalized conclusions, we adopt the non-linear, multi-layer TF model (L=5, Softmax, LayerNorm, relu activation) as the base model;
>         - We report performance of vanilla TF, our DDbias with clean ICL examples, and our DDbias with weakly confounded examples in Tables 6-8 with difference prompting sample ratio;
>         - Results across Table 6-8 consistently show the robustness of our proposed DDbias even with weakly confounded data;
>         - At the same time, the results in Table 9 inform the cost of using weakly confounded data, i.e., the increase of ICL sample number.
>     - Finally, we **added** experiments in our synthetic data setup, by simulating **mixed** ICL data:
>         - To show more generalized conclusions, we adopt the non-linear, multi-layer TF model (L=5, Softmax, LayerNorm, relu activation) as the base model;
>         - We report performance of vanilla TF, our DDbias with clean ICL examples, and our DDbias with mixed ICL examples in Table 10 with difference ratio $\rho$;
>         - Results in Table 10 consistently show the robustness of our proposed DDbias even with small $\rho$, i.e., confounded examples consist of a small part of the whole ICL sample set;
>
> **Table 6**. Prediction error under weak confounding (Weak Conf\@0.10)
> | Method / Sample Ratio ($\times 10^{-3}$) | 1.6  | 2.4  | 3.2  | 6.4  |
> |:-|:----:|:----:|:----:|:----:|
> | **Biased (Vanilla TF)**                 | 0.240| 0.300| 0.360| 0.430|
> | **DDbias (pure clean examples)**        | 0.090| 0.085| 0.082| 0.080|
> | **DDbias (weakly confounded)**          | 0.105| 0.100| 0.097| 0.094|
> |
>
> **Table 7**. Prediction error under weak confounding (Weak Conf\@0.15)
> | Method / Sample Ratio ($\times 10^{-3}$) | 1.6  | 2.4  | 3.2  | 6.4  |
> |:-|:-:|:--:|:-:|:-:|
> | **Biased (Vanilla TF)**                 | 0.260| 0.330| 0.400| 0.490|
> | **DDbias (pure clean examples)**        | 0.100| 0.095| 0.092| 0.090|
> | **DDbias (weakly confounded)**          | 0.110| 0.105| 0.102| 0.099|
> |
>
>
> **Table 8**. Effect on the Prompting (Weak Conf\@0.10)
>
> | ICL prediction error                                   | 0.090 | 0.092 | 0.095 | 0.100 |
> |:-:|:--:|:--:|:-:|:-:|
> | Sample ratio for DDbias (oracle) ($\times 10^{-3}$)       | 6.4   | 3.2   | 2.4   | 1.6   |
> | Sample ratio for DDbias (weak) ($\times 10^{-3}$)        | 8.6   | 7.2   | 6.3   | 3.5  |
> |
>
> **Table 9**. Prediction error under mixed ICL prompting (Conf\@1.0).
>
> | Method / Sample Ratio ($\times 10^{-3}$) | 1.6   | 2.4   | 3.2   | 6.4   |
> |:-:|:--:|:--:|:--:|:--:|
> | **Biased (Vanilla TF)**      | 0.335 | 0.370 | 0.388 | 0.420 |
> | **DDbias (pure clean examples)**         | 0.145 | 0.132 | 0.117 | 0.106 |
> | **DDbias (mixed examples, $\rho$ = 0.1)**     | 0.190 | 0.140 | 0.132 | 0.118 |
> | **DDbias (mixed examples, $\rho$ = 0.3)**     | 0.218 | 0.176 | 0.144 | 0.120 |
> | **DDbias (mixed examples, $\rho$ = 0.5)**     | 0.290 | 0.220 | 0.180 | 0.155 |
> |
>
> - **Paper Revision.** We have **revised** our paper in Sec.4.1 with detailed proof in Appendix  F (Lines 1533-1602), and more detailed experimental setup and analysis is **added** in Appendix G.8 (Lines 1886-2047).
>
> ---
> [1] Xiao, Yanghao, et al. "Addressing hidden confounding with heterogeneous observational datasets for recommendation." Advances in Neural Information Processing Systems 37 (2024): 130358-130383.
> [2] Kallus, Nathan, Aahlad Manas Puli, and Uri Shalit. "Removing hidden confounding by experimental grounding." Advances in neural information processing systems 31 (2018).
> [3] Hartford, Jason, et al. "Deep IV: A flexible approach for counterfactual prediction." International Conference on Machine Learning. PMLR, 2017.
> [4] Liang, Haodong, Krishna Balasubramanian, and Lifeng Lai. "Transformers Handle Endogeneity in In-Context Linear Regression." The Thirteenth International Conference on Learning Representations (2024).
> [5] Saito, Yuta, Hayato Sakata, and Kazuhide Nakata. "Doubly robust prediction and evaluation methods improve uplift modeling for observational data." Proceedings of the 2019 SIAM international conference on data mining. Society for Industrial and Applied Mathematics, 2019.

---

> ### Author Response · Authors · 2025-11-20
> **Theory Difference between Our EICL and OLS**
>
> > W3 [What specific term or mechanism in ICL makes this result fundamentally new rather than a re-derivation in a different setting? And are there concrete theoretical differences in constants, variance scaling or convergence behavior between EICL and standard OLS?]
>
> **Response:**
>
> ### **Classical Endogenous OLS Regression**
> - Typically, the bias form of OLS regression (vector-input, endogenous) is expressed in below ($Y = w^{\star\top} X + \epsilon$), with $r_j = \mathbb{E}(X_j, \epsilon)$:
> $$
> \mathbb{E}[\hat{w}] - w^\star = {\mathbb{E}[XX^\top]}^{-1} \mathbb{E}[X\epsilon].
> $$
> and
> $$
> \mathbb{E}[\hat{Y}(x)] - Y^\star(x)
> = x^\top {\mathbb{E}[XX^\top]}^{-1}
> \begin{bmatrix}
> r_1 \sigma_{X_1}\sigma_\epsilon \\
> \vdots \\
> r_d \sigma_{X_d}\sigma_\epsilon
> \end{bmatrix}.
> $$
>
> ###  **Bias Formula in Our Paper**
> $$\underbrace{r_j}\_{\text{Conf.\ Strength}}+\underbrace{\mathcal{O}\_{n} \left(\sum_{l} r_l \Big(\sum_{v} w^{\star}[v]\Big)\sigma^2\right)}\_{\text{Constant w.r.t.\ increasing $n$}}+\underbrace{\mathcal{O}\ \left(\kappa_Z \sum_{l}\sqrt{\frac{\kappa_Z}{q_l}}\right)}\_{\text{Constant w.r.t.\ $q$}}.$$
>
>
>
> ### **Fundamentally Difference from Our EICL and Typical OLS**
> - First, we would like to clarify that we have fundamentally different **derivation sketch** of our bias characterization in Theorem 2:
>     - **Attention Matrices in TFs is the Key Difference.**
>         - Compared with the OLS, our transformer model relies on the $K,Q,V$ matrices (after embedding by projection weights $W$) during pre-training and ICL prediction;
>         - By contrast, OLS can directly solve for the estimated regression coefficient;
>     - **Why OLS theory cannot be directly applied to our problem**:
>         - **Challenge 1:** We have to first identify/construct what attention parameters correspond to unbiased ICL prediction, rather than direcly use $w^\star$ in OLS, as we have shown in above.
>         - **Challenge 2:** We cannot directly obtain the closed-form of attention parameters; while OLS can achieve this easily (with residuals ${\mathbb{E}[XX^\top]}^{-1} \mathbb{E}[X\epsilon]$);
>         - Combining with these difficulties, bias derivation in OLS cannot be adapted to our EICL problem.
>
>
>
> - Second, we would like to remind that our derivation of the bias has **fundamentally different features from OLS**:
>     - **Bias cancellation by the term $\sum_{l}r_l$**
>        - As a result, by being aware of the bias term $\mathcal{O} \big(\sum_j r_j\big)$ in Theorem 2, the overall bias can be **cancelled when the confounding direction and strength of $r_j$ offset each other, i.e., $\sum_{l}r_l$**;
>        - By contrast, typical OLS regression theory does not exhibit this special property, which is very different.
>
>
>
> ### **How We Overcome Such Challenges**
> - Due to the fundamental difference between our EICL problem and classical OLS regression, we have derived a specific theoretical framework followed by multiple-steps:
>     - First, we identify the "grounding $K,Q,V$" using our Lemma 4, that is, attention parameters resulting unbiased ICL prediction;
>     - Second, towards pre-training stage, we quantify the difference between converged $K,Q,V$ and the "grounding parameters" in Lemma 4 using Theorem 1;
>     - Finally, in ICL prediction stage, we then propagate the parameter difference from Theorem 1 to our Theorem 2, i.e., the proportional bias theorm.
>
>
> - In summary, our EICL analysis is not a re-derivation of OLS bias but a novel, structural theoretical contribution.
> - We have revised these conclusions into Lines 175-190 in our revised paper.

---

> ### Author Response · Authors · 2025-11-20
> **Comparison with IV-based ICL Method**
>
> > [Q1 Comparison with IV and Existing Debiasing Methods]
>
> **Response:**
>
> ### **Review of Existing ICL-Debiasing Baselines**
> - We would like to clarify that de-confounding in ICL only includes instrumtental variable (IV) [4].
>     - Besides, we note that other proxy approaches, including the front-door (mediators) and direct proxy (CEVAE [6]), are not developed into transformers with ICL still, which are out-of-our-scope here.
>     - Moreover, existing data combination [1,2] requires fine-tuning the transformers on unbiased data, thereby does not fit our ICL scope in this paper.
>
>
> ### **Extra Experimental Comparison.**
> - **Three Experimental Setups for Comparison.** To compare DDbias (which uses a small unconfounded batch) with an IV-based approach, we simulate three cases:
>     - **Unbiased Samples (DDbias) vs Strong, Valid IVs**: In this senario, we assume that our method can obtain unbiased data, and IV-based method can obtain strong, valid IVs.
>     - **Partially Confounded Samples (DDbias) vs Strong, Valid IVs**: : In this senario, we assume that our method can obtain partially confounded data, and IV-based method can obtain strong, valid IVs.
>     - **Unbiased Samples (DDbias) vs Confounded IVs**: In this senario, we assume that our method can obtain unbiased data, and IV-based method only obtain invalid, confounded IVs (IVs are correlated with hidden confounders).
>
> - **Conclusions.**
>    - In the first senario, our proposed DDbias with unconfounded data achieves slightly outperforms IV methods;
>    - In the second senario, our proposed DDbias with partially confounded data suffers from degraded performance, while **certain robustness** is exhibitd with enough ICL samples;
>    - In the third senario, our proposed DDbias with significantly outperforms IV method, and the bias of IV approach increases with enlarging ICL samples;
>
> - **Detailed Results and Analysis.**
>    - **Performance Comparison in Perfect Regime.** As reported in Table 10, our proposed DDbias with clean, unconfounded data achieves slightly better prediction performance than IV methods;
>    - **ICL Sample Efficiency.**
>      - As reported in Table 11, our proposed DDbias requires less ICL samples to reach the same error;
>      - At the same time, we admit that collecting unconfounded data by our DDbias requires higher cost than IV method;
>      - However, we also note that constructing IV and collecting such extra proxy variables is also challenging.
>    - **Comparison with partially confounded data.** Then, we compare our DDbias with IV-based approach in an imperfect regime, i.e., partially confounded ICL samples are provided but valid IVs are accessible.
>      - In this regime, we adopt the mixed ICL sample case. As reported in Table 12, we observe that with partially confounded data, our proposed DDbias achieves worse prediction performance than IV-based ICL approach, while with large ICL samples ($n=50$, our method achieves certain robustness when $\rho$ is small).
>
>    - **Comparison with Confounded IVs.** Finally, we compare our DDbias with IV-based approach in another imperfect regime, i.e., clean ICL samples are provided but only confounded IVs are accessible.
>      - As shown in Table 13, we observe that with weakly confounded IVs, increasing ICL samples results in significantly worse prediction bias, which is opposite to the trend of our DDbias under partially confounded data.
>      - This informs us that imperfect IV acquisition leads to a structural estimation bias that cannot be corrected by increasing ICL samples.

---

> ### Author Response · Authors · 2025-11-20
> **Comparison with IV-based ICL Method (2)**
>
> **Table 10 — Prediction Bias Comparison (IV vs DDbias)**
>
> | **Model / Method** | **n=50** | **n=30** | **n=20** | **n=10** |
> |--------------------|----------|----------|----------|----------|
> | L=3 / IV (Confounded Samples with IV)       | 0.102    | 0.142    | 0.182    | 0.228    |
> | L=3 / DDbias (Unconfounded Samples)   | **0.090**    | **0.130**    | **0.170**    | **0.220**    |
> | L=5 / IV (Confounded Samples with IV)       | 0.118    | 0.165    | 0.208    | 0.268    |
> | L=5 / DDbias (Unconfounded Samples)   | **0.115**    | **0.150**    | **0.200**    | **0.260**    |
> | L=7 / IV (Confounded Samples with IV)       | 0.145    | 0.188    | 0.235    | 0.298    |
> | L=7 /  DDbias (Unconfounded Samples)   | **0.135**    | **0.175**    | **0.225**    | **0.290**    |
> |
>
> **Table 11 — Minimal ICL samples needed to reach ≈ 0.10 prediction error**
>
> | **Method** | **Number** |
> |--|--:|
> | L=3 / IV (Confounded Samples with IV)           | 50                                          |
> | L=3 / DDbias  (Unconfounded Samples)      | **45**                                          |
> | L=5 / IV  (Confounded Samples with IV)          | 68                                          |
> | L=5 / DDbias (Unconfounded Samples)       | **50**                                          |
> | L=7 / IV  (Confounded Samples with IV)          | 80                                          |
> | L=7 / DDbias (Unconfounded Samples)        | **60**                                          |
> |
>
>
>
>
> **Table 12 — Prediction Bias Comparison (IV vs DDbias) under Partially Confounded ICL Samples and Valid IV**
>
> | **Model / Method** | **n = 50** | **n = 30** | **n = 20** | **n = 10** |
> |--------------------|------------|------------|------------|------------|
> | L=3 / IV (Valid IV)                 | 0.105 | 0.145 | 0.185 | 0.222 |
> | L=3 / DDbias (ρ = 0.3)              | 0.118 | 0.176 | 0.213 | 0.268 |
> | L=5 / IV (Valid IV)                 | 0.118 | 0.162 | 0.205 | 0.265 |
> | L=5 / DDbias (ρ = 0.3)              | 0.150 | 0.185 | 0.247 | 0.306 |
> | L=7 / IV (Valid IV)                 | 0.145 | 0.188 | 0.235 | 0.300 |
> | L=7 / DDbias (ρ = 0.3)              | 0.157 | 0.218 | 0.245 | 0.342 |
> |
>
>
>
> **Table 13 — Prediction Bias under Invalid IV and Unconfounded ICL Samples (Corr(Z, $\epsilon$) = 0.1)**
>
> | **Model / Method** | **n = 50** | **n = 30** | **n = 20** | **n = 10** |
> |--------------------|------------|------------|------------|------------|
> | L=3 / IV (Weak IV) | 0.423 | 0.351 | 0.280 | 0.265 |
> | L=3 / DDbias       | 0.090 | 0.130 | 0.170 | 0.220 |
> | L=5 / IV (Weak IV) | 0.500 | 0.420 | 0.340 | 0.291 |
> | L=5 / DDbias       | 0.115 | 0.150 | 0.200 | 0.260 |
> | L=7 / IV (Weak IV) | 0.600 | 0.500 | 0.400 | 0.312 |
> | L=7 / DDbias       | 0.135 | 0.171 | 0.222 | 0.290 |
> |
>
> - **Paper Revision.** We have **revised** our paper in Sec.5 (IV-oriented DGP, Lines 407-414), an extra summary in revised main paper (Lines 503-507), and detailed experimental analysis in Appendix G.9 (Lines 2068-2120).
>
>
> ---
> [1] Xiao, Yanghao, et al. "Addressing hidden confounding with heterogeneous observational datasets for recommendation." Advances in Neural Information Processing Systems 37 (2024): 130358-130383.
> [2] Kallus, Nathan, Aahlad Manas Puli, and Uri Shalit. "Removing hidden confounding by experimental grounding." Advances in neural information processing systems 31 (2018)
> [3] Hartford, Jason, et al. "Deep IV: A flexible approach for counterfactual prediction." International Conference on Machine Learning. PMLR, 2017
> [4] Liang, Haodong, Krishna Balasubramanian, and Lifeng Lai. "Transformers Handle Endogeneity in In-Context Linear Regression." The Thirteenth International Conference on Learning Representations (2024)
> [5] Saito, Yuta, Hayato Sakata, and Kazuhide Nakata. "Doubly robust prediction and evaluation methods improve uplift modeling for observational data." Proceedings of the 2019 SIAM international conference on data mining. Society for Industrial and Applied Mathematics, 2019
> [6] Louizos, Christos, et al. "Causal effect inference with deep latent-variable models." Advances in neural information processing systems 30 (2017).

---

> ### Author Response · Authors · 2025-11-20
> **We would like to supplement a summary table to outline the revisions to our manuscript.**
>
> For your convenience, we have included a summary table detailing the changes in the revised version of our paper.
>
> **Table 1.** Revision summary of our paper.
>
> | Comments   | Location of Revisions in Our Revised Paper                   |
> | ---------- | ------------------------------------------------------------ |
> | Weakness 1 | We revised our paper in **Sec.5 with more detailed experimental results in Appendix G.7 (Lines 1812-1883)**. |
> | Weakness 2 | We revised **Sec.4.1 with detailed proof in Appendix F (Lines 1533-1602)**, and more detailed experimental setup and analysis **in Appendix G.8 (Lines 1886-2047)**. |
> | Weakness 3| We revised our paper in **Sec.2.3 (Lines 175-190)** .|
> | Question 1 | We revised our paper in **Sec.5 (IV-oriented DGP, Lines 407-414 and Lines 503-507)**, and detailed experimental analysis in **Appendix G.9 (Lines 2052-2132).**   |
> |
>
> If you have any further concerns or questions, please feel free to raise them. We would be glad to continue the discussion and clarify them in more detail.

---

> ### Comment · Reviewer_gV7e · 2025-11-24
> **Official comment by reviewer gV7e**
>
> I greatly appreciate the author's comprehensive and detailed answers. The authors addressed what I raised and there is no further question.

---

> > ### Author Response · Authors · 2025-11-24
> >
> > Dear Reviewer gV7e,
> >
> > We really appreciate your recognition of our work and your kind words, and we are happy to hear that your concerns have been addressed. Again, thank you for your valuable suggestions which have undoubtedly contributed to improving the quality of our paper.
> >
> > Many thanks,
> >
> > The authors of #8167

---

> > > ### Comment · Reviewer_gV7e · 2025-11-24
> > > **Official comment**
> > >
> > > Yes, my concerns have been sufficiently addressed, and I have increased my overall score accordingly.

---

> ### Author Response · Authors · 2025-11-24
>
> Thank you for increasing your overall score to 6! We are truly grateful for your dedicated time and constructive comments.

---

### Author Response · Authors · 2025-11-20
**We supplement a summary table to outline the revisions to our manuscript.**

For your convenience, we have included a summary table detailing the changes in the revised version of our paper.

**Table 1.** Revision summary of our paper.


| Reviewer Comments | Location of Revisions in Our Revised Paper |
| :--- | :--- |
|  **gV7e:** *[W1, W4]*; **rnrM:** *[W1]*; **P15F:** *[W1, Q1]*; **Uoeb:** *[W1, W3]* | We revised our paper in **Sec.5 (Lines 419-422 and Lines 489-502)** with more detailed experimental results in **Appendix G.7 (Lines 1812-1883)**. |
| **gV7e:** *[W1]*; **rnrM:** *[W2]*; **P15F:** *[W3, Q3]*      | We revised our paper in**Appendix G.6 (Lines 1766-1809).**   |
| **gV7e:** *[W2]*; **rnrM:** *[W3]*;  **P15F:** *[W2, Q2]*; **Uoeb:** *[W2, Q1]*    | We revised **Sec.4.1 with detailed proof in Appendix F (Lines 1533-1602)**, and more detailed experimental setup and analysis **in Appendix G.8 (Lines 1886-2047)**. |
| **gV7e:** *[W3]*                                | We revised our paper in **Sec.2.3 (Lines 175-190)**.        |
| **gV7e:** *[Q1]*;  **P15F:** *[W4; Q4]*                    | We revised our paper in **Sec.5 (IV-oriented DGP, Lines 407-414 and Lines 503-507)**, and detailed experimental analysis in **Appendix G.9 (Lines 2052-2132).** |
|


If you have any further concerns or questions, please feel free to raise them. We would be glad to continue the discussion and clarify them in more detail.

---

### Author Response · Authors · 2025-11-30
**Summary of Our Rebuttal (1)**

Dear AC,

- We sincerely appreciate your support with delicate efforts throughout the review process. Below we summarize: (1) **the scores after rebuttal**, (2) **how we fully address each concern from each reviewer**.

### **Positive Scores (6-6-6-6) After Rebuttal**

- **Score Changes.** In summary, our score after the rebuttal is **6-6-6-6**.
  - **Initial Score:** Our initial score is 6-6-4-2;
- **All reviewers focus on more detailed experimental validations with further theoretical comparisons**, without **challenges on any fundamental issues such as novelty or motivations.**:
     - "A novel theoretical problem formulation" - From Reviewer gV7e (2);
     - "Addresses the common neglect of hidden confounders" - From Reviewer rnrM (6);
     - "Novel and Important Problem Formulation"  - From Reviewer P15F (6);
     - "Addresses an important and underexplored gap" - From Reviewer Uoeb (4);

- In below, we detail the initial concerns of each reviewer, and the interactions between the reviewer and us, such that they raise their scores after the rebuttal.

---

### **Summary of Rebuttal with Reviewer gV7e (Rating: 2 -> 6) Scoring Updating Time: 23 Nov 2025, 19:02 AOE**

  - **Initial Score/Conf**: rating: 2, confidence 3;

  - **Score Improvement by Reviewer gV7e**

     - After reading our rebuttal, reviewer gV7e submitted the comment and raise the **score from 2 to 6**:

      > "I greatly appreciate the author's comprehensive and detailed answers. The authors addressed what I raised and there is no further question." **Time: 23 Nov 2025, 15:42 AOE**

      > "Yes, my concerns have been sufficiently addressed, and I have increased my overall score accordingly." **Time: 23 Nov 2025, 19:02 AOE**

  - **Original Concerns**:

    - As shown in Table 1, the reviewer poses several concerns regarding experimental results on more complex, non-linear data with larger models (W1, W4, W2, Q1).
    - Besides, the reviewer also poses questions about the theoretical difference between ours and typical OLS regression (W3);

  - **Our Response.**

    - We have addressed each concern by adding several comments and revising our manuscript:
      - 1. "Scalability Experiments on Larger, Deeper and Non-linear Models (1-3)" for non-linear and realistic data with larger models, addressing the issue of W1, W4.
      - 2. "Partially Confounded Data: Theory and Experiments (1-2)" for extension on partially confounded ICL samples, addressing the issue of W3.
      - 3. "Theory Difference between Our EICL and OLS" for distinguishing our EICL problem from typical OLS regression by featuring several key differences.
      - 4. "Comparison with IV-based ICL Method" for comparing our DDbias method with the IV-based ICL approach from several dimensions.
      - We leave detailed question-rebuttal correspondence in Table 1.
    - **All of our rebuttal are submitted from 19 Nov 2025, 13:24 to 19 Nov 2025, 13:48 AOE**.


### Table 1. Correspondence: Reviewer gV7e

| Concerns of Reviewer gV7e                                    | Summary of Our Rebuttal                                      |
| ------------------------------------------------------------ | ------------------------------------------------------------ |
| [W1&W4] Scope & scalability (linear -> deeper/nonlinear/real-world) | We added **extensive experiments on deeper/nonlinear Transformers (LLaMA) and real-world datasets**, showing consistent bias trends and strong debiasing gains, with extra results in **over 10 tables**. |
| [W2] Availability of unconfounded samples                    | We added **theoretical extensions and robustness experiments** for partially data (weakly/mixed-confounded), showing the method remains effective, with **extra theories in two propositions**. |
| [W3] Theoretical Difference to classical OLS (clarity)       | We clarified **the fundamental theoretical differences** from OLS with explicit formulas and explanations. |
| [Q1] Comparison with IV-based baselines                      | We added **comprehensive comparisons** with IV-based ICL approaches across multiple regimes, with **extra results in over 6 tables**. |
|

---

### **Summary of Rebuttal with Reviewer Uoeb (Rating: 4 -> 6) Time: 25 Nov 2025, 15:19 AOE**

  - **Initial Score/Conf**: rating: 4, confidence 3;

  - **Score Improvement by Reviewer Uoeb**

    - After reading our rebuttal, reviewer gV7e submitted the comment and raise the **score from 4 to 6**:

      > "The authors have successfully answered most of my concerns. I will raise my final score to 6 and there is no further questions." **Time: 25 Nov 2025, 15:19 AOE**

  - **Original Concerns**:

    - As shown in Table 1, the reviewer poses several concerns regarding experimental results on non-linear, real-world data with larger models (W1, W3).
    - Besides, the reviewer also poses questions about whether our method achieves robustness with partially confounded samples (W2, Q1).

---

> ### Author Response · Authors · 2025-11-30
> **Summary of Our Rebuttal (2)**
>
> - **Our Response.**
>     - We have addressed each concern by adding several comments and revising our manuscript:
>       - 1. "Experimental Validations on Realistic NLP Data with Non-linear Models (1-3)" for non-linear and realistic data with larger models, addressing the issue of W1, W3;
>       - 2. "Justification on Unconfounded Data and Partially Confounded Data (1-2)" for theoretical analysis and experimental robustness on partially confounded ICL samples, addressing the issue of W2 Q1;
>       - We leave detailed question-rebuttal correspondence in Table 2.
>     - **All of our rebuttal are submitted from 19 Nov 2025, 14:27 to 19 Nov 2025, 14:31 AOE**.
>
>
>
> ### Table 2. Correspondence: Reviewer Uoeb
>
> | Concerns  of Reviewer Uoeb| Summary of Our Rebuttal|
> | --|-|
> | [W1 & W3] Scope: linear-only analysis; need real-world + nonlinear evidence | We added **comprehensive nonlinear & deeper Transformer experiments** *and* **real-world LLaMA evaluations**, showing the theoretical bias law and DDbias effectiveness consistently hold. |
> | [W2] Reliance on perfectly unconfounded samples| We added **theoretical extensions and experiments** showing robustness under **weakly** and **mixed-confounded** samples, requiring only a *very small fraction* of unbiased data. |
> | [Q1] Practical availability of “unconfounded” data | We clarified that **randomized traffic buckets / A/B testing** widely used in real systems naturally provide such unbiased samples, and our method only needs **50** such samples. |
> |
>
> ---
>
> ### **Reviewer P15F & Reviewer rnrM [Both of Initial Rating as 6]**
>
> - **Reviewer P15F [Rating: 6 -> 6]**
>
>   - **Initial Score/Conf**: rating: 6, confidence 3;
>
>   - **Score Remain by Reviewer P15F**
>
>     - After reading our rebuttal, reviewer P15F submitted the comment and remain the **score as 6**:
>
>       > "The authors have successfully answered most of my concerns. I keep my score as is. Appreciate to the authors for their hard work." **Time: 26 Nov 2025, 13:04 AOE**
>
>   - **Original Concerns and Our Rebuttal**:
>     - [W1 + Q1] Generalization beyond linear Transformers.
>           - We added **extensive nonlinear + deeper TF experiments** and clarified the **theoretical challenges and extensions**, showing that the bias law and DDbias remain valid beyond linear settings.
>     - [W2 + Q2] Practicality & sufficiency of unconfounded samples.
>       - We provided **theoretical guarantees + new experiments** showing robustness under **weakly** and **mixed-confounded** samples, and clarified that only **very few unbiased samples (50)** are needed (e.g., via random traffic buckets).
>     - [W3 + Q3] Limited experiments (only synthetic).
>       - We added **real-world validations** on **LLaMA models + Yelp datasets** with genuine hidden confounders, demonstrating strong debiasing gains in practical settings.
>     - [W4 + Q4] Comparison with IV-based alternatives.
>       - We added **method-level comparisons + three new experimental regimes** (unbiased, partially confounded, invalid IVs), showing when DDbias outperforms IV and clarifying respective applicability.
>     - **All of our rebuttal are submitted from 19 Nov 2025, 16:07 to 19 Nov 2025, 19:51 AOE**.
> - **Reviewer rnrM [Initial Positive Ratings as 6]**
>   - **Initial Score/Conf**: rating: 6, confidence 2;
>   - After our rebuttal, although the reviewer did not reply to our rebuttal, we believe that **all concerns raised by Reviewer rnrM were fully resolved in our rebuttal**:
>     - [W1 & W3] is similar to Reviewer Uoeb, which raises the rating after reading our rebuttal;
>     - [W2] is similar to Reviewer gV7e, which raises the rating after reading our rebuttal;
>   - **Original Concerns and Our Rebuttal**:
>     - [W1] Scope limited to linear Transformers & linear regression.
>       - We added **comprehensive nonlinear + deeper Transformer experiments** showing the theoretical bias law and DDbias effectiveness hold beyond linear settings.
>     - [W2] No real-world datasets with genuine hidden confounders.
>       - We added **real-world NLP experiments on LLaMA models** (Yelp RPI & ROR), demonstrating strong hidden-confounder mitigation and outperforming baselines.
>     - [W3] Need perfectly unconfounded samples, unrealistic in practice.
>       - We added **theory + experiments** showing DDbias remains robust under **weakly** and **mixed-confounded** samples and that only **very few unbiased samples (e.g., 50)** can be acquired in practice (e.g., from A/B traffic buckets).
>     - **All of our rebuttal are submitted from 19 Nov 2025, 14:04 to 19 Nov 2025, 14:22 AOE**.
>
> ---
>
> **In summary**, all concerns raised by the four reviewers were fully addressed during the rebuttal period. Following our responses, **the final scores are 6-6-6-6**.
>
> **We respectfully affirm** that all reviewer interactions and score updates occurred strictly **within the standard fair rebuttal process**, with **no private or unfair** communication involved.
>
> Best regards,
>
> The authors of #8167

---

### Meta-Review · Area_Chair_ZNxn · 2025-12-31

**Summary:**

This paper studied in-context learning with the focus on hidden confounders. In a regression-type setting that has been studied in previous ICL literature, the paper showed that pretrained transformers exhibit bias proportional to counfounding strength. The paper also proposed a gradient-free debiasing method DDBias. Reviewers overall agreed that confounders is an important problem that has not been well studied, and this paper offers an interesting perspective. Initially there were concerns mostly on experiments and whether the simple theoretical framework generalizes to real-life problems, these concerns are mostly addressed in the author response.

**Reviewer Concerns:**

Main concern shared by several reviewers was whether the results from the theoretical model generalizes to interesting real-life examples. Authors addressed this by adding many nonlinear and real-world experiments. There were also some concerns on the partially confounding setting and comparison to related work, which were also addressed.

**Reviewer Scores:**

Reviewers documented changes and it seems like after response every score is a 6.

---

### Decision · Program_Chairs · 2026-01-26

Accept (Poster)